# Motor cortex somatostatin interneurons adaptively shape the structure of action sequences

Jeong Oen Lee[1], Sebastiano Bariselli[1,2,3], Giacomo Sitzia[1,4], Abigail Holder [1] & David M. Lovinger [1] ✉

The brain flexibly reorganizes action sequences to optimize behavioral outcomes through reinforcement learning and adaptive motor control. Although the primary motor cortex (M1) is essential for skill learning and dexterous movement, how cortical microcircuits refine the timing and structure of action sequences remains unclear. We show that M1 somatostatin interneurons (SST-Ins) display synchronized, action-locked calcium activity during acquisition of a lever-press task in freely moving mice, in contrast to the sequential activation of pyramidal neurons. Following extended training under a stable task schedule, SST-IN activity was no longer coupled to action execution. However, when task demands were modified to require faster and more temporally constrained action sequences, SST-IN activity redistributed and correlated with trial-by-trial changes in sequences, rather than diminishing. Inhibition of SST-INs disrupted temporal organization and impaired efficient motor execution. These findings highlight the unexpected role of M1 SST-INs in refining motor programs into more efficient and task-specific structures.

Humans and other animals organize action sequences by integrating discrete movements into cohesive units. With extensive, repeated motor training, these learned sequences become automatized and are executed as unified, holistic actions[1–4]. However, in many other behavioral situations, the organization of action sequences requires flexible modulation of their temporal and structural features depending on task demands[5,6]. This neural capacity for dynamic sequence modulation in response to changing behavioral contexts is essential for producing efficient and adaptive motor behaviors[7–9]. Despite its importance, the neural circuit mechanisms underlying the regulation of temporal and structural aspects of action sequences remain unclear.

Both cortical and subcortical structures, particularly the striatum and primary motor cortex (M1), are implicated in action sequence organization[10]. The striatum is involved in controlling well-learned action sequences by encoding neural signals related to sequence initiation, execution, and termination[2,3,10]. In contrast, M1 exhibits dynamic activity patterns that vary depending on the task demands and the stage of learning[11–14]. M1 may become functionally dispensable for executing overtrained behaviors, as evidenced by lesion studies showing that damage to M1 does not impair the performance of well-practiced routines[2,15]. These findings support the "cortical disengagement" model[16], which proposes that cortical regions are critical for acquiring new skills but become less essential once motor behaviors are extensively trained.

However, other evidence suggests that M1 continues to encode movement or action-sequence-related parameters even after extensive training[17]. For example, primate studies have shown that M1 exhibits anticipatory activity representing memorized action sequences during their execution[18]. Additionally, M1 neurons encode discrete elements of rapid, skilled action sequences, maintaining

[1]Laboratory for Integrative Neuroscience (LIN), National Institute on Alcohol Abuse and Alcoholism, Bethesda, MD, USA. [2]Present address: IRCCS Humanitas Research Hospital, Via Manzoni 56, 20089 Rozzano, Milano, Italy. [3]Present address: Department of Biomedical Sciences, Humanitas University, Via Rita Levi Montalcini 4, 20072 Pieve Emanuele, Milan, Italy. [4]Present address: Department of Neuroscience, Københavns Universitet, København, Hovedstaden, Denmark. ✉e-mail: lovindav@mail.nih.gov

representations of individual movement parameters throughout sequence performance[4]. M1 activity also remains crucial for flexible execution of cue-guided sequences. For instance, cortical lesions impair adaptability to external instructions while sparing the execution of overtrained sequences[19,20]. Thus, it is possible that the neural circuitry in M1 is actively involved in organizing action sequences on a trial-by-trial basis, particularly when tasks demand temporal modulation or flexible reuse of learned action sequences.

Despite extensive research on M1 in general motor skill learning, how distinct neural circuit components within M1 contribute to the temporal and structural modulation of action sequences remains largely unexplored. Pyramidal (PYR) neurons and inhibitory interneurons (INs) constitute[21] key elements of heterogeneous microcircuitry in M1. They orchestrate action learning and execution through dynamic and plastic activity patterns finely tuned to specific movement parameters[22]. In M1 layer 2/3 (L2/3), PYR neurons progressively refine their activity during associative learning[14], encoding both sensory and motor components as learning advances[23]. Also, neighboring cortical INs, particularly somatostatin-positive (SST) interneurons(INs), play a critical role in regulating learning-dependent network plasticity by modulating synaptic and structural dynamics of PYR neurons[24]. Notably, SST IN activity in L2/3 exhibits task-specific modulation correlating with retention and maintenance of previously acquired motor skills[25,26]. Thus, prior work has primarily focused on the role of SST INs in L2/3 during motor skill learning.

In contrast, neural circuits in M1 layer 5/6 (L5/6) are specialized for relaying motor commands to downstream targets such as the brainstem, spinal cord[27], and striatum[11], via pathway-specific subpopulations that independently encode movement direction and amplitude[10]. During action sequence organization, the PYR projection neurons in L5/6 convey sequence-related information; the repetitive components of these sequences are directly regulated via projections to the dorsolateral striatum (DLS), as demonstrated by optogenetic manipulations[28]. Although the activity of L5/6 PYR neurons is strongly influenced by local inhibitory circuits, the functional role of inhibitory interneurons, such as SST INs, in the generation and control of structured action sequences remains unclear. Given the critical role of M1 L5/6 microcircuitry in sequence organization[28], we hypothesize that SST INs play a distinct role in this process. Specifically, we investigate how SST INs in L5/6 shape the temporal and structural features of action sequences on a trial-by-trial basis.

In this study, we investigated single-cell calcium activity of SST INs in M1 L5/6 while freely moving mice progressively reorganized reinforced action programs—from a single lever press (fixed ratio 1, FR1), to four lever presses (fixed ratio 4, FR4), and ultimately to a rapid action sequence requiring four consecutive presses within a specified time window (time-constrained FR4, or TC-FR4). We found that SST INs in layer 5/6 exhibited action-specific calcium activity in a single-lever-press task. To test whether this synchronized SST IN activity did not decrease as the motor program consolidated, we extended the FR1 training while maintaining the same task schedule. We found that the amplitude of synchronized calcium transients in L5/6 SST INs decreased with task consolidation. Conversely, during TC-FR4 training, as mice continuously reorganized more rapid action sequences, SST INs redistributed their activity and maintained a consistent action-locked response during sequence execution. Specifically, we identified two distinct patterns of single SST IN activity that encoded sequence initiation and trial-by-trial sequence duration. Chemogenetic and closed-loop optogenetic inactivation of SST INs interfered with action sequence kinematics and impaired efficient motor program execution. These results highlight the functional contribution of M1 SST INs in actively regulating and maintaining the timing of motor structures, thereby facilitating efficient motor execution and adaptation to varying task demands.

## Results

### Action-locked SST IN network activation emerges upon motor program acquisition

Our experimental setup and representative calcium traces are shown to illustrate raw activity patterns of cortical neurons in relation to the single lever press action (Fig. 1). To examine the activity of identified cortical neuron subtypes during motor program acquisition, we trained mice on a self-paced lever-press task (Fig. 1a). In the first paradigm, mice were trained to perform a single lever press to obtain a reward, known as a fixed-ratio 1 (FR1) schedule. This task was implemented over 7 days to characterize M1 neural responses associated with individual lever presses. The same training was then extended 21 days to examine how neural dynamics changes during the execution of a simple, repeated motor program. In a second set of experiments, mice were trained on a more complex paradigm requiring four lever presses per reward (fixed-ratio 4 or FR4). Initially, the task was performed under a non-time-constrained FR4 schedule (NTC-FR4). The timing requirement was then progressively tightened so that mice were required to complete four or more lever presses within a specific time window, forming a time-constrained FR4 schedule (TC-FR4). This design allowed us to examine how neural activity correlates with the organization of lever presses into structured action sequences. During task performance, we monitored in vivo single-cell calcium activity in the M1 with a miniaturized fluorescence microscope and implantable gradient-index (GRIN) lenses. After viral transduction of the genetically-encoded calcium indicator GCaMP6f, we assessed calcium transients in PYR neurons and SST INs across distinct groups of mice while animals performed a fixed-ratio one (FR1) lever-pressing task (Fig. 1a). From Day 1 (D1) to Day 7 (D7) of training, animals displayed a dramatic increase in lever-press rate (Fig. 1b), indicating the acquisition of the novel motor task. On D7, calcium traces ($\Delta F/F_0$ in %, where $F_0$ indicates fluorescence baseline) of individual SST INs exhibited less complex patterns than calcium traces from PYR neurons (Fig. 1c, d). This SST activity decreased when the animal was not actively involved in the FR1 task (Supplementary Fig. 1a, b). We reconstructed these task-related calcium traces into peak-normalized activity maps (Fig. 1e, f) to identify the timing of peak calcium increases relative to lever press. Consistent with previous studies in head-fixed mice[14,25], the activity pattern of PYR neurons revealed a sequential action-locked activation pattern (Fig. 1e), which resulted in a distributed calcium event probability (Fig. 1g). In contrast, SST INs displayed calcium event probabilities that were highest around lever-presses (Fig. 1h). We then characterized the layer-specificity of action-locked calcium traces ($\Delta F/F_0$ in %) in PYR neurons and SST INs by adjusting the depth of GRIN lens implants and recording neurons in L2/3 and L5/6 of the M1 (Fig. 1i–k).

We found that the action-locked $\Delta F/F_0$ amplitude was higher in SST INs in M1 L5/6 compared to that of SST INs in M1 L2/3 and PYR in both L2/3 and L5/6 (Fig. 1j,k, and Supplementary Fig. 1c,d). Representative raw calcium traces that contribute to the population-averaged $\Delta F/F_0$ profiles of SST INs in L2/3 and L5/6 are shown in Fig. 1l,m,n and Fig. 1o,p,q, respectively. In these example traces, SST activity in layer 2/3 was increased around individual lever presses, but the occurrence and amplitude of the increases showed considerable variability around each press (Fig. 1l). In contrast, activity in layer 5/6 showed more consistent increases, including relatively consistent transient amplitudes, around each press (Fig. 1o). Nevertheless, there are limitations to comparing cell-to-cell activity solely based on $\Delta F/F_0$ values. To examine this characteristic at the network level independently of $\Delta F/F_0$ amplitude, we assessed Pearson's correlation coefficients across cells. This parameter calculates the linear relationship between a single cell calcium trace and all other neuronal traces during a 10 s time window centered around the lever-press. The analysis revealed that both PYR neurons and SST INs in L5/6 exhibit higher cell-to-cell correlation compared to the calcium responses in L2/3

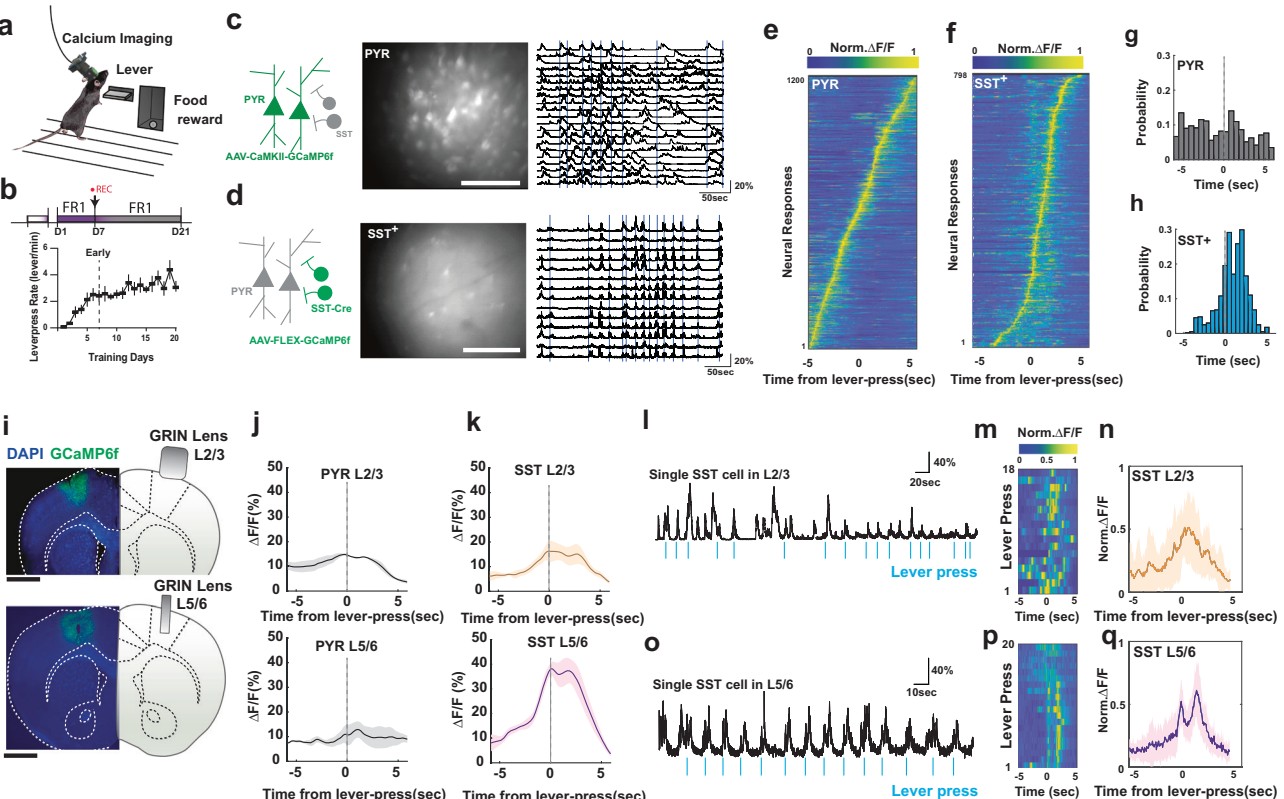

**Fig. 1 | Single-cell and population activity of SST INs in M1 display action-specific calcium response. a** Schematic of operant behavior training and in-vivo calcium activity imaging. **b** FR1 training schedule and the lever-press rate, the lever-press kinematic was stabilized from D7 to D21 ($n = 7$ mice). Error bars denote SEM. **c** Representative raw fluorescence image of CaMKII-driven GCaMP6f expression in M1 PYR neurons (left and mid panels). Scalebars in fluorescence images denote 100 μm. Black traces in the right panel indicate de-synchronized $\Delta F/F_0$ calcium activity associated with lever-press events (vertical blue lines). Experiment was independently repeated 3 times with similar results. **d** Representative raw fluorescence image of Cre-dependent GCaMP6f expression in M1 SST(+) interneurons (left and mid panels) and synchronized calcium activity. Experiment was independently repeated 8 times with similar results. Scalebar: 100 um. **e** Peak normalized $\Delta F/F_0$ activity map of M1 PYR neurons displays sequential activation pattern around the lever-press onset. **f** Peak normalized $\Delta F/F_0$ activity map of SST(+)

interneurons displays a temporally confined activation around the lever-press onset. **g** Calcium event probability distribution of PYR neurons, and (**h**) the same for SST(+) interneurons around the lever-press onset. **i** 1 mm and 0.6 mm GRIN lens were implanted in layer 2/3 and layer 5 respectively, Scalebar: 1.0 mm. **j** Population average of $\Delta F/F_0$ response in PYR neurons in L2/3 (483 neurons, $n = 3$ mice) and L5 (347 neurons, $n = 4$ mice) during lever-press. Shded area denotes SEM. **k** Population average of $\Delta F/F_0$ response in SST INs (L2/3, 117 neurons, $n = 4$ mice) and (L5/6, 103 neurons, $n = 8$ mice) during lever-press. **l** Example of single cell $\Delta F/F_0$ trace of M1 SST interneuron in L2/3. Shded area denotes SEM. **m** Peak normalized $\Delta F/F_0$ activity map of single cell in (**l**) time relative to the lever-press onset. **n** Averaged of peak normalized $\Delta F/F_0$ activity map in (**m**). **o** Example of single cell $\Delta F/F_0$ trace of M1 SST interneuron in L5/6. **p** Peak normalized $\Delta F/F_0$ activity map of single cell in (**o**) time relative to the lever-press onset. **q** Averaged of peak normalized $\Delta F/F_0$ activity map in (**p**).

(Supplementary Fig. 1e,f). It must be noted that placement of the GRIN lens for imaging in L5/6 entails some damage to superficial layers in M1, and this could affect L5/6 SST-IN activity by disrupting cross-layer communication. Thus, the differences in lever-press related SST-IN activity in the different cortical layers must be interpreted with caution. The remainder of the study focused on L5/6, a layer that plays a central role in motor output control but is less well characterized than L2/3 in previous imaging studies[24–26].

## Action-locked M1 SST IN network activity attenuates during motor program consolidation

To understand whether action-locked SST activity would change upon motor program consolidation, we compared neuronal population activity between early (day 7, D7) and extended (day 21, D21) FR1 training in the same experimental subjects (Fig. 2a and Supplementary Fig. 2a). In M1 L5/6 SST INs, we measured single-cell calcium activity (Fig. 2b,c) and observed decreased action-locked fluorescence responses ($\Delta F/F_0$ in %, where $F_0$ indicates fluorescence baseline) at D21 compared to D7 (Fig. 2d-g). We further validated the stability of our $\Delta F/F_0$ measurements over the measurement period to ensure that this action-locked $\Delta F/F_0$ decrease was not due to long-term fluorescence

degradation. For this, we assessed the maximum $\Delta F/F_0$ peaks and action-locked $\Delta F/F_0$ responses across training sessions, normalized to D1. While the maximum $\Delta F/F_0$ and spontaneous $\Delta F/F_0$ response normalized to D1 did not change across training sessions (Supplementary Fig. 2b,c,d,e), the normalized action-locked $\Delta F/F_0$ responses decreased over training sessions (Supplementary Fig. 2f) and were smaller at D21 compared to D7 (Supplementary Fig. 2g). These results indicate that, in these experimental settings, long-term imaging and training did not decrease the overall amplitude of calcium transients, but rather resulted in the selective reduction of action-locked SST IN activation.

By detecting peak locations of $\Delta F/F_0$ responses, we also analyzed discrete calcium events (Fig. 2h,i) from individual SST INs. At a single-cell level, action-locked calcium event frequency decreased from D7 to D21 (Fig. 2j-m), with no changes in the temporal distribution of event probability (Fig. 2n,o). We also observed a significant decrease in Pearson cell-to-cell correlation coefficients of SST activity across subsequent lever presses at D21 compared to D7 (Fig. 2p,q). Thus, SST IN activity exhibited a progressive decrease in action-locked calcium responses (Supplementary Fig. 2f). These results indicate that when the task schedule remained consistent during extended training, the action-related activation of M1 SST INs was not maintained.

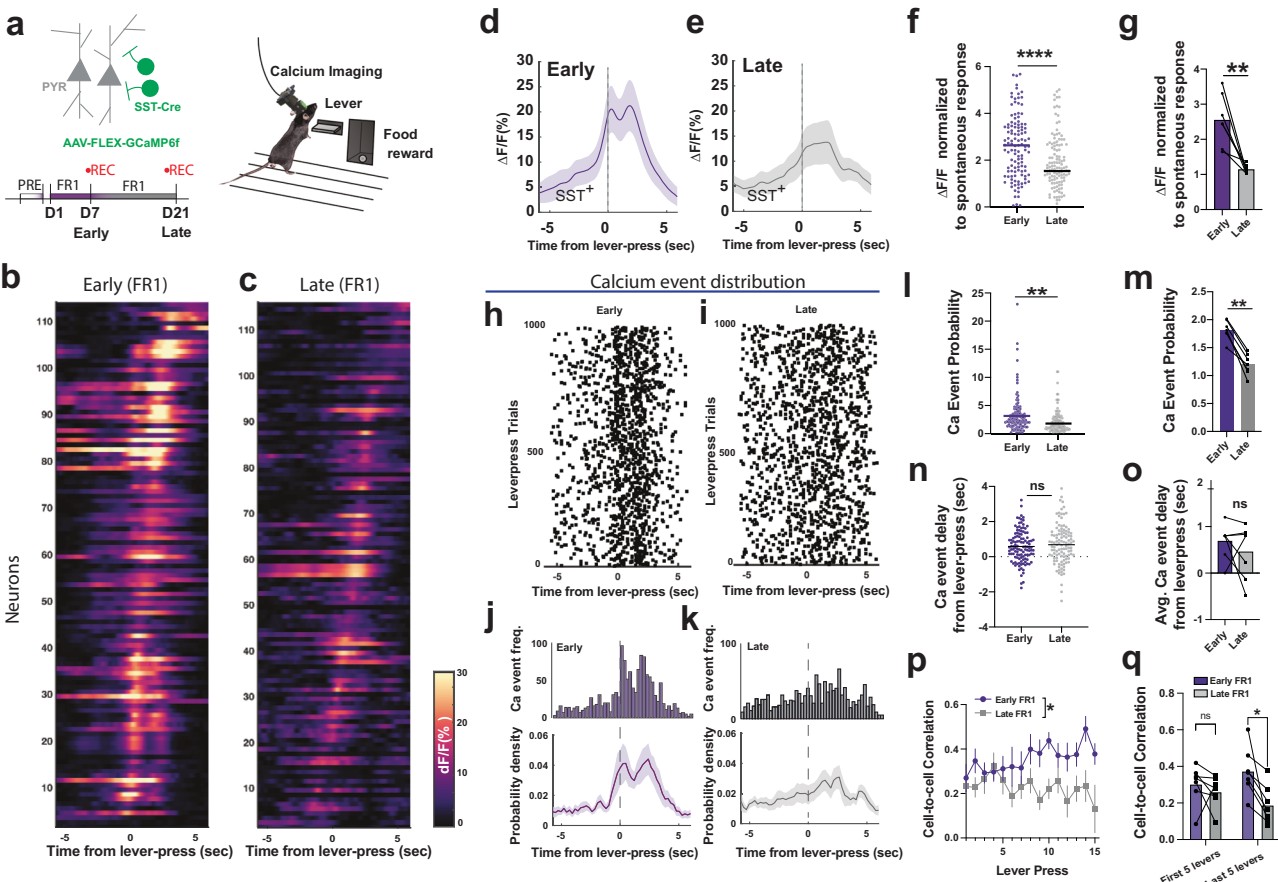

**Fig. 2 | Action-locked SST-IN activity in M1 L5/6 correlates with action acquisition and consolidation. a** FR1 training schedule with imaging M1 SST calcium activity. **b, c** Calcium activity traces ($\Delta F/F_0$) of SST interneurons that are time-locked to the lever press on FR1 schedule D7 early ($n = 117$ cells) and D21 late ($n = 119$ cells). **d, e** Averaged $\Delta F/F_0$ (%, $F_0$ is spontaneous baseline) of SST interneurons time-locked to the lever press in early training (D7) and press in late training (D21). **f** Action-specific $\Delta F/F_0$ peak of individual SST+ interneurons, comparison between early (D7, $n = 117$ cells) and late (D21, $n = 119$ cells) training (two-tailed unpaired t-test, Mann-Whitney, ****$p = 0.0001$), The peak $\Delta F/F_0$ is time-locked to the lever press and normalized to spontaneous $\Delta F/F_0$ peak response. **g** The same action-specific $\Delta F/F_0$ peak in each animal, spontaneous $\Delta F/F_0$ peak response, comparison between early (D7) and late (D21) training ($n = 7$ mice, two-tailed paired t-test, $p = 0.0036$). **h, i** Calcium event distribution in D7 and D21 time-locked to the lever press, each dot denotes the retrieved peak location of $\Delta F/F_0$ traces, 1000 lever-press trials with corresponding calcium traces are randomly selected and shuffled ($n = 7$ mice). **j** Accumulated distribution of calcium event frequency in (**i**) and averaged probability density function (bottom panel, $n = 7$ mice) in D7. **k** Accumulated distribution of calcium event frequency in (**j**) and averaged probability density function (bottom panel, $n = 7$ mice) in D21. **l, m** Action-specific

calcium event probability in individual SST+ interneurons (each dots denote cells, $n = 117$ cells in D7, $n = 119$ cells in D21, two-tailed Mann-Whitney U test, $p < 0.0001$) and animals (each dots denote animals, $n = 7$ mice, two-tailed paired t-test, $p = 0.0001$), normalized to the average calcium event probability in spontaneous activity. **n, o** Average time delay of calcium event from the lever-press onset in individual cells is not significantly different (each dots denote cells, $n = 117$ cells in D7, $n = 119$ cells in D21, two-tailed unpaired t-test, $p = 0.3684$) and animals (each dots denote animals, $n = 7$ mice, two-tailed paired t-test, $p = 0.3864$). The shaded area denotes SEM. **p** Correlation coefficients of action-specific SST calcium activity across lever-press trials were higher in early D7 training, while decreased in late D21 training. (two-way ANOVA, mixed-effects model, D7 early vs D21 late effect, $n = 7$ mice, $p = 0.0421$, $F_{(1,6)} = 6.627$). **q** Correlation coefficient comparison between the first 5 levers and last 5 levers in early (D7) and late (D21) training sessions. In D21, the correlation coefficients of SST network activity are significantly lower on the last 5 lever-press in D7 early training compared to those in D21 late training (two-way ANOVA, repeated measure, D7 early vs D21 late effect, $p = 0.0349$, Sidak multiple comparison, last 5 levers of D7 early vs D21 late, $n = 7$ mice, $p = 0.0374$). In all plots, the shaded area and error bars denote SEM.

## M1 SST IN network activity is redistributed during motor program reorganization

Next, we examined how stabilized SST network activity in M1 L5/6 changes as animals reorganized simple actions into more complex action sequences, such as time-constrained rapid sequential lever presses. First, we trained animals for seven days on an FR1 schedule followed by 7 days of non-time-constrained FR4 (NTC-FR4) training[3] to allow for the acquisition and consolidation of baseline motor programs (Supplementary Fig. 3a,b). Next, we trained mice to progressively increase their lever-press rate and eventually construct a sequence of 4 or more consecutive lever presses within progressively shorter time limits (4, 2, and 1 sec) to obtain a reward. This time-constrained schedule (TC-FR4) increased lever-press frequency across sessions (Fig. 3a). At the 4 Hz schedule in TC-FR4 training, mice began to organize lever pressing

into structured sequences of four consecutive presses within 1 s (Fig. 3b–d). We defined a unit sequence (SEQ) as a series of consecutive lever presses followed by either a head entry or a pause longer than 3 s (Fig. 3b), then analyzed kinematic parameters within a SEQ. In addition to the overall increase in lever press rate, the TC-FR4 training schedule led mice to increase the frequency (Fig. 3c) and the number (Fig. 3d) of lever presses within a SEQ compared to the last session of NTC-FR4. This indicates alterations in SEQ structure that were not observed across NTC-FR4 training (Supplementary Fig. 3c,d). To simplify the representation of SEQ kinematic parameters, we defined the sequence efficiency (SEQ efficiency, Fig. 3e) as a function of lever-press rate and number of lever presses in a SEQ. As such, efficient SEQs (e.g., fast-quadruple lever presses at >4 Hz) were always followed by reward delivery and projected into a bright region on the color-coded map

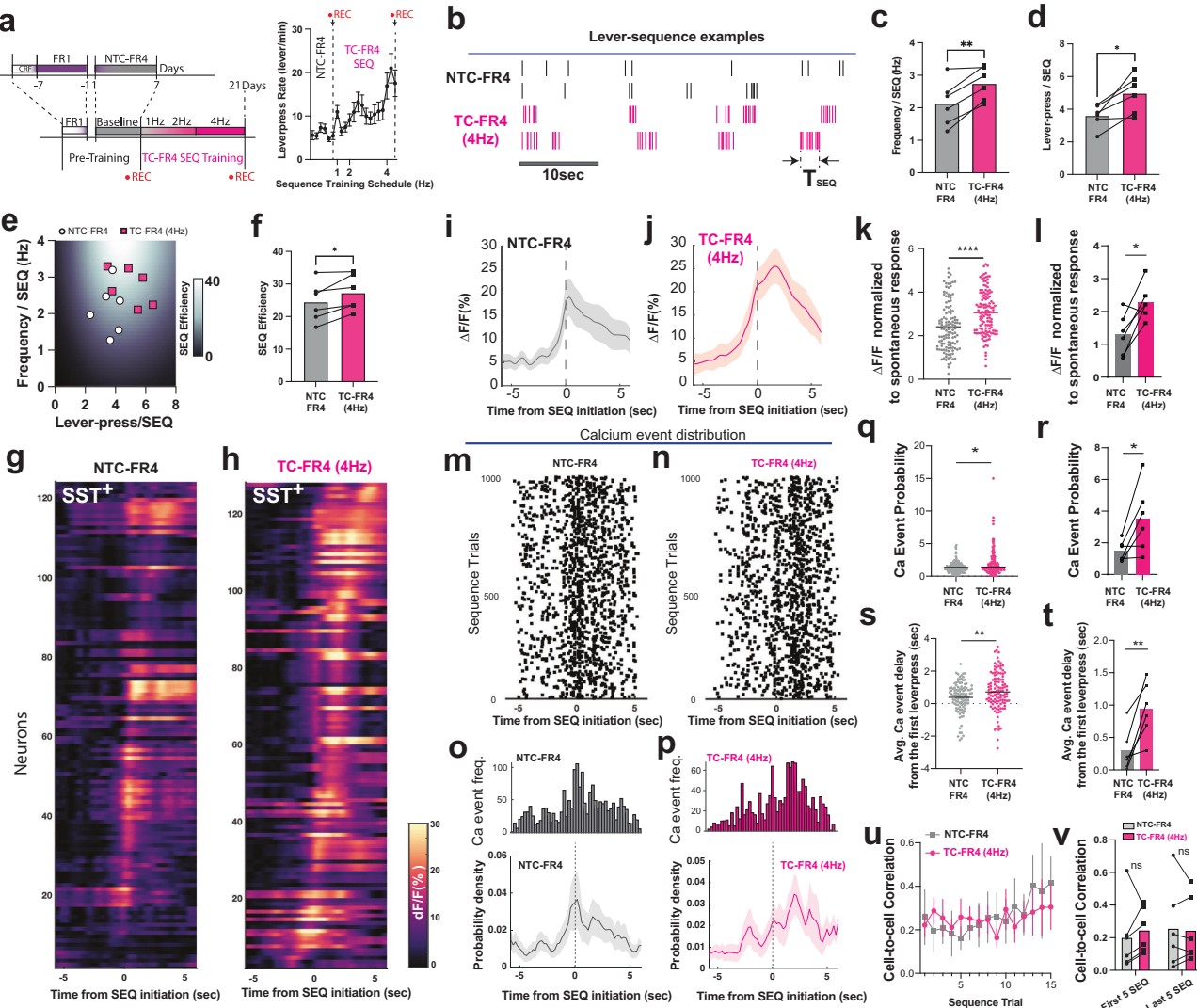

**Fig. 3 | SST IN activity in M1 L5/6 is redistributed during execution of complex sequence. a** Time-constrained FR4 (TC-FR4) sequence lever-press training schedule led mice to progressively increase lever-press rate in a session. Non-time-constrained criteria were assigned between D1 and D7 for the FR4 schedule (NTC-FR4). From D8 to D21, mice had to press the lever 4 consecutive times within 4 s (1 Hz), 2 s (2 Hz), 1 s (4 Hz). **b** Example lever-press patterns are displayed as black bars (NTC-FR4 schedule), and magenta bars (TC-FR4 schedule, 4 Hz). **c** The lever-press frequency (Hz) of a unit sequence trial significantly increased ($n = 6$ mice, two-tailed paired t-test, **$p = 0.0050$ from NTC-FR4 to 4Hz-TC-FR4 schedule. **d** The lever-press number within a unit sequence significantly increased ($n = 6$ mice, two-tailed paired t-test, *$p = 0.0126$). **e** Lever-press number and frequency (Hz) are projected onto the sequence efficiency coefficient map. A higher coefficient indicates a structured and efficient lever-press sequence at the 4 Hz schedule. **f** The sequence structure coefficients between the NTC-FR4 schedule and 4Hz-TC-FR4 schedule. ($n = 6$ mice, two-tailed paired-t test, $p = 0.0244$). **g** Fluorescence calcium activity traces ($\Delta F/F_0$) of SST interneurons ($n = 126$ cells), time-locked at the sequence initiation, during the NTC-FR4 schedule. $F_0$ is the spontaneous fluorescence baseline. **h** The same data ($n = 128$ cells) from the 4Hz-TC-FR4 training. **i** Averaged $\Delta F/F_0$ (%, $F_0$ is spontaneous baseline) of SST interneurons time-locked to the sequence initiation in the NTC-FR4 training schedule. **j** Averaged $\Delta F/F_0$ (%, $F_0$ is spontaneous baseline) of SST interneurons time-locked to the sequence initiation in the 4Hz-TC-FR4 training schedule. **k** Action-specific $\Delta F/F_0$ peak of individual SST interneurons, normalized to the spontaneous $\Delta F/F_0$ peak response, comparison between the NTC-FR4 schedule ($n = 126$ cells) and 4Hz-TC-FR4 SEQ schedule

($n = 128$ cells, two-tailed unpaired t-test, p < 0.0001). **l** Action-specific $\Delta F/F_0$ peak in each animal, normalized to the spontaneous $\Delta F/F_0$ peak response, comparison between NTC-FR4 and 4Hz-TC-FR4 training ($n = 6$ mice, two-tailed paired t-test, $p = 0.0171$). **m, n** Calcium event distribution in NTC-FR4 schedule and 4Hz-TC-FR4, each dot denotes the retrieved peak location from $\Delta F/F_0$ traces, 1000 lever-press trials were randomly selected ($n = 6$ mice). **o** Accumulated distribution of calcium event frequency in (m) and corresponding averaged probability density function. **p** Accumulated distribution of calcium event frequency in (n) and corresponding averaged probability density function. **q, r** Action-specific calcium event probability in individual SST+ interneurons (each dot in (m), two-tailed Mann-Whitney U test, $p = 0.0170$) and animals (each dot in panel n, two-tailed paired t-test, $p = 0.0348$), normalized to the average calcium event probability in spontaneous activity. **s** Average time delay of calcium event from the lever-press sequence onset in individual cells during NTC-FR4 schedule ($n = 113$ cells) and 4Hz-TC-FR4 schedule ($n = 120$ cells, two-tailed unpaired t-test, Mann-Whitney $p = 0.0101$). **t** The same time delay of calcium event from animal average ($n = 6$ mice, two-tailed paired-t test, $p = 0.0071$). **u** Correlation coefficient across lever-press trials and training periods was not significantly different (Two-way ANOVA mixed-model, $n = 6$ mice, sequence trial effect $p = 0.5448$, training schedule effect, $p = 0.7550$) between NTC-FR4 schedule and 4Hz-TC-FR4 schedule. **v** Correlation coefficient between the training schedules was not significantly different in the animal average (two-way ANOVA repeated measures, $n = 6$ mice, sequence trial effect $p = 0.3750$, training schedule effect, $p = 0.7261$). In all plots, the shaded area and error bars denote SEM.

(Fig. 3e). In contrast, inefficient SEQs (e.g., slow-and-many lever presses or fast-and-few lever presses) have low SEQ efficiency coefficients and fall in the darker regions of the colormap. As mice learned to perform 4Hz-TC-FR4 SEQs, the SEQ efficiency increased compared to NTC-FR4 training sessions (Fig. 3f), indicating successful action sequence reorganization. Concurrent with motor program reorganization, the action-locked single-cell or population fluorescence responses of SST INs increased during 4Hz-TC-FR4 compared to NTC-FR4 (Fig. 3g,h,i,j,k,l). Note that when the SEQ efficiency remained unchanged during the NTC-FR4-training (Supplementary Fig. 3e,f), there was a reduction in the population fluorescence response (Supplementary Fig. 3g-j), which indicated SST INs did not continue to exhibit action-related calcium response during the NTC-FR4-training.

We further examined the distribution characteristics of calcium events of SST INs. In the 4Hz-TC-FR4 schedule, the frequency of action-locked calcium events increased compared to the NTC-FR4 schedule (Fig. 3m-r). Importantly, the action-locked calcium event probability during the 4Hz-TC-FR4 training schedule was significantly delayed (Fig. 3s,t) compared to the NTC-FR4 schedule. Despite this calcium event redistribution, we did not detect differences in Pearson correlation coefficient of SST IN single-cell activities between 4Hz-TC-FR4 and NTC-FR4 training sessions (Fig. 3u,v). Consequently, these data demonstrate that, while the temporal distribution of M1 SST activity shifts during the reorganization of complex motor programs, the cell-to-cell correlation of SST IN activity was not significantly altered.

### Redistributed SST IN network activity correlates with structural classes of complex action sequences

As mice generated various types of SEQ structures even within a single session, we hypothesized that the amplitude and profile modulation (e.g., $\Delta F/F_0$ of GCaMP6f) in the action-locked SST network activity in M1 L5/6 may reflect structural changes of action sequences in each trial. To examine this, we first performed a correlation analysis between SEQ efficiency and the corresponding amplitude of calcium response for all SEQs within the 4Hz-TC-FR4 training session. We observed a positive correlation between these parameters (Fig. 4a), suggesting that larger calcium transients underlie more efficient SEQs.

To directly compare calcium transients between SEQs with different kinematic structures, we performed unsupervised hierarchical clustering of SEQs within a 4Hz-FR4 training session (Fig. 4b). The analysis revealed three distinct SEQ sub-classes; Class1 (C1), which are the most represented (46.0% of all SEQs) at this training stage, consisted of SEQs with an action-tight lever-press distribution and rapid lever-press rates (Fig. 4c). Class 2 (C2, 29.5%) contained SEQs with a less confined lever-press distribution and slower lever-press frequency, while Class 3 (C3, 24.5%) comprised SEQs with numerous lever-press and slower lever-press rates (Fig. 4c). Thus, we reorganized the fluorescence profiles and behavioral data according to this classification. C3 SEQs showed a higher number of presses per SEQ relative to C1 and C2 (Fig. 4d). In contrast, C1 showed the highest frequency within a SEQ (Fig. 4e). Thus, action sequences in C1, C2, and C3 were also separately clustered on the SEQ efficiency map (Fig. 4f). When we compared fluorescent transients of SST INs across different SEQ classes within the same training session, we observed that the most efficient C1 SEQs had higher peak amplitudes compared to Class 2 and 3 SEQs (Fig. 4g).

To characterize the calcium transients underlying the most efficient SEQs that led to reward retrieval, we further defined a subset of C1 SEQs as "complete SEQ" when they contained fast/confined (>4 Hz) consecutive lever presses and were followed by reward retrieval (Fig. 4h,i). In contrast, a subset of C3 SEQs was defined as "incomplete SEQ" when mice executed slow and elongated lever-press sequences that failed to meet the temporal requirement for consecutive four or more lever presses. In these cases, no reward was dispensed. (Fig. 4j). Less than half of C3 SEQs fall in the "incomplete" category (Fig. 4h). Incomplete SEQs contained a higher lever-press number (Fig. 4k) and

had a lower lever-press frequency (Fig. 4l) compared to complete SEQs. As a result, complete and incomplete SEQs have high and low SEQ efficiency coefficients, respectively (Fig. 4m). Importantly, distinctive neural responses are associated with the execution of these two SEQ subclasses. Calcium transients during complete SEQs had significantly larger peak amplitude than those associated with incomplete SEQs and persisted for a few seconds after action initiation (Fig. 4n,o). Notably, plotting the relationship of the amplitude of SST IN calcium responses and SEQ efficiency segregated the complete and incomplete SEQs (Fig. 4p). Additionally, C3 SEQs could also be categorized into rewarded and non-rewarded SEQs (Supplementary Fig. 4a-c). The rewarded SEQs exhibited increased calcium transients (Supplementary Fig. 4d). Altogether, different kinematics and structural classes of SEQs were distinctively encoded in SST IN activity. Specifically, SST IN activity was higher during efficient SEQs that were appropriately structured and led to successful reward retrieval.

### Specific SST IN activity reflects trial-by-trial structural changes in complex action sequences

To better understand how trial-by-trial action components in TC-FR4 SEQs relate to SST IN activity at a single-cell level, we broke down the population average of SST IN activity (Fig. 3 and Fig. 4) in M1 L5/6 into single-cell calcium responses (Fig. 5a,b). The population average of calcium events exhibited a bimodal distribution (Supplementary Data Fig. 5a), suggesting the presence of two distinct modes of calcium activity. We examined whether these activity modes were represented within individual cells or by mutually exclusive groups of cells. We first identified a primary pattern of SST IN activity that consisted of preferential calcium responses just prior to and during SEQ initiation (Fig. 5a, c). The SST INs showing this activity pattern, which we termed "instant-onset SST," were characterized by a dominant peak in their averaged calcium response occurring within 0.5 s of SEQ initiation. Unexpectedly, among the remaining SST INs that did not respond preferentially at SEQ initiation, we found a distinct set of neurons exhibiting dominant calcium response peaks between 0.5 and 5 s after SEQ initiation (Fig. 5b, d). We defined this group as "delayed-onset SST". Based on this classification, the two SST IN activity patterns appear to define mutually exclusive neuronal subpopulations that can be distinguished by the timing of peak calcium responses relative to SEQ initiation (Fig. 5e). Also, during extended TC-FR4 SEQ training on a 4 Hz schedule, as well as when the schedule was updated to 8 Hz (Fig. 5f), these two distinct SST IN activity patterns were consistently observed across multiple sessions (Supplementary Fig. 5b-f). Notably, the instant calcium response, time-locked to the onset of action sequences during the 8 Hz training schedule, was reduced and showed less variability compared to the 2 Hz and 4 Hz schedules as training progressed (Supplementary Fig. 5g,h). The SEQ structures and calcium profile during performance on the 1 Hz schedule were similar to those in NTC-FR4 training, with no significant differences observed. Conversely, the delayed calcium response became more pronounced during performance on both the 4 Hz and 8 Hz schedules (Supplementary Fig. 5i,j) compared to the 1 Hz schedule, indicating an increase in calcium activity associated with the more complex structure of action sequences.

While the "instant-onset SST" (Fig. 5c, Supplementary Fig. 5c, e) exhibited time-locked calcium responses to the SEQ initiation, we examined if the calcium response of "delayed-onset SST" (Fig. 5d, Supplementary Fig. 5d, f) directly correlated with structural parameters of action sequences. We categorized SEQs into two distinct groups based on the distribution of SEQ durations (Fig. 5g): short SEQs (lower 30% in the SEQ duration distribution, Fig. 5h) and long SEQs (upper 30%, Fig. 5i). We then compared the corresponding calcium peaks. In both the 4 Hz and 8 Hz schedules, we found that the average calcium responses of "delayed-onset SST" were further delayed when mice performed long SEQs compared to short SEQs (Fig. 5j).

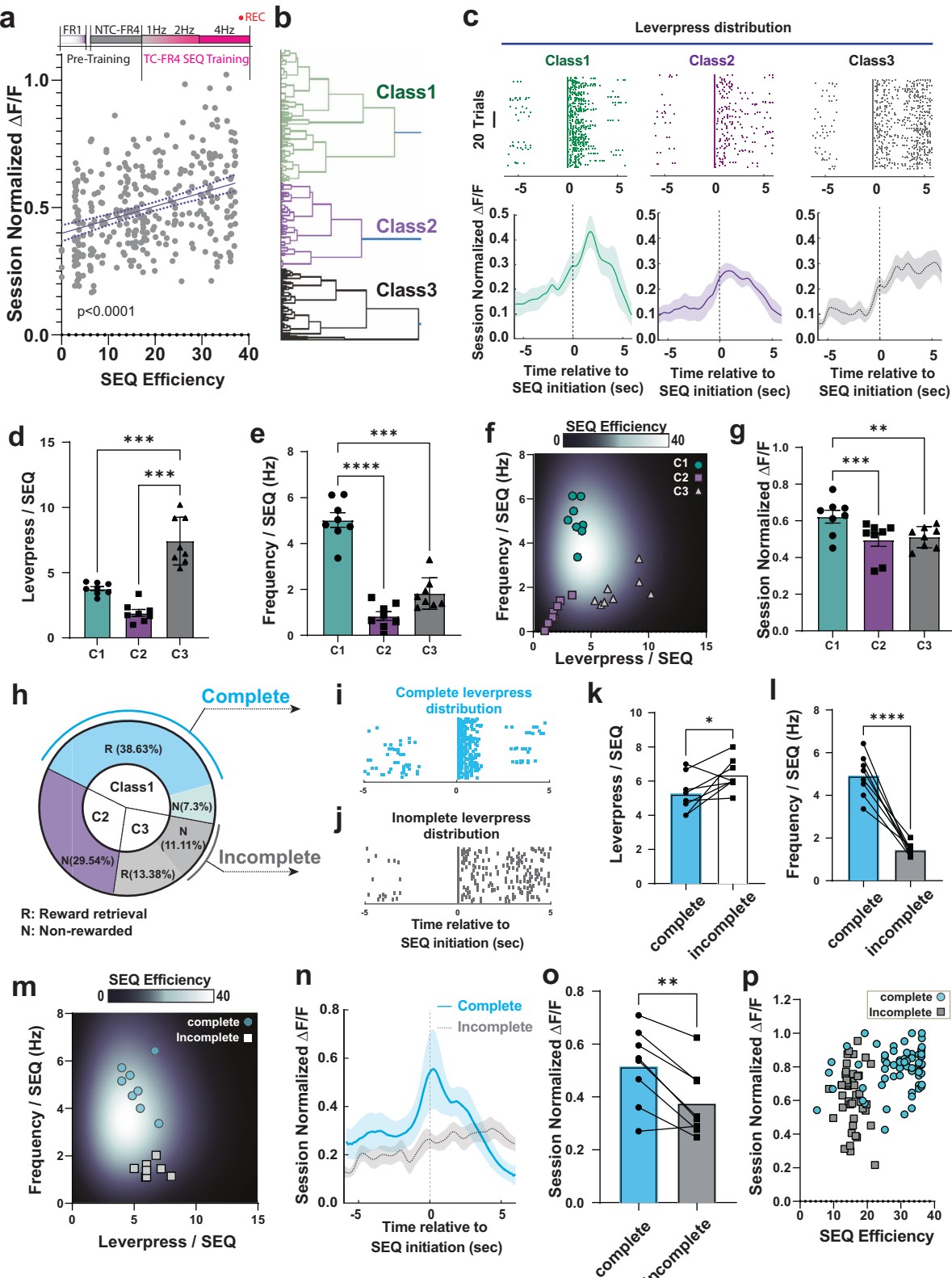

Next, we investigated whether this SEQ duration-dependent calcium peak modulation occurs at the single-cell level within a session. We selected "delayed-onset SSTs" that generated pronounced calcium peaks in both short and long SEQs within 0.5 and 5 s of SEQ initiation and compared the locations of their calcium peaks. We found that the calcium peak locations of each SST INs were significantly delayed during long SEQs compared to short SEQs within a training session for both 4 Hz and 8 Hz training schedules (Fig. 5k,l). This suggests that the individual SST INs are actively modulated in relation to ongoing action sequences. Finally, when this analysis was extended to general SEQ trials across multiple sessions without specific SEQ categorization, we found a linear relationship between SEQ duration and the calcium peak response in "delayed-onset SST" INs (Fig. 5m,n). These results demonstrate that the specific calcium

**Fig. 4 | Redistributed SST population activity correlates with structure of action sequences. a** Relationship between SEQ efficiency and peak ΔF/F$_0$ of SST interneurons, normalized within a session and time-locked to the sequence initiation, each dot denotes sequence trials. (Linear regression for total 330 SEQs; $n = 8$ mice; goodness of fit $R^2 = 0.1331$; slope deviation from zero $p < 0.0001$. Dotted lines indicate 95% confidence bands). **b** Unsupervised hierarchical clustering reveals three major classes (C1, C2, and C3) of lever-press sequences based on kinematic parameters. **c** Example lever-press distribution of classified sequence classes; (Class 1: fast and confined lever-press, $n = 62$ SEQ trials), (Class 2: slow and temporally confined lever-press, $n = 37$ SEQ trials), and (Class 3: slow and temporally elongated lever-press, $n = 127$ SEQ trials) with corresponding fluorescence average of SST population activity ΔF/F$_0$ (%, F$_0$ is spontaneous baseline). **d** Lever-press number per sequence with respect to Class 1, 2, and 3. (one-way ANOVA repeated measure, $n = 8$ mice, ***$ < 0.0001$, Sidak multiple comparison test, C1 vs C3 ***$p = 0.0005$, C2 vs C3 ***$p = 0.0002$. **e** Lever-press frequency (Hz) per sequence with respect to Class 1,2, and 3. (one-way ANOVA repeated measure, $n = 8$ mice, ****$p < 0.0001$, Sidak multiple comparison test, C1 vs C2 ****$p < 0.0001$, C1 vs C3 ***$p = 0.0001$). **f** Lever-press frequency (Hz) and lever-press number per sequence of C1, C2 and C3 are projected onto the sequence efficiency coefficient map, each dot denotes

average coefficient of an animal ($n = 8$ mice). **g** Maximum ΔF/F calcium responses of C1, C2, and C3, normalized within a session measurement and time-locked to the sequence initiation (One-way ANOVA repeated measure, $n = 8$ mice, ***$p = 0.0003$, Sidak multiple comparison test, C1 vs C2 ***$p = 0.0002$, C1 vs C3 **$p = 0.035$). **h** Definition of complete (rewarded and temporally confined in C1) and incomplete (non-rewarded and temporally elongated in C3) sequences. **i** Example lever-press distribution of complete sequences and (**j**) incomplete sequences. **k** Lever-press number per sequence in complete and incomplete sequences (two tailed paired t-test, $n = 8$ mice, *$p = 0.0420$). **l** Lever-press frequency (Hz) per sequence in complete and incomplete sequences (two-tailed paired t-test, $n = 8$ mice, ****$p < 0.0001$. **m** Complete and incomplete sequences projected onto a sequence efficiency coefficient map. **n** Fluorescence traces of SST population activity corresponding to complete and incomplete sequences, time-locked to the sequence initiation. **o** Maximum ΔF/F calcium responses (between 0-5 sec) of complete and incomplete sequences, time relative to the sequence initiation (two-tailed paired t-test, $n = 8$ mice, **$p = 0.0037$). **p** Clustered distribution of complete and incomplete sequences in relation to SEQ efficiency coefficient and corresponding maximum ΔF/F normalized within a session (each dot denotes sequence trials). In all plots, the shaded area and error bars denote SEM.

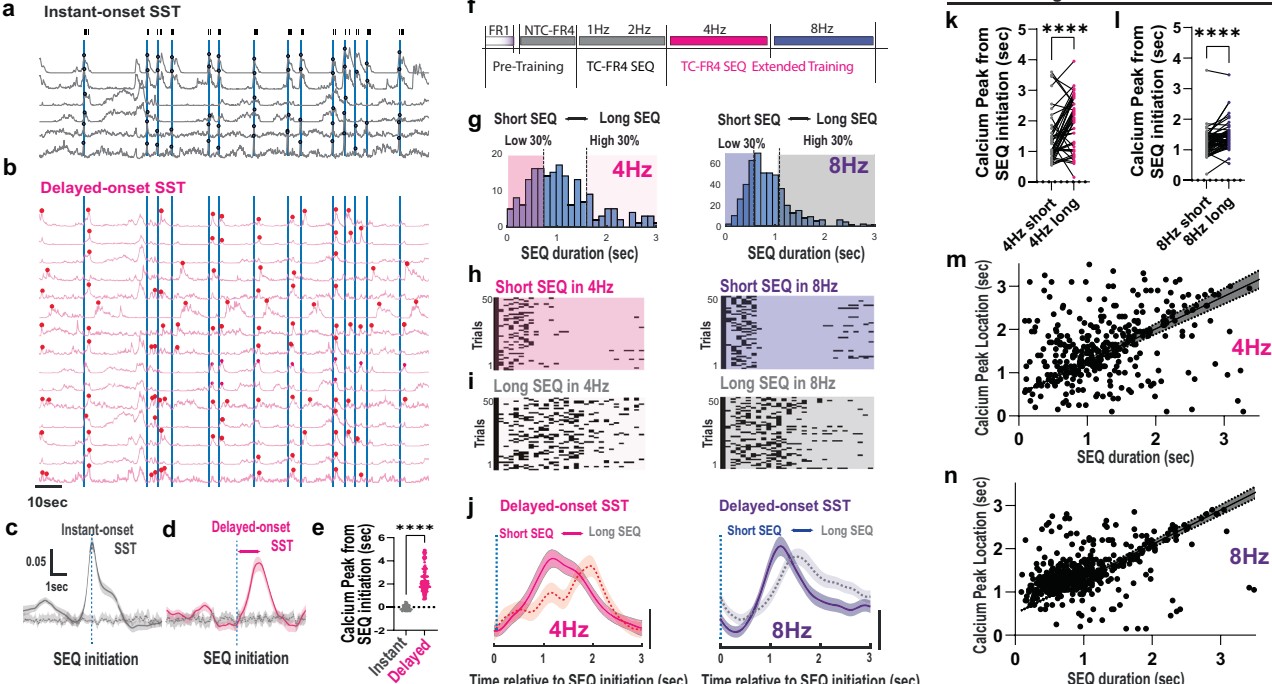

**Fig. 5 | Two distinct activity patterns of SST-INs in M1 L5/6 encode the initiation and trial-by-trial structural modulation of complex action sequences.**
**a** Representative calcium traces (session normalized ΔF/F$_0$) for "instant-onset SST" in the 4 Hz training schedule. Each dot represents a detected calcium transient peak within ±0.5 s windows from SEQ initiation (vertical blue bars). Black bars at the top indicate individual lever-press. **b** Representative calcium traces (session normalized ΔF/F$_0$) for "delayed-onset SST". Unlike "instant-onset SST" cells, these calcium transient peaks are detected within 0.5 to 5 s after SEQ initiation. **c, d** Average calcium responses for "instant-onset SST" (solid gray line, $n = 62$ cells, 3 mice) and "delayed-onset SST" (solid magenta line, $n = 163$ cells, 3 mice) in the 4 Hz training schedule. Dotted gray lines indicate baselines obtained from shuffled data. Dotted vertical blue lines mark SEQ initiation. Error bars denote SEM. **e** Significant difference in calcium peak locations from SEQ initiation between "instant-onset SST" (gray dots, $n = 62$ cells) and "delayed-onset SST" (magenta dots, $n = 163$ cells, Two-tailed Mann–Whitney U test, $p < 0.0001$). **f** Extended TC-FR4 SEQ training with 4 Hz and 8 Hz schedules. **g** SEQ duration distribution during the 4 Hz (left) and 8 Hz (right) training schedules. Short/Long SEQs are defined as the lowest 30% and highest 30% of the distribution, respectively. **h** Example of lever-press patterns (50 randomly selected trials in the last 3 days) for short SEQs during 4 Hz (left) and 8 Hz

(right) schedules. **i** Example of lever-press patterns (50 randomly selected trials in the last 3 days) for long SEQs during 4 Hz (left) and 8 Hz (right) schedules. **j** Average calcium response of "delayed-onset SST" for short SEQs (solid lines) and long SEQs (dotted lines) during the last 3days of 4 Hz(left; two-way repeated measure ANOVA: time × SEQ, **$p = 0.0064$; time, ****$p < 0.0001$; SEQ $p = 0.278$) and 8 Hz(right; two-way repeated measure ANOVA: time × SEQ, ****$p < 0.0001$, time,*** $p < 0.0001$, SEQ $p = 0.9779$) training schedules. Shaded areas denote SEM. The vertical bar represents a value of 0.1 in the session-normalized ΔF/F$_0$. **k, l** Statistical summary of plot (**j**), showing calcium peak locations of individual SST INs shifted significantly within a training session depending on short and long SEQs during the 4 Hz schedule (two-tailed Wilcoxon paired test, $n = 46$ cells, $p < 0.0001$) and 8 Hz schedule (two-tailed Wilcoxon paired test, $n = 66$ cells, p < 0.0001). **m, n** Linear correlation between calcium peak location of "delayed-onset SST" and SEQ duration during 4 Hz (linear regression for 364 SEQ trials, slope non-zero, $p < 0.0001$; two-tailed Spearman's rank correlation, $r = 0.32$, $p < 0.0001$) and 8 Hz (linear regression for 506 SEQ trials, slope non-zero, $p < 0.0001$; two-tailed Spearman's rank correlation, $r = 0.49$, $p < 0.0001$) training schedules. Each dot denotes a SEQ trial. The shaded area denotes a 95% confidence band.

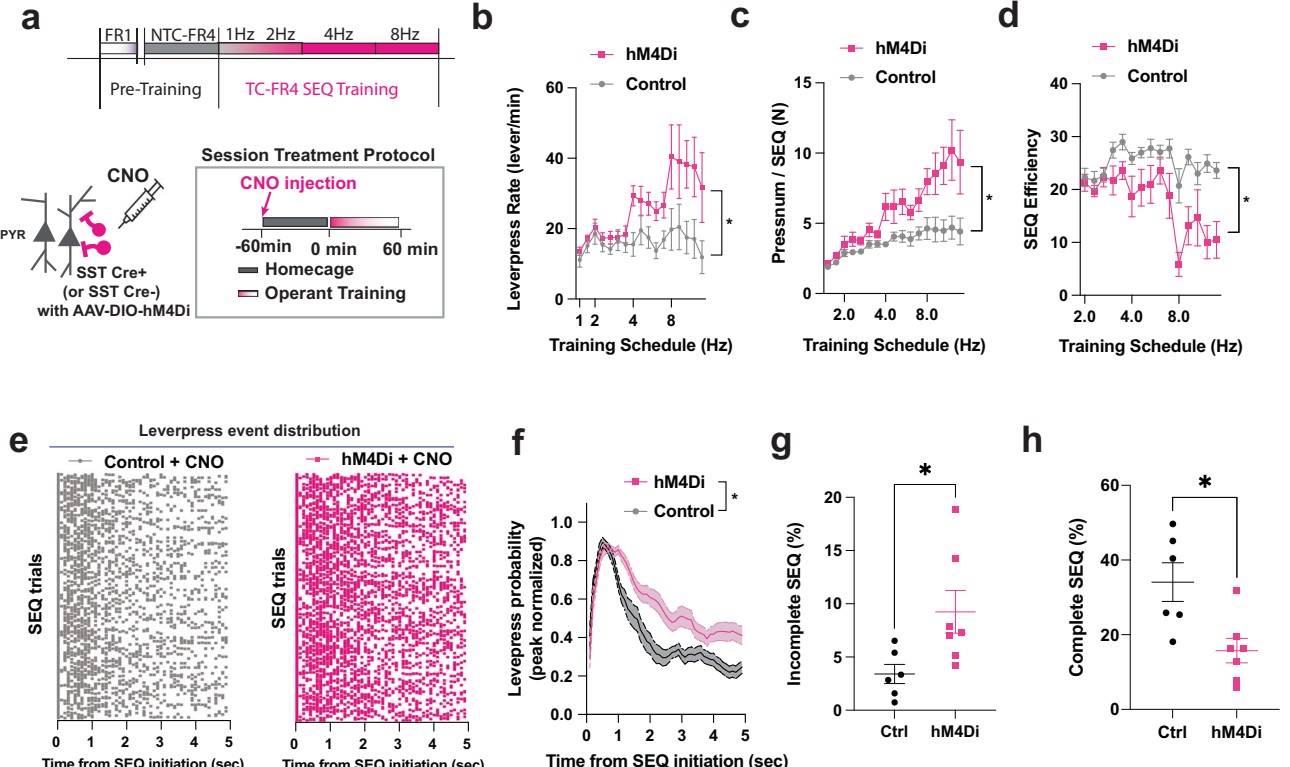

**Fig. 6 | Inhibition of SST interneurons disrupts the kinematics of action sequences. a** Time-constrained FR4 sequence training schedule with hM4Di chemogenetic modulation. **b** Lever-press rate (levers/min) during a session increased in hM4Di-DREADD expressing mice (SST Cre-positive, hM4Di, $n = 7$) and control mice (SST Cre-negative, control, $n = 6$) with CNO injection (5 mg/kg) on both groups (two-way ANOVA repeated measure. schedule ***$p = 0.0005$, CNO treatment *$p = 0.0275$, schedule × CNO ****$p < 0.0001$). **c** Number of lever presses per sequence increased in the hM4Di group, compared to the control group (two-way ANOVA repeated measure, hM4Di = 7 mice, control = 6 mice, schedule *$p = 0.0461$, CNO treatment *$p = 0.03553$, schedule x CNO *$p = 0.0165$). **d** Sequence efficiency coefficient (SEC) decreased in the hM4Di group compared to the control group (two-way ANOVA repeated measure, hM4Di = 7 mice, control = 6 mice, schedule ****$p < 0.0001$, CNO treatment **$p = 0.0099$, schedule x CNO ***$p = 0.0003$). **e** Representative lever-press event distribution in the control group (left, gray) and hM4Di group (right, magenta) during TC-FR4 8 Hz training. Each dot denotes an individual lever-press. Total 500 sequence (SEQ) trials were randomly selected from each group (control $n = 6$ mice, hM4Di $n = 7$ mice). **f** Peak-normalized leverpress distribution probability. Time 0 is sequence initiation. The very first leverpress (SEQ initiation lever-press) was not included in the probability distribution. The hM4Di group shows elongated leverpress probability (two-way ANOVA, hM4Di = 7 mice, control = 6 mice, Time ****$p < 0.0001$, Group ***$p = 0.0001$, Time × Group ****$p < 0.0001$). **g** Proportion (%) of incomplete sequence within in the total unrewarded trials increased in the hM4Di group compared to the control group (two-tailed unpaired t-test hM4Di = 7 mice, control = 6 mice, *$p = 0.0297$). Each dot denotes the average proportion of incomplete sequence per animal. **h** Proportion (%) of complete sequence in the total rewarded SEQ trials decreased in the hM4Di group (two-tailed unpaired t-test hM4Di = 7 mice, control = 6 mice, *$p = 0.0102$). In all plots, the shaded area and error bars denote SEM.

responses of SST IN are linked to trial-by-trial structural changes in action sequences.

## Negative modulation of SST IN function impairs action sequence kinematics

Given the close association between SST INs activity and the structural modulation of action sequences, we hypothesized that inhibition of SST INs would interfere with efficient lever pressing during motor program re-organization. We tested this idea using chemogenetic modulation with the Gi-coupled hM4Di Designer Receptor Exclusively Activated by Designed Drug (DREADD) designed to interfere with SST IN activity. In hM4Di expressing M1 interneurons (Supplementary Fig. 6a, b), we used ex vivo brain slice recording to measure resting membrane potential and observed that hM4Di expression in SST-Cre mice induced neuronal hyperpolarization upon CNO bath application (Supplementary Fig. 6c). Next, we analyzed lever-pressing behavior in control (SST Cre-negative) and hM4Di-DREADD expressing mice (SST Cre-positive) during the TC-FR4 training schedule (Fig. 6a). Intraperitoneal injection of Clozapine-N-oxide (CNO, 5 mg/kg) increased the average lever-press rate (Fig. 6b) in the SST Cre-positive, but not Cre-

negative mice, particularly during the most time-constrained training schedules (i.e., 4 and 8 Hz). We found that this increase was due to increased lever-press numbers within a SEQ in hM4Di-expressing mice compared to the control group across training days (Fig. 6c). This effect also reduced SEQ efficiency in hM4Di-expressing mice compared to control animals (Fig. 6d). These data suggest that, despite an overall increase in lever-press rate, chemogenetic-mediated hyperpolarization of M1 SST interneurons deteriorates sequence organization. To further investigate this point, we analyzed lever-press distribution within each SEQ. While control mice injected with CNO displayed a temporally confined lever-press distribution around sequence initiation, hM4Di-CNO inhibition led to prolonged lever pressing (Fig. 6e). In fact, the normalized lever-press probability distribution after sequence initiation was elongated in hM4Di-expressing animals compared to controls (Fig. 6f). Disruption of efficient lever pressing in hM4Di-expressing mice was also highlighted by an increased and decreased proportion of "incomplete" (Fig. 6g) and "complete" sequences (Fig. 6h), respectively. The hM4Di induced negative modulation of SST INs did not alter general locomotion in the operant box, tested in the absence of a lever, and did not affect lever pressing during the slower

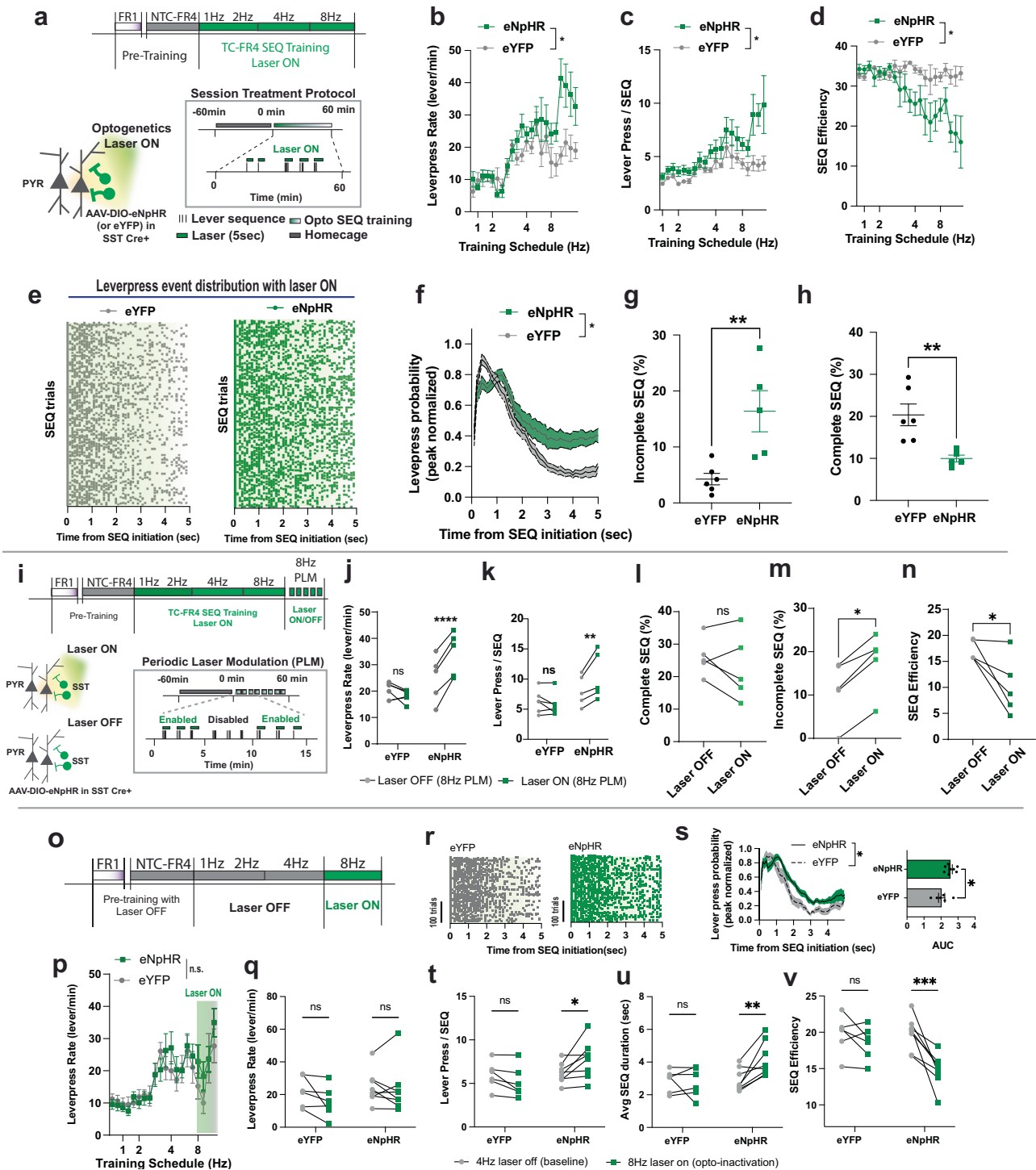

(1–2 Hz) sequences (Fig. 6b–d, Supplementary Fig. 6d–f). These data indicate a direct role SST IN activity in the organization and execution of efficient operant action sequences.

## Action-locked inhibition of SST INs disrupts efficient action sequences

The DREADD approach might interfere with SST interneuron function throughout the duration of each training session, perhaps leading to long-lasting effects on basal SST activity and broader cortical networks on a timescale of minutes-to-hours. To avoid these confounding effects, we aimed to silence action-locked neuronal activity through closed-loop optogenetics specifically during task execution. To do so, we expressed either the inhibitory opsin (eNpHR) or control

fluorescence protein (eYFP) in M1 SST INs using SST-Cre mice. We delivered laser light (wavelength: 532 nm, intensity: ~4 mW) during TC-FR4 training sessions (Fig. 7a). Within these sessions, each action initiation (first lever press) triggered a continuous 5 s-long laser illumination (Fig. 7a) to inhibit sequence-related neuronal activation that persists typically for several seconds as the lever-press sequence continues. Laser illumination in eNpHR-expressing mice led to an increase in average lever-press rate (Fig. 7b) and the number of lever presses within a SEQ (Fig. 7c) compared to eYFP animals, particularly during the most time-constrained training schedules (4 and 8 Hz). Similar to the results obtained with chemogenetic negative modulation, the SEQ efficiency decreased in eNpHR compared to eYFP-expressing mice (Fig. 7d). Additionally, SST optogenetic silencing

**Fig. 7 | Action-specific inhibition of SST interneurons disrupts the execution of an efficient action sequence. a** Time-constrained FR4 sequence training schedule and optogenetic modulation protocol. Each lever press triggers 5 s of laser illumination (532 nm, ~4 mW). **b** Lever-press rate (levers/min) during a session increased in the eNpHR expressing mice ($n = 5$), compared to the eYFP expressing control group ($n = 6$) with the same opto-inhibition protocol. (two-way ANOVA repeated measure. Schedule ****$p < 0.0001$, Treatment *$p = 0.0179$, Schedule x Treatment *$p = 0.0463$). **c** Number of lever presses per sequence increased in the eNpHR mice, compared to the eYFP mice (two-way ANOVA repeated measure. eNpHR = 5 mice, eYFP = 6 mice, Schedule ****$p < 0.0001$, Treatment *$p = 0.0293$, Schedule × Treatment ***$p = 0.0002$). **d** Sequence efficiency coefficient (SEC) decreased in the eNpHR group compared to the eYFP group (two-way ANOVA repeated measure eNpHR = 5 mice, eYFP = 6 mice, Schedule **$p = 0.0024$, Treatment *$p = 0.0114$, Schedule x Treatment ****$p < 0.0001$). **e** Representative lever-press event distribution in the eYFP group (left, gray) and eNpHR group (right, green) during TC-FR4 8 Hz training. Each dot denotes individual lever-press. Laser illumination is on for 5 s. Total 500 sequence (SEQ) trials were randomly selected from each group (eYFP $n = 6$ mice, eNpHR $n = 5$ mice). **f** Peak-normalized lever-press distribution probability in eNpHR and eYFP mice. Time 0 is sequence initiation. The very first lever-press (SEQ initiation lever-press) was not included in the probability distribution curve (two-way ANOVA, eNpHR = 5 mice, eYFP= 6mice, Time ****$p < 0.0001$, Group **$p = 0.0078$, Time × Group ****$p < 0.0001$). **g** Proportion (%) of incomplete sequence within in the total unrewarded trials increased in the eNpHR group compared to the control group (two-tailed unpaired t-test, eNpHR =5 mice, eYFP = 6mice, **$p = 0.0071$). Each dot denotes average proportion of incomplete sequence per animal. **h** Proportion (%) of complete sequence in the total rewarded SEQ trials decreased in the eNpHR group (two-tailed unpaired t-test, eNpHR = 5 mice, eYFP= 6 mice, **$p = 0.0066$). **i** Periodic Laser ON/OFF modulation (PLM) training schedule. **j** eNpHR group exhibited alteration in lever-press rate (levers/min) depending on the periodic laser modulation (5 min ON and 5 min OFF). Each dot represents an animal average (two-way ANOVA repeated measure, eYFP ($n = 6$ mice) vs eNpHR ($n = 5$ mice) effect: *$p = 0.0177$, OFF vs ON effect: **$p = 0.001$, Sidak multiple comparison within a group: OFF vs ON within eYFP $p = 0.1924$, OFF vs ON within eNpHR ****$p < 0.0001$). **k** In the same training, only eNpHR group increased number of lever presses per sequence in the laser ON period than laser OFF period (Two-way ANOVA repeated measure, OFF vs ON effect: *$p = 0.0285$, eYFP ($n = 6$ mice) vs eNpHR ($n = 5$ mice) effect $p = 0.0624$, Sidak's multiple comparison test, laser on/off

effect within eYFP $p = 0.5593$, within eNpHR **$p = 0.0032$). **l** In the same PLM training schedule, proportion (%) of complete sequence was not significantly altered (two-tailed paired t-test, n=5mice, $p = 0.1279$). **m** Within the same eNpHR expressing subject, proportion (%) of incomplete sequence increased during laser ON periods compared to laser OFF periods (two-tailed unpaired $t$ test, $n = 5$ mice,*$p = 0.0105$). Each dot denotes average proportion (%) of incomplete sequence per animal. **n** Sequence efficiency coefficient (SEC) decreased during laser ON periods compared to laser OFF periods (two-tailed Wilcoxon test, $n = 5$ mice, *$p = 0.0312$). In all plots, shaded area and error bars denote SEM. **o** TC-FR4 sequence training schedule and optogenetic modulation protocol during the 8 Hz task without prior treatment. **p** Average lever-press rate (lever presses/min) during late-phase opto-inhibition showed no significant difference between eYFP ($n = 6$ mice) and eNpHR ($n = 8$ mice) groups (two-way repeated-measures ANOVA: Group, $p = 0.68$; Training Schedule, ***$p < 0.0001$; Group × Schedule, $p = 0.37$). Error bars denote SEM. **q** Lever-press rate (lever presses/min) on the first day of the TC-FR4 8 Hz schedule was not significantly different between eYFP and eNpHR groups relative to baseline performance at the TC-FR4 4 Hz schedule (two-way repeated-measures ANOVA: Group, $p = 0.3928$; Opto-inactivation, $p = 0.1079$; Group × Opto-inactivation, $p = 0.2758$, eYFP 6mice, eNpHR 8mice). **r** Representative lever-press event distribution in the eYFP group (left, gray, 6mice) and eNpHR group (right, green, 8 mice) on the first day of TC-FR4 8 Hz training in (i). **s** Lever-press distribution (two-way ANOVA: Time, ****$p < 0.0001$; Group,*$p = 0.0060$, Time x Group $p = 0.2889$) and corresponding area under the curve (AUC, two-tailed unpaired $t$ test: *$p = 0.0378$) are significantly different between eYFP ($n = 6$ mice) and eNpHR ($n = 8$ mice). Error bars denote SEM. **t** Lever-presses per sequence (SEQ) showed a significant Group × Schedule interaction and a main effect of Schedule (two-way repeated-measures ANOVA: Group, $p = 0.1194$; Opto-inactivation, $p = 0.3188$; Group × Opto-inactivation, **$p = 0.0078$; Sidak's multiple comparisons test: eYFP ($n = 6$), $p = 0.3285$; eNpHR ($n = 8$), *$p = 0.0144$). **u** Average SEQ duration (s) showed a significant Group × Schedule interaction and main effects of both factors (two-way repeated-measures ANOVA: Group, $p = 0.095$; Opto-inactivation, *$p = 0.0314$; Group × Opto-inactivation, *$p = 0.0174$; Sidak's multiple comparisons test: eYFP ($n = 6$), $p = 0.9731$; eNpHR ($n = 8$), **$p = 0.0038$). **v** SEQ efficiency also showed significant main effects and interaction (two-way repeated-measures ANOVA: Group, $p = 0.1005$; Opto-inactivation, **$p = 0.001$; Group × Opto-inactivation, *$p = 0.0206$; Sidak's multiple comparisons test: eYFP ($n = 6$), $p = 0.5127$; eNpHR ($n = 8$), ***$p = 0.0004$).

---

elongated the lever-press probability distribution at action initiation (Fig. 7e, f), increased the proportions of incomplete sequences and decreased the percentage of complete sequences (Fig. 7g, h) compared to controls.

To determine whether the pronounced behavioral differences observed during the TC-FR4 8 Hz schedule (Fig. 7b-d, f-g) required ongoing optogenetic inactivation during task performance, we introduced a Periodic Laser ON/OFF Modulation (PLM) protocol after animals completed TC-FR4 training on the 1, 2, 4, and 8 Hz schedules (Fig. 7i) and examined how alternating laser ON/OFF conditions influenced sequence organization within a session. When we analyzed the action sequences generated during laser-OFF periods, the eNpHR group demonstrated significantly lower lever-press rate (Fig. 7j) and fewer lever presses per sequence (Fig. 7k) compared with laser-ON periods. This modulation was not observed in the eYFP group (Fig. 7j,k). Moreover, the elongated lever-press probability distribution was significantly altered by the Laser-ON/OFF control only in eNpHR group (Supplementary Fig. 7a-c). While the proportion of complete SEQs remained unaffected within the PLM training schedule (Fig. 7l), the proportion of incomplete SEQs within the eNpHR group during the laser-OFF periods was significantly lower than laser-ON periods (Fig. 7m). Sequence efficiency was also reduced when the condition switched from laser-OFF to laser-ON (Fig. 7n).

These findings demonstrate that dynamic laser ON/OFF control continued to modulate behavior, indicating that SST-IN activity remains actively engaged in organizing high-frequency action sequences. However, this finding does not necessarily indicate that the change in TC-FR4 8 Hz task was only due to immediate optogenetic inactivation as it is possible that cumulative effects induced by long-

term optogenetic suppression may also contribute to these behavioral changes.

To clarify the contribution of immediate optogenetic inactivation, we next investigated whether a single-session, immediate SST-IN inactivation could reproduce the behavioral differences observed after long-term optogenetic suppression. We applied a short-term optogenetic inactivation protocol applied exclusively during the 8 Hz schedule, without any prior treatment history (Fig. 7o). On the first day of TC-FR4 8 Hz training, SST-IN activity in eNpHR-expressing mice was silenced using lever-locked closed loop laser activation, and behavioral metrics were examined. Lever presses per minute did not differ significantly between eYFP and eNpHR mice (Fig. 7p,q). However, both the lever-press distribution probabilities and their corresponding area under the curve (AUC) showed significant differences between eYFP- and eNpHR-expressing mice (Fig. 7r,s), indicating that even a single session of SST-IN inactivation was sufficient to alter sequence structure.

To determine how these changes in lever-press distribution probabilities were reflected in sequence parameters, we compared each mouse's performance during the 8 Hz schedule against their own baseline performance at 4 Hz schedule (Fig. 7t-v). Both eYFP controls and eNpHR mice exhibited a significant Group × Laser interaction for sequence parameters, including lever presses per sequence (Fig. 7t; two-way ANOVA, Group × Laser interaction: $p = 0.0078$), sequence duration (Fig. 7u; $p = 0.0174$), and sequence efficiency (Fig. 7v; $p = 0.0206$). Post hoc tests confirmed that the laser-training effect was specific to the eNpHR group. However, the main effects of group (eYFP vs. eNpHR) were not significant for lever presses per sequence ($p = 0.1194$), sequence duration ($p = 0.0950$), or sequence efficiency

($p = 0.1005$). We further introduced the PLM protocol and found similar alteration in lever-press probability distributions (Supplementary Fig. 7d,e) and significant laser-dependent modulation in sequence parameters (Supplementary Fig. 7f-h). Thus, a single-session, immediate optogenetic inactivation produced changes in sequence structure (Fig. 7s–v), but is not sufficient to alter overall lever-press rate (Fig. 7p, q).

In summary, both immediate and long-term SST-IN inactivation disrupted the temporal organization of lever-press sequences (Fig. 7f, s). However, significant alteration of lever-press rate by immediate inactivation was observed only after long-term optogenetic suppression (Fig. 7b, j). This suggests that cumulative suppression of SST-INs may create a condition permissive for overly exuberant lever pressing. In our TC-FR4 training, this effect was only observed when SST-IN activity was inactivated during high-frequency sequence execution, even following long-term SST-IN inactivation (Fig. 7b).

## Discussion

We investigated single-cell calcium activity in the M1 of freely moving mice as they progressively reorganized reinforced motor programs from a single lever press to a time-constrained sequence of four lever presses. We observed that action-related calcium responses in M1 SST INs diminished as the consolidation of the single lever-press task progressed (Figs. 1, 2 and Supplementary Fig. 2), but did not decrease and temporally redistributed as the timing structure of rapid action sequences was reorganized (Fig. 3). Redistributed M1 SST IN activity encoded the structural information of on-going action sequences, such as the number and frequency of lever presses within a sequence, which correlated with efficient motor program execution (Fig. 4). Specifically, distinct neuronal activity patterns emerged in non-overlapping M1 SST INs, independently encoding sequence initiation and the duration of rapid action sequences (Fig. 5). Chemogenetic inhibitory modulation and action-locked optogenetic silencing of SST INs (even on a trial-by-trial basis) disrupted the timing structure of rapid action sequences and decreased motor efficiency, demonstrating a causal link between SST IN activity and the organization of efficient, task-specific action sequences (Figs. 6, 7 and Supplementary Fig. 7).

### Dynamic neural representation in SST INs across varying task complexities

A key challenge in studying the cortical motor circuit is that neural representation in M1 is dynamic, varying with the learning phase and degree of behavioral complexity[29]. Several in vivo studies have shown that neural circuits in M1 exhibit different patterns of activity depending on whether a motor program is in the process of being learned or has already become a well-established, stereotyped behavior. Previous evidence from our laboratory indicates that increased calcium activity of both associative and motor cortex striatal projection neurons is observed during early training. However, as learning progresses, associative inputs to dorsomedial striatum show stronger disengagement compared to the M1-dorsolateral striatal pathway[11]. These data align with the idea that neural activity in M1 persistently represents action performance and kinematics even after skill learning, although the M1 contribution may undergo some refinement[4,12,13]. On the contrary, the cortical disengagement model proposes that M1 activity becomes less critical[2,15,30] or decorrelated from action execution upon consolidation of a given motor program[16]. This model implies that cortical output is less important for movement control as proficiency is gained. These perspectives on M1 are not mutually exclusive, but instead suggest that, while simple, repetitive motor tasks or stereotyped movements may require minimal cortical regulation, more flexible and complex motor tasks likely demand continuous cortical involvement.

We tested this hypothesis by monitoring neural activity of cortical microcircuits, particularly SST INs, at the single cell level, to determine if their action-related neural activity changes during the maintenance of a simple motor task or when more complex motor training schedules are imposed. Unlike previous studies that focused on examining the plasticity of neighboring PYR neurons during isolated forelimb movement or treadmill running tasks in head-fixed settings[24–26], our study directly monitored the calcium activity of M1 SST INs in freely moving mice and assessed their action-locked calcium responses across various training schedules with differing the structural complexity of action sequences.

In the simple FR1 task schedule, our finding of diminished SST IN activity following extended training (Fig. 2) is consistent with the cortical disengagement model[9,16,30]. For instance, previous studies have shown a decline in the correlation of neural activity in M1 PYR neurons during repetitive motor training[16]. Similarly, we observed decreased M1 SST IN activity and correlation during the execution of the well-learned FR1 task. As cell-specific inhibition during the FR1 task produced no measurable behavioral effect, these changes in SST IN activity may reflect the broader disengagement feature of M1, rather than region-specific motor output. Cortical dynamics often display widespread activation; however, only specific subsets functionally contribute to the behavior. In the more complex, time-constrained action sequences (e.g., TC-FR4), we observed that a strong correlation was maintained between SST INs activation and action sequence execution as task demands increased and proficiency decreased (Fig. 3). If individual lever-presses or discrete action components evoke a calcium transient in SST INs, the elevated calcium signal observed during time-constrained action sequences may reflect the accumulation of calcium across multiple presses or concatenated actions.

As mentioned in the Results, implantation of the GRIN lens can cause damage to superficial cortical layers, which may introduce a potential confound when measuring L5/6 SST-IN activity. This is a risk for all invasive procedures that require hardware implants in brain. At present, it remains challenging to image deep-layer M1 SST-INs in freely moving mice without any cortical damage, and the consequent potential circuit effects should be considered when interpreting the results. Nevertheless, it is notable that neurons can often survive initial damage from local implantation, as indicated by the many single-unit recordings performed with indwelling electrodes and electrode arrays[3,4,10,17]. In addition, a previous study[13] reported that action-locked M1 activity of superficial and deep layer cortical pyramidal neurons assessed with non-invasive 2-photon imaging in a head-fixed situation showed layer-dependent neural activity differences similar to what we observed in the present study (Supplementary Fig. 1e, f). Thus, the layer-specific activity of cortical neurons may be a general feature of action coding. Our observation that L5/6 SST-INs show distinct activity patterns with different operant training schedules (Fig. 2 and Fig. 3) and tasks (Fig. 4 and Fig. 5) indicate that these neurons retain the ability to change their activity patterns even when measured with this invasive imaging procedure. In future work, head-mounted two-photon imaging or prism-based cross-sectional imaging may help address concerns regarding superficial layer damage in our characterization.

This SST INs activity was modulated based on the structure of on-going action sequences (Fig. 4). Prior research has monitored PYR neurons during manipulations of SST INs in motor skill learning tasks and found that SST INs regulate task-dependent, branch-specific $Ca^{2+}$ activation patterns in PYR neurons[26] and influence their dendritic spine plasticity[24]. However, our study aimed to identify the specific motor information directly encoded by SST INs during adaptive motor tasks. We found distinct neural patterns encoding action sequence initiation and duration in non-overlapping SST IN populations (Fig. 5), with a strong correlation to sequence structure under the TC-FR4 at 4 Hz and

8 Hz schedules. Effects of SST IN inhibition were also specifically pronounced during these training schedules (Fig. 6). Moreover, optogenetic silencing of action-locked SST IN activity disrupted both sequence structure and motor execution on a trial-by-trial basis (Fig. 7l,m,n). These results demonstrate that M1 SST INs exhibit distinct activity patterns based on task complexity. While simple, repetitive tasks like FR1 show a significant reduction in SST IN activity over time (Supplementary Fig. 2), more complex and time-constrained tasks, such as TC-FR4, reveal strong correlations between SST IN activation and the regulation of action sequence structure.

## Task-specific inhibition effects of SST INs

We examined the functional contribution of M1 SST INs using cell-specific inhibition approaches across varying motor training schedules. We recognize that previous studies reported that optogenetic inhibition of SST INs disrupts the learning of stereotyped forelimb movements in head-fixed mice during lever-maneuver tasks[29], with significant learning impairments observed from the early stages of training. Given previous findings, it was unexpected that we did not observe significant motor impairments in freely moving mice during FR1 training, or during NTC-FR4 and TC-FR4 training at 1 Hz and 2 Hz, even with the application of chemogenetic inhibition and closed-loop optogenetic silencing of SST INs (Fig. 6 and Fig. 7). However, both chemogenetic and optogenetic inhibition eventually disrupted the structure and efficiency of action sequences specifically during 4 Hz and 8 Hz TC-FR4 lever-press training (Fig. 6d and Fig. 7d). This difference could be attributed to both the cumulative effects of long-term suppression and the immediate optogenetic inactivation during ongoing sequence production. Even when we examined how immediate optogenetic inactivation disrupts sequence organization without any prior opto-inactivation history, significant alterations in sequence parameters were observed only during the 8 Hz schedule (Fig. 7o-v and Supplementary Fig. 7j-k), whereas these differences were not detectable during the 1 Hz schedule (Supplementary Fig. 7j-k), suggesting that involvement of SST-IN activity in behavior differs depending on the task.

While optogenetic inhibition was somewhat layer-specific—given that the placement of the optical fiber likely targeted SST INs in L5/6—the chemogenetic approach was not layer-specific. Nonetheless, both manipulations resulted in consistent behavioral disruptions. This disruption coincided with the emergence of temporally confined lever-pressing patterns (Fig. 6 e,f and Fig. 7 e,f), leading to an increase in incomplete lever press sequences and a decrease in complete sequences (Fig. 6g,h and Fig. 7g, h). Thus, there appears to be an inconsistency: while SST IN inhibition specifically affected complex motor sequences in our study (e.g., TC-FR4 4 Hz and 8 Hz), previous studies with head-fixed mice performing stereotyped forelimb tasks observed motor skill impairment from the beginning.

To understand this discrepancy, it is essential to consider the differences in motor assessment metrics and training schedules between previous studies and our own. Previous studies using head-fixed mice implemented a constant training schedule to establish stereotyped forelimb movements, focusing on forelimb trajectory or motor parameters (e.g., speed or acceleration) as indicators of motor learning[12,14,16,22,24]. In contrast, by analyzing lever press patterns on a trial-by-trial basis in our study, we assessed action performance and efficiency as opposed to motor parameters reflecting isolated forelimb movements. Thus, our motor metrics might not fully capture subtle variations in stereotyped forelimb movements (some of which may show inter-individual variability), but they do capture the organized action patterns needed to successfully complete the task regardless of these variations. Future studies incorporating detailed analyses of forelimb or whole-body kinematics would provide a more comprehensive understanding of the underlying motor control mechanisms. In addition, the role of the motor cortex and SST IN activity in isolated

forelimb control in head-fixed tasks may differ fundamentally from their role in our freely moving conditions, as the necessity of cortical activity and its correlation with behavior are highly task-specific[14,29,31]. For example, our behavior training took approximately 200–300 trials (7 days) to establish a stable FR1 task in freely moving mice and around 600 trials (21 days) to measure neural decorrelation between M1 SST IN activity and movement (Fig. 2). These training conditions, which do not rely on specific kinematics for task completion, are substantially faster than the 6000 training trials over 60 sessions required for head-fixed mice to perform stereotyped movements[16]. Our training criteria may be significantly "simpler" than motor tasks in other studies, where motor cortex lesions immediately impaired performance and learning, yet 20,000 to 30,000 training trials were necessary to achieve stereotyped forelimb movement patterns[2,15]. We speculate that the motor tasks involving such stereotyped forelimb tasks may be more "complex" in terms of skill learning, requiring greater cortical involvement.

For example, learning stereotyped forelimb joystick control[12,24,32] or precise reaching tasks[33,34], under head-fixed conditions likely demands not only dexterous movement control but also significant motor adaptation to compensate for unnatural biomechanical constraints. These motor performances are typically sensitive to cortical perturbations, consistent with a view that M1 is crucial for dexterous motor behaviors and for adaptive or corrective movements when stereotyped trajectories need to be altered[32,33,35,36]. In contrast, more naturalistic movements, such as walking, rely less on M1[37,38]. The functional role of M1 likely differs depending on the motor task, particularly whether it involves dexterous movements.

In some experimental designs using freely moving animals, it is possible to decouple the precise kinematics or dexterous movements from task completion. Recent studies demonstrate that M1 is not necessary for performing fully automated, stereotyped lever-press sequences[2] but remains essential for sequences requiring flexibility and external cue guidance[20]. These findings indicate that M1 engagement varies depending on task demands, even when specific dexterity or precise kinematics of movement are not necessarily required. Similarly, our experimental design employs freely moving mice performing various sequential lever-press tasks that are learned within 100–1000 of training trials. Based on previous studies showing that M1 exhibits two distinct engagement dynamics depending on behavioral context[2,20], we hypothesized that local microcircuits within M1, including SST INs, also exhibit task-dependent activity patterns. To test this, we varied the training schedules to progressively increase the demands on motor performance required to complete the action sequences and examined how SST interneurons contributed to shaping their temporal structure.

Using cell-specific calcium imaging, we observed distinct activity patterns of M1 SST-INs between (1) the simpler FR1 task—which allows for greater motor variability and more naturalistic movements to complete the task—and (2) the more demanding TC-FR4 SEQ task, which imposes tighter temporal constraints and requires an unnatural, structured lever-press sequence that is more difficult to learn. This distinction is further supported by our behavioral findings, which show that inhibition of M1 SST-INs selectively impaired performance on the more demanding TC-FR4 SEQ tasks under the 8 Hz schedule, compared to the 1 Hz schedule (Supplementary Fig. 7i-l). Overall, it is possible that M1 SST-IN activity partially reflects broader M1 circuitry, exhibiting task-dependent engagement characteristics. However, it remains an open question whether broader M1 dynamics shape SST IN activity or, conversely, whether task-dependent M1 SST IN activity directly influences broader M1 circuitry during our behavioral paradigms.

Nevertheless, our experimental result underscores the task-specific properties of M1 SST INs activity within our assessment metrics. Notably, when the calcium activation in "delayed-onset SST" became more pronounced (e.g., in TC-FR4 4 Hz and 8 Hz, Supplementary Fig. 5i,j), the effects of chemo- and opto-inhibition also

became more noticeable (Fig. 6 and Fig. 7). Particularly, the abrupt change that coincided precisely with the transition to a more demanding task stage appears to be associated with the task-specific characteristics of SST-INs (Supplementary Fig. 7i–l). However, the effects of long-term suppression are likely superimposed on these results, as immediate optogenetic inactivation alone was not sufficient to induce changes in lever-press rate (Fig. 7b,p).

Previously, task-specific properties of SST INs activity have been shown by examining sequential activity patterns of PYR neurons in forward and backward treadmill running tasks[25]. This suggests that the contribution of SST INs may vary between motor programs that are previously learned or newly acquired. Our study extends this view by demonstrating that the involvement of M1 SST INs in action control varies with the complexity of motor tasks. Despite difference in motor tasks and testing configurations, both previous work and our results support the same idea of task-specific functional contribution of M1 SST INs, consistent with recent predictions regarding cortical IN functions[25,26,39] and task-dependent necessity of cortical regions[29,31].

## Specific motor information encoded in SST IN activity

Identifying the specific motor information encoded by M1 cortical INs is crucial for understanding their role in adaptive motor behaviors. Thalamic inputs to cortical layer 5[40] are critical for processing somatosensory information and generation of movement-related patterns in M1[34]. Additionally, M1 local circuits integrate sensory information through inhibitory networks that involve different subtypes of cortical INs (e.g., PV, SST, VIP)[21,41]. Thus, while neural signals continuously flow into M1 local circuits from other brain regions[42,43], cortical INs may be crucial in processing and integrating sensory information[41] and regulating neighboring pyramidal neurons[44]. A sub-population of PYR neurons in M1 layer 5/6 (L5/6) then convey the processed motor information to the striatum[11] or the spinal cord[27]. Therefore, determining whether the neural activity patterns in cortical microcircuit in M1 resemble those from sensory inputs or motor outputs is crucial in understanding functional role of M1. Our study specifically focused on monitoring M1 SST IN activity to determine what aspects of action sequence-related information are encoded.

One discernible motor-related signal observed in M1 SST INs is the initiation of action sequences. Specifically, this activity pattern emerged in particular SST INs and showed a continuous decrease in calcium response from 2 Hz, 4 Hz to 8 Hz, along with a "sharpening" of activation duration (Supplementary Fig. 5g,h). As sequence training progressed, the activity of M1 SST INs also underwent significant stabilization. Notably, chemogenetic inhibition did not alter sequence initiation, initial learning of TC-FR4 sequence training, or open-field locomotor activity (Supplementary Fig. 6d,e,f). Despite the limitations of our chemogenetic approach and transgenic mice in targeting specific sub-population within SST INs, these findings suggest that M1 SST INs may have a limited role in action initiation or influencing locomotor vigor.

Another activity pattern of SST INs is independently characterized by a delayed calcium response following the onset of action sequences. This response was specifically pronounced in the 4 Hz or 8 Hz training schedules compared to 1 Hz training schedule (Supplementary Fig. 5i,j). These two distinct population dynamics—"instant" and "delayed" onset—were not layer-dependent; rather, both types of activity were observed within L5 /6 SST neurons. We observed that inhibition of SST activity (including inhibition on a trial-by-trial basis) prolonged the temporal distribution of lever press probability (Fig. 6f and Fig. 7f), suggesting that the ongoing activity of SST INs (especially those that code for SEQ duration) might also regulate the termination of action sequences, which is crucial for organizing and reorganizing the temporal structure of motor programs[45]. While delayed-onset SST INs show a dispersed activity pattern when aligned to the initiation of the action sequence (Supplementary Fig. 5a), likely due to the probabilistic distribution in sequence duration (Fig. 5g), their responses become more temporally

aligned when referenced to the final lever press or the subsequent transition to reward port access (Supplementary Fig. 9). Notably, our optogenetic protocol was specifically targeted to silence this activity. Despite some similarities in neural representation, the effects of manipulating M1 SST INs are quite discernable from those observed in striatal circuits that encode action sequence initiation, termination, and modulation[3,10]. It is also interesting that the activity pattern in M1 SST INs (Fig. 5j and Supplementary Fig. 5d,f) resembles the neuronal population response in the motor cortex which exhibits brief but strong oscillatory patterns across species[17].

Our study mainly focuses on neural activity of SST INs in M1 L5/6, which exhibit consistent action-locked calcium responses (Fig. 1k, o, p, q, and Supplementary Fig. 1) compared to those in L2/3 or PYR neurons. However, limitations inherent to fluorescence intensity measurements ($\Delta F/F_0$ %)—such as variability in background signal or contributions from endogenous calcium fluctuations—should be considered when interpreting these results. Although our $\Delta F/F_0$ (%) analysis was intended to minimize cell-to-cell variability in background fluorescence and to exclude non-action-related calcium signals (see Methods), future studies utilizing in vivo electrophysiological recordings will be valuable for further characterizing layer- and cell-specific neural dynamics.

In the cell-specific inhibition experiments (Figs. 6, 7), we assessed the effects of SST IN inhibition on sequence organization by defining sequence (SEQ) efficiency. Although increased lever pressing in the hM4Di group (Fig. 6b) might be expected to enhance reward earning (e.g., higher reward rate), we found no significant difference in reward rate between hM4Di and control groups across all training schedules (two-way repeated measures ANOVA: time effect ****$p < 0.0001$; treatment effect $p = 0.9327$; time × treatment interaction $p = 0.5632$). Thus, despite the increased lever pressing observed in the hM4Di group (Fig. 6c, e, f), these additional presses did not translate into more advantageous action patterns for reward earning. Consequently, SST IN inhibition causes mice to generate unnecessary lever presses unrelated to reward acquisition.

Although our definition of SEQ efficiency does not incorporate reward rate directly, it specifically penalizes unnecessary actions by counting lever presses that exceed the optimal number (Δ = actual presses − optimal presses, for values > 0) within individual action sequences. In this view, lever presses beyond the sequence criterion, or at rates exceeding 4 Hz, are considered unnecessary muscle activity and energy expenditure rather than an optimal strategy for the task. This interpretation aligns with the observation that control mice typically produce ~4–5 lever presses per sequence trial (Fig. 6c or 7c). The sequence efficiency metric is therefore designed to incentivize the optimal sequence generation pattern naturally exhibited by wild-type mice. Although our efficiency function relies on empirical parameters due to the absence of a theoretical behavioral model, this analysis may provide insight into the functional role of SST INs in organizing optimal, energy-efficient action sequences.

Overall, our findings implicate SST INs in M1 as fundamental upstream regulators for the efficient execution of complex action sequences. Beyond their crucial role in facilitating plasticity in PYR neurons during motor learning[24,25], the dynamic activity of M1 SST INs may directly influence the structural regulation of trial-by-trial action sequences. More specifically, it is plausible that the net output of M1 circuitry contributes to the behavioral modulation observed in our experimental results (Figs. 6 and 7), while SST INs serve as modulatory elements within M1 circuits. Completion of TC-FR4 task at 4 Hz and 8 Hz schedules requires substantial motor control. Within this framework, our data suggests that a key function of M1 SST INs may be to "fine-tune" complex motor programs. In addition, identifying the cortical information processing in M1 Layer 5/6 can provide new insights into cortico-striatal circuit function during motor program execution. Future investigations should focus on elucidating how M1 SST INs regulate cortical network output and their impact on the

striatal circuit with respect to reorganization of complex motor programs. Also, the established SST activity potentially reflects some learning-related changes, e.g., delayed onset activity. A comprehensive analysis of these dynamics remains an important direction for future research. These studies will deepen our understanding of the role of SST IN activity and cortical microcircuit mechanisms in sophisticated motor control. In addition, they may have implications for therapeutic approaches to neurological movement disorders, particularly in restoring efficient action execution[45,46].

## Methods

### Animals

All animal protocols were performed in accordance with the guidelines for care and use of laboratory animals of the US National Institutes of Health approved by the US National Institute on Alcohol Abuse and Alcoholism (NIAAA) Animal Care and Use Committee. Mice were housed in the NIAAA animal facility under standard conditions with a 12 h light/dark cycle, controlled ambient temperature and humidity. Viral injections and implant surgery were performed on 3- to 8-month-old male and female mice. Wild-type C57/Bl6J mice (JAX Stock #: 000664) were purchased from the Jackson Laboratory. Homozygous SST-Cre mice (JAX stock #: 013044; Sst tm2.1(cre)Zjh /J) were crossed with the wild-type C57BL6/J mice (JAX Stock #: 000664) in the NIAAA animal facility. The Cre-positive (SST+) heterozygous offspring were used for imaging and behavioral experiments. Depending on experimental design, we also obtained Cre-positive (SST+) and Cre-negative (SST−) offspring as cage mates from the wild-type C57BL6/J mice (JAX Stock #: 000664) and the heterozygous SST Cre-positive (SST+) mice. When the viral expression was not driven by Cre recombinase, C57/Bl6J mice (JAX Stock #: 000664) were purchased and used in the behavioral experiments. Calcium imaging animals were single-housed or housed in pairs (maximum two mice per cage); no differences in behavior or neural activity were observed between housing conditions. Optogenetic and chemogenetic animals were group housed (1–4 mice per cage), with treatment and control groups assigned based on genotype. All experiments were conducted during the light phase between 06:00 and 18:00, and repeated measurements were performed at similar times on consecutive days for the same animal.

### Recombinant adeno-associated viral (AAV) vectors

The following Cre recombinase-dependent adeno-associated viral vectors were used in the calcium imaging, optogenetic, and chemogenetic experiments: AAV1/9 CAG FLEX GCaMP6f (titer $1.9 \times 10^{13}$, Addgene # 100836-AAV1), AAV9.CamKII.GCaMP6f (titer, $2.5 \times 10^{13}$, Addgene# 100834-AAV9), AAV2-hSyn-DIO-hM4D(Gi)-mCherry (titer $1.9 \times 10^{13}$, Addgene# 44362-AAV2), AAV1-Ef1a-DIO-eNpHR 3.0-EYFP (titer $7 \times 10^{12}$, Addgene# 26966-AAV1), AAV1-Ef1a-DIO-eYFP (titer $2.7 \times 10^{12}$, Addgene# 27056-AAV1).

### Virus injections

To achieve Cre-dependent expression of genetically encoded fluorescent calcium indicators in SST+ cells, AAV vectors containing FLEX-GCaMP6f were micro-injected bilaterally (or unilaterally if noted) into the primary motor cortex (forelimb area[44,47]) of SST-Cre mice by stereotaxic surgery (AP, anterior-posterior to bregma: 0.0–1.0 mm, ML, medio-lateral to bregma: ±1.5 mm, DV, dorso-ventral:−0.5 mm from the brain surface). For the expression of calcium indicator in pyramidal neurons, AAV vectors containing a CaMKII promotor were injected into C57BL6/J mice. All other viral injections for optogenetic and chemogenetic experiments followed similar microinjection procedures. AAV2-hSyn-DIO-hM4D(Gi)-mCherry, AAV1-Ef1a-DIO eNpHR 3.0-EYFP, and AAV1-Ef1a-DIO EYFP were bilaterally injected depending on the experimental design. First, mice were anesthetized (Isoflurane 1–2% with respect to oxygen volume) on a secured stereotaxic frame (Kopf

Instruments, Germany). After the mice were deeply anesthetized, a longitudinal head skin incision was made to expose the skull. Craniotomies were carried out using a dental drill and the virus solution was microinjected using a 32-gauge syringe (Hamilton,#65458-01) at a volume of 300 nL/site with a rate of 50 nL/min. After infusion, the syringe needle was left in place for an additional 5–8 min before removal. Finally, the incisions were closed with VetBond (3M, 1469Sb). The animals were then placed into recovery cages for 3 days with partial heating, and back to home cages for at least 2–3 weeks before secondary chronic implant surgery. In chemogenetic experiments, mice recovered for 5-6 weeks in their home cages without implant surgery.

### Imaging lens and fiber implantation

2–3 weeks after the virus injection, craniotomies were performed above the injection site again under Isoflurane anesthesia. The skull surface was cleaned with a diluted hydroperoxide solution. After drilling and enlarging the hole in the skull (1–2 mm in diameter) to expose the target brain surface, the dura mater was removed from the surface using gentle aspiration. To image neurons in deep cortical layers (M1 layer 5/6), a 500-μm diameter, 4.0-mm long gradient-index (GRIN) lens (Inscopix,#1050-004417 or GoFoton, # CLES050GFT120) was slowly implanted (-0.2 mm/min) until the target depth was reached (DV:−0.5 mm from the brain surface). For superficial M1 layer (layer 2/3) imaging, a GRIN lens with a large diameter (1.0 mm, Inscopix#1050-004637 or 1.8 mm, Edmund #64-519) was placed on top of the brain surface using a stereotaxic frame. If necessary, a lens adaptor was attached after unilateral implantation of the GRIN lens. For optogenetic experiments, fiber ferrules (Thorlabs CFMLC12L02, 1.25 mm, 0.39NA, $L = 2$ mm) were bilaterally implanted (DV: −0.5 mm from the brain surface). Once the GRIN lens or fiber ferrules were securely located, cyanoacrylate (Parkell, C&B Metabond, Radiopaque L-Powder, S396) and dental cement (Stoelting, #51459) were applied to fix the optical implant. After the implantation surgery, animals received ketoprofen injections to reduce pain and inflammation for 3 days and then recovered in partially heated cages before returning to their home cages. Mice for calcium imaging experiments were singly housed after surgery. Mice expressing eNpHR or eYFP for optogenetic experiments were littermates and group-housed after fiber ferrule implantation.

### In-vivo calcium imaging

2–3 weeks after the implantation surgery, the quality of in vivo fluorescence images and spontaneous fluorescence transients were assessed in home cages and operant boxes without behavioral tasks (no levers, no cues). Before the operant training, this inspection was conducted for a week to validate the stability of the fluorescence images. During this time, the mice were also habituated to carrying a head-mounted microscope device and to handling. After the daily image inspection, the mice underwent additional habituation training (1 h a day), during which they carried a 3D-printed mock device (2.5 g) in their home cages. Once it was confirmed that the fluorescence imaging provided stable recordings of spontaneous cellular calcium dynamics, the mice underwent food restriction protocols for operant training. Fluorescence images were collected through three open-source head-mounted microscopes (LABmaker, Miniscope V3.2) and one commercial head-mounted microscope (Inscopix, nVoke System) with a sampling frequency of 10 Hz. Each head-mounted microscope device and specific recording parameters were exclusively assigned to designated animals throughout the duration of the experiments. Synchronized behavior log acquisition and fluorescence image recording were controlled by a digital controller (Med-Associates, MED-PC IV Behavioral Control Software Suite). The raw video images were analyzed using image processing software (NIH ImageJ 1.53, and Inscopix, IDEAS platform) and custom MATLAB scripts.

## Calcium image data analysis

The raw fluorescence images were compressed with a 1:4 spatial ratio using image processing software (NIH, ImageJ 1.53 or Inscopix, IDEAS platform) and movement artifacts were corrected (Inscopix, IDEAS platform). The field of view (FOV) was then adjusted and cropped to include images within the GRIN lens. To generate $\Delta F/F_0$ image stacks, the change in fluorescence ($\Delta F = F - F_0$) was calculated for each individual pixel using a custom MATLAB script, where F represents the fluorescence intensity of the pixel, with the baseline $F_0$ set as the fitted average curve of spontaneous fluorescence intensity of individual pixels along the time-axis (analysis code is available online). A CNMF-E algorithm was then employed within the custom MATLAB script to identify the cell locations/features and to extract the single-cell $\Delta F/F_0$ traces. In brief, the CNMF-E algorithm identified the cellular features and their central locations based on pixel-to-pixel signal correlation in the time domain. The custom MATLAB script automatically assigned regions-of-interest (ROIs) around these identified centers of individual cells. Within each ROI, it eliminated rank-1 background fluorescence and crosstalk components from other neurons. The script then recalculated the pixel-to-pixel signal correlation within the ROI and set the boundary of the dominant neuron. The average $\Delta F/F_0$ was computed within the assigned cell boundary. Thus, the $\Delta F/F_0$ in the single cell characteristics (Fig. 1j, k, Fig. 2d, e, and Fig. 3i, j) preserves the same unit representation ($\Delta F/F_0$ in %, where $F_0$ indicates the spontaneous fluorescence baseline) as in the $\Delta F/F_0$ image stacks. If post-analysis normalization was required for single-cell characterization it is indicated. Following automated trace extraction, identified neurons and their corresponding traces were manually inspected, and any mis-identified artifact cells were removed.

## Single lever-press (FR1) training

Food restriction started 3 days before any behavioral training. During the entire training period, the body weight of mice was maintained around 85–90% of their baseline weight by restricting food supply. On the first day in an operant chamber, the mice were trained to retrieve food pellets from the magazine with a random delivery schedule (no cue, no levers). On the following 3 days, the mice were trained on a continuous reinforcement schedule with a fixed ratio (FR1) schedule, where mice earned the reward by making a single lever press. Only one active lever was available. The total number of rewards gradually increased to 5, 15, and 30 each day. Each daily session ended when the maximum session time of 90 min or the maximum number of rewards set for that session was reached, whichever came first. During this 4-day pre-training, all animals were trained under untethered conditions. Then the following day was designated as Day 1(D1) of the FR1 training while calcium imaging was started. We defined Day 7(D7) as the baseline for the FR1 training, and the FR1 schedule was maintained until Day 21 (D21). In-vivo calcium imaging was performed once or twice a week. On days when the calcium image recording was not performed, mice carried a mock device with a 3-D-printed structure identical to the head-mounted microscope (minisciope.org 3D-models or inscopix, dummy microscope#1050-003762) during their daily operant task training. All mice underwent the same daily training protocol and received additional daily food supply (1–3 g depending on the daily body weight) after the operant task training in their home cages.

## Lever-press sequence (NTC-FR4 and TC-FR4) training

Mice underwent the same food restriction and followed the 4-day pre-training schedule (magazine and FR1 training) and the subsequent 7-day FR1 training schedule described above. After completion of this initial FR1 training, the sequence training schedule began. On the subsequent seven days, all mice were trained on a fixed ratio 4 (FR4) schedule, where the reward was delivered after the 4th lever press without a time-limit. We refer to this FR4 training as non-time-constrained (NTC)-FR4 training to contrast with the following time-constrained (TC)-FR4 training. The following day, the TC-FR4 training was conducted, in which mice were trained to perform organized and sequential lever presses within specific time criteria to earn a reward. In the first stage of the TC-FR4 training, four consecutive lever-press sequences had to be made within 4 s (1 Hz for 4 lever presses) to earn a reward. The 1 Hz schedule was maintained until all mice could achieve 20 complete sequences within 90 min. This training took 2–5 days. Next, a 2-s time limit (2 Hz for 4 lever presses) was assigned until all mice in a cohort accomplished 20 complete sequences within a 90 min session. If this training was insufficient, some mice could not earn any reward in the next updated schedules and experienced extinction responses. To complete the 2Hz-TC-FR4 sequence, freely moving mice took 5 days, optics-fiber tethered mice took 7 days and mice carrying the head-mounted microscope took 9 days to meet these criteria. Next, the 1-s (4 Hz for 4 lever presses) time limit was assigned. The calcium imaging experiment stopped on D21 after the first day (D1) of the NTC-FR4 training. Otherwise, the 4 Hz schedule continued for 5 days in chemogenetic and optogenetic experiments. The final 0.5-s time limit (8 Hz for 4 lever presses) was assigned for another 5 days to compare the behavior between control group and experimental group.

## Chemogenetic inhibition during sequence lever-press (TC-FR4) training

Operant training was conducted at least 6 weeks after the stereotaxic injection of AAV-DIO-hM4Di-DREADD. The SST-Cre-positive (Cre+) and Cre-negative (Cre-) cagemates were allocated into the hM4Di group and the control group, respectively, and were co-housed during the period of surgical recovery and behavior training. The food restriction, 4-day pre-training (magazine and FR1 training), 7-day FR1 training, and 7-day NTC-FR4 training were the same as described earlier. The progressive TC-FR4 training schedule followed the same criteria as described above. During the food restriction and 4-day pre-training periods, saline (10 mL/kg) was injected in the hM4Di and control groups. From the first day of the FR1 training, all mice got i.p. clozapine-N-oxide (CNO) (Tocris,#6329, 5 mg/kg 10 mL/kg) injection before daily training sessions until the last day of the TC-FR4 training. The CNO injection was performed 40–60 min before the operant task training and the mice stayed in their home cages until each session started. The training chamber assignment and daily training order were counterbalanced within each genotype. The male and Female mouse ratio was 5:5 within each genotype. All mice underwent the same daily training protocol and received additional daily food supply (1-3 g depending on the daily body weight) after the operant task training in their home cages.

## Optogenetic Inhibition during sequence lever-press (TC-FR4) training

Within the SST-Cre-positive (Cre+) cagemates, individual mice were randomly assigned to either the eNpHR or eYFP groups, and AAV-DIO-eNpHR (or eYFP) was injected accordingly. As a result, all the mice in the eNpHR group had cage mates from the eYFP group, and they were housed together, including during the period of surgical recovery and the entire behavioral training. The food restriction protocol started 2-3 weeks after the implantation surgery. Training protocols, including food restriction, 4-day pre-training (magazine and FR1 training), 7-day FR1 training, and 7-day NTC-FR4 training, remained consistent with previous descriptions. The 4-day pre-training was conducted under untethered conditions, while mice were tethered to optic fibers for 1 h per day in their home cages. Starting from the first day of the 7-day FR1 training, all mice were tethered to optic fibers (Doric Lenses Inc, #SBP(2)_200/220/900-0.37_0.5m_FCM-2xMF1.25) and received closed-

loop optogenetic inactivation during the operant training tasks. The laser illumination (Shanghai Laser, 532 nm, 15 mW at the cable tip with 200 μm core diameter) was turned on for 5 s (continuous illumination) following every lever-press. The progressive TC-FR4 training schedule followed the same criteria as previously described. The eNpHR and eYFP groups were counterbalanced for daily training order and training chamber assignment. Both male and female mice were used with 5:5 ratio. All mice underwent the same daily training protocol and received additional daily food supply (1–3 g depending on the daily body weight) after the operant task training in their home cages. The lever press rate during TC-FR4 1 Hz and 2 Hz training in tethered mice (Fig. 7b) was lower (~10 presses/min) than that observed in the freely moving condition (~20 presses/min, Fig. 6b). Nevertheless, this difference did not affect the overall consistency of behavioral results and their interpretation in the context of our experiments. However, such differences may be important to consider in future studies.

## Unsupervised hierarchical clustering for sequence (SEQ) classification

During TC-FR4 training, rewards were delivered by counting the most recent four consecutive lever-presses in real-time. If the total duration of these lever-presses fell within a specified time limit, the rewards were delivered immediately (delay less than 0.1 sec). Timestamps for individual lever-presses, reward deliveries, and head entries were recorded in logfiles. In post-hoc analysis, sequence (SEQ) trials were defined as consecutive lever-presses followed by either a head entry or a pause longer than 3 s. For the SEQ clustering, each sequence was itemized and stored in a structured database, including key parameters such as lever-press/SEQ, frequency/SEQ, inter-sequence interval, reward success, optogenetic laser activation status, types of injected opsins, biological information (e.g., sex, age, genotype) of mice, acquired $\Delta F/F_0$ traces from calcium imaging, and the calculated SEQ efficiency. These itemized SEQ database, totaling 330 trials across 8 mice, were randomly shuffled before classification. In the unsupervised hierarchical clustering, only three structural kinematic parameters, lever-press/SEQ and frequency/SEQ, inter-sequence intervals, were initially used without considering any other sequence information, such as reward presentation or calculated SEQ efficiency. Once the classes of SEQ (e.g., C1, C2, and C3) were assigned by the clustering algorithm, the classification information was updated onto the itemized SEQ database. Then the data were reorganized depending on these classes (Fig. 4c, f, and g). Reward-based classification (Fig. 4h-p and Fig. S4) was performed after the assignment of unsupervised classes. The hierarchical clustering and dendrogram analyses were conducted using a custom Python script (Python 3.10.6) with Ward linkage as implemented in scipy.cluster.hierarchy.linkage and sklearn.cluster.AgglomerativeClustering. The distance metric used was Euclidean.

## Definition of sequence (SEQ) efficiency

Sequence (SEQ) efficiency was developed as a mathematical expression for the simple description of structural features of efficient action sequences. It is defined to describe the highest coefficient value when the structure and kinematics of action sequences align with ideal patterns for a given task schedule, such as 4 lever-presses at 4 Hz in the 4Hz-TC-FR4 schedule. Although mice could potentially receive rewards by executing more lever-presses at a faster rate during a sequence trial, such structures are inefficient due to wasted kinematics. From this optimal point, SEQ efficiency gradually decreases as the sequence contains an excessive or inadequate number of lever-presses or operates at an exceptionally high or low lever-press frequency. We implemented an equation for SEQ efficiency by employing a positively skewed Gaussian distribution, denoted as *S(x)*, which resulted in a 3-dimensional curve as a function of lever-press number ($x_{LEV}$) and lever-press frequency ($x_{Hz}$) within a sequence. The

definition of SEQ efficiency is as follows:

$$SEQ\,efficiency(x_{LEV}, x_{HZ}) = 100 \times S(x_{LEV}) \times S(x_{Hz})$$

The skewed gaussian function *S(x)* is multiplication of a Gaussian function $\varnothing(x, |, \mu, \sigma)$ and a cumulative distribution function of the Gaussian distribution $\Phi(x, |, \mu, \alpha)$, and defined by the following formulas.

$$S(x) = 2\,\varnothing(x|\mu, \sigma) \times \Phi(x|\mu, \alpha)$$

$$\varnothing(x|\mu, \sigma) = \frac{1}{\sqrt{2\pi}} \exp \times \left( \frac{-(x - \mu)^2}{2\sigma^2} \right)$$

$$\Phi(x|\mu, \alpha) = \frac{1}{\sqrt{2\pi}} \int_{-\infty}^{\alpha(x-\mu)} \exp\left(\frac{-t^2}{2}\right) dt = \frac{1}{2}\left[1 + \mathrm{erf}\left(\frac{\alpha(x - \mu)}{\sqrt{2}}\right)\right]$$

By adjusting coefficients ($\mu$, $\sigma$, $\alpha$), the asymmetric curvature of the skewed gaussian function is determined. We set $\mu = 2$, $\sigma = 4$, and $\alpha = 0.5$ to generate the mapping function shown in Supplementary Fig. 8, where SEQ efficiency reaches its maximum value at Lever-press number/SEQ ($x_{LEV}$) = 4 and lever-press frequency/SEQ ($x_{Hz}$) = 4 Hz. This mapping function empirically aligned with the behavioral data in TC-FR4 training (Fig. 3), the modulation of sequence structures (Fig. 4), and results in chemogenetic/optogenetic experiments (Figs. 6 and 7).

## Brain slice electrophysiology recording

Mice were anesthetized with isoflurane and decapitated after deep anesthesia. The brain was rapidly harvested and transferred to a slicing chamber filled with ice-cold sucrose-based cutting solution containing the following (in mM): 194 sucrose, 30 NaCl, 4.5 KCl, 26 NaHCO$_3$, 1.2 NaH$_2$PO$_4$, 10 D-glucose, 1 MgCl$_2$, saturated with 95% O$_2$/5% CO$_2$. Coronal sections (250–300 μm) were collected using a Leica VT1200S Vibratome (Leica Microsystems, Buffalo Grove, IL, USA) and incubated in a chamber filled with artificial cerebrospinal fluid (aCSF) containing (in mM): 124 NaCl, 4.5 KCl, 26 NaHCO$_3$, 1.2 NaH$_2$PO$_4$, 10 D-glucose, 1 MgCl$_2$ and 2 CaCl$_2$, saturated with 95% O$_2$/5% CO$_2$. Slices were incubated for 30–60 min at 32 °C and then maintained at room temperature prior to recording.

For patch-clamp measurement, brain slices containing hM4Di-mCherry-expressing SST cells in M1 were transferred to an upright microscope (Scientifica, Uckfield, UK) and visualized using a LUM-PlanFL N×40/0.80 NA water immersion objective (Olympus, Waltham, MA, USA). For fluorescence imaging, excitation light (wavelength: 509–551 nm) was generated via X-Cite LED driver (Lumen dynamics) and a cy3 filter-cube set and delivered through the objective to identify hM4Di-mCherry-expressing SST cells. Current clamp recordings were performed using an intracellular solution containing the following (in mM): 140 mM K-Glu, 10 mM HEPES, 0.1 mM CaCl$_2$, 2 mM MgCl$_2$, 1 mM EGTA, 2 mM ATP-Mg, 0.2 mM GTP-Na. Micropipettes were made from 1.5 mm OD/1.12 mm ID borosilicate glass with a filament (World Precision Instruments, Sarasota, FL, USA) and resistance was 2–4 MΩ. Recordings were performed using a Multiclamp 700B and a Digidata 1550B digitizer with a low-pass filter setting of 2–10 kHz and a sampling frequency of 10 kHz. Data were analyzed using pClamp 10.6 software (Molecular Devices). After entering the whole cell configuration, we waited for 10 min and measured the baseline membrane potential. Afterwards, CNO (Tocris #6329, 10 μM) dissolved in aCSF was continuously bath applied for at least 10 min.

## Histology

After behavioral training and assessment, the mice were perfused with phosphate-buffered saline (PBS) solution, followed by 4%

paraformaldehyde. The brains were then extracted and postfixed in 4% paraformaldehyde. After 24 h of fixation, brains were transferred to PBS and stored at 4 °C. For sectioning, the posterior brain region from the superior colliculi was removed using a flat blade. Then, the flat coronal cross-section of the brain was positioned and fixed on the flat vibratome stage using superglue. 40 μm (or 80 μm)-thick sections containing M1 were prepared through the anterior to posterior axis using a Leica VT1200S Vibratome (Leica Microsystems, Buffalo Grove, IL, USA). After slice preparation, eNpHR-eYFP-expressing and GCaMP-expressing brain slices were mounted on slides and covered with thin cover glasses after applying DAPI Fluoromount-G (Southern Biotech, 0100-20). The slices expressing hM4Di-mCherry or GCaMP6f were inspected under an epi-fluorescence wide-field microscope (Zeiss AxioZoom.V16), and if necessary, additional immunofluorescence staining was performed. First, the sections were washed four times for 20 min each in PBS. Then, the sections were incubated in PBS containing 5% BSA and 0.2% Triton X in room temperature. One hour later, the solution was replaced with a 0.5% BSA and 0.2% Triton X/PBS solution (PBST) containing a 1:1000 dilution of Rabbit anti-DsRed (Takara, #632496) or Chicken Polyclonal anti-GFP antibody (Abcam, #ab13970), and the sections were incubated for 24 h at 4 °C. After the incubation in the primary solution, the sections were rinsed four times for 20 min each with new PBST solution before incubation in a secondary antibody solution containing a 1:1000 dilution of Goat anti Rabbit-AlexaFluor568 (Invitrogen, A-11011) or Goat anti-chicken AlexaFlour 488(Invitrogen, #A-11039) in PBST for 24 h at 4 °C. The sections were washed four times for 20 min in PBS before being mounted on slide glasses with DAPI Fluoromount-G (SouthernBiotech, 0100-20).

## Statistics

For the data presentation illustrating neuronal responses (Fig. 1 m,n,p,q, and Fig. 2b, c, f, i, n, and Fig. 3g, h, k, q, s), we averaged the trial responses of individual neurons to represent their characteristics. For all remaining data presentations, we initially computed the mean of these neuronal responses to represent an individual mouse and then calculated the averages across these individual mouse averages. The error bars represent the standard error of the mean (s.e.m.) across mouse averages. All unpaired and pairwise comparisons were two-tailed. Exact $P$ values are reported in the figure captions. For analysis of two experimental results (Fig. 2p, Fig. 3u) or two groups (Fig. 6 b,c,d and Fig. 7 b,c,d) across multiple time periods, two-way repeated-measures ANOVA was used. For analysis of a single group across variables (Fig. 4 d,e,g), one-way repeated-measures ANOVA was used. When there were multiple comparisons, the Sidak post hoc test was used when needed. Pairwise Pearson correlations between cell activity profiles were computed using MATLAB, yielding a cell-by-cell correlation matrix for each mouse. Mean off-diagonal correlation coefficients were used as a descriptive measure of population synchrony and are shown for individual mice

## Reporting summary

Further information on research design is available in the Nature Portfolio Reporting Summary linked to this article.

## Data availability

Raw data can be found in Source Data file provided with this paper. Source data are provided with this paper.

## Code availability

Examples of custom codes used in this manuscript are deposited in the database and publicly accessible at https://github.com/Jeongoenlee/SST.

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

## Acknowledgements
We would like to thank Drs. Karina Abrahao, Armando Salinas, Shana Augustin for advice on the project, technical support, and personal training. We also thank Drs. Joseph Cheer, Qing Liu, Kuan Hong Wang, Morgana Favero on technical support on head-mounted microscope systems, Dr. Andrew Kesner for help in use of the optogenetic laser controller, and Guoxiang Luo and Aurora Sheridan for tissue sample preparation. This work was supported by the Division of Intramural Clinical and Biological Research of the National Institute on Alcohol Abuse and Alcoholism (NIAAA).

## Author contributions
J.O.L., S.B., and D.M.L. conceived and designed the study. J.O.L. performed imaging, software programming, and behavior experiments, and all data analysis with input from S.B. and D.M.L. G.S. performed brain slice electrophysiology experiments with input from S.B. and D.M.L. A.H. performed behavioral experiments, input from J.O.L. and D.M.L. J.O.L., S.B., G.S., A.H., and D.M.L. discussed data and wrote the manuscript. D.M.L., J.O.L., and S.B. directed the project.

## Competing interests
The authors declare no competing interests.
