## [Transparent Peer Review file · Nature Communications]

Motor cortex somatostatin interneurons adaptively shape the structure of action sequences

Corresponding Author: Dr David Lovinger

Version 0:

Reviewer comments:

Reviewer #1

(Remarks to the Author)

This paper is a comprehensive investigation of the role of SST neurons on both single and repeated lever pressing (simple vs complex actions). The results as presented suggest that SST neurons play a role in high-frequency lever pressing behavior. This interpretation is supported by both single cell calcium data as well as optogenetic and chemogenetic manipulations. Overall the paper is a solid contribution to our understanding of how interneurons modulate action.

I mainly have concerns with the framing/presentation of this paper. In the introduction results from a number of different aspects of motor processing are presented/conflated together (skilled motor learning, action generation, sequence generation, etc.). Thus, I can't tell right now if the main point about this paper is related to an argument linked with learning or instead if the main point about this paper is about action execution.

This confusion (in my mind) persists through the results section. Authors seem to suggest that their investigation is on the role of SST neurons on the execution of a learned motor sequence (i.e. that it's about learning). This could be the case. It could also be the case that SST neurons are simply involved in lever-pressing actions - and thus, when animals lever press at a high frequency, there is more SST neural activity (in other words, that this is unrelated to learning). Despite going over the paper carefully, it's unclear to me which argument authors are making...though they seem to be leaning towards the "learning" aspect?

The second issue (which relates directly to the above): "action sequence" to me implies some care about the sequence of the actions being performed. In other words, if I have to move my arm up, to the right and then down in order to complete some action (or, alternatively, respond in a patterned way "left lever, right lever, right lever, left lever". In this case, there is no "sequence" per se - it's just repetitive motor responding and authors are training the animal on how fast they need to respond. Thus in my mind it shouldn't really be framed as action sequence (and, in fact, there is a confound in naming it as an "action sequence", when really the results could just be related to action frequency (it's the same action over and over again, and how fast this action is being performed determines the outcome).

I made some specific suggestions below regarding additional analyses that could be done to help flesh these out.

In summary - I think the results are interesting and important, but the framing is confusing on whether the authors are claiming a role of SST neurons on learning vs. action execution; and whether this should be termed an "action sequence" as opposed to, again, more simple action execution.

Some specific comments below:

1. Intro:

A. As noted above, I found this intro a bit all over the place. In particular, they mention roles of motor cortex in skilled motor learning; skilled motor execution; execution of action sequences; associative learning, etc.) and discuss theories as if all of these were the same. This makes it a bit hard to follow exactly what it knows regarding the role of cortical circuits in an action sequence task (i.e. in the task they are studying). I would recommend editing/revising intro to be a bit more focused on what is known about cortical circuits involved in action sequencing, make that really crystal clear, and then it may be ok to

introduce other tasks in a way that makes it clear these are distinct from what is being studied in this paper.

B. Similarly - I don't really need to know what VIP and PV neurons do in the intro, and the role that SST neurons play is thus a bit opaque and confusing (I had to read the paragraph a few times to figure out what you are trying to say about the SST neurons specifically). I'd recommend focusing on theories specifically related to SST neurons (discussion of other INs can well be relegated to discussion).

C. Finally, in the last paragraph, authors jump right into "FR1 and FR4 sequences", which MAY BE well known to them, but likely deserves at least a quick introduction for other readers in terms of how the task fits into the larger literature of motor tasks referenced.

In summary - please spend more time introducing the points directly relevant to this paper, make the hypotheses a bit more clear (based on other papers, what do you expect SST IN to be doing); and describe the task a bit more.

RESULTS

I discuss based on figures:

Fig 1

There is no statistical reporting that I could see in the results or figure for any of the conclusions made about data from Fig 1. There are many things that, by eye, look like they would be quite statistically significant...but, again, without any form of testing it's just hard to say definitively. Authors need to refine the writing/description of this figure in the results section to reflect that findings are "examples" (and it's unclear if they are representative examples at that given the lack of significance testing). In other words: either add some statistics or make it very clear that the point of Fig 1 is purely for illustration purposes.

"Trial consistency" subpanel is just the mean effect, which doesn't really tell much about trial consistency.

The only statistical analysis I could see from Fig 1 was in the extended fig. In that analysis, (Pearson's correlation with population activity), I wonder if there is a difference in sample size between PYR neurons and SST neurons, and whether this difference in sample size drives differences observed (I imagine the mean Pearson's correlation statistic is sensitive to very small sample sizes which can then spuriously inflate or reduce effects).

Fig 2. Fig 2d-g (and supp fig). Authors show that the "peak" calcium response is unchanged with continued imaging, but the event locked peak goes down. This is a nice finding. It may be also nice to also analyze a "base-line" period pre lever press (i.e. in the -5 to -4 second period prior to the lever press, for example) to show that the background/baseline calcium activity is similar across days, it's just the peak effect post-lever press that is diminished. It is very nice that authors statistically analyzed calcium activity both at a single cell level and at an animal level for the major findings in this figure.

Fig 3. Authors show greater activity in SST neurons from TC-FR4 (4Hz) compared to NTC-FR4. However, as the authors themselves note, in the time window analyzed, there are (on average) 2-4 times more motor actions in the "motor sequence" of TC-FR4 compared to NTC-FR4 (i.e. there is a higher response frequency). This, coupled with the slow calcium dynamics, complicates interpretation. In my opinion, most of the results of this paper could be explained by a more simple idea: SST neurons are simply active with every lever press. Thus, with repeated lever presses (at a higher frequency) result in greater calcium influx (smoothed of course by the slow temporal dynamics of calcium sensors). Even the early findings related to FR1 could be explained by this: often, in motor learning, early on animals move a lot...and with training they settled down and have more efficient movements. Thus, it could well be that on D1 FR1 training animals press the levers multiple times (thus resulting in greater SST activity), but that at D7 FR1 training animals only press the lever once, thus resulting in diminished SST activity.

Thus - my major point for this (and other figures) - there is no clear evidence (YET) that changes in SST activity are a function of learning per se. Authors either need to make this more clear throughout...or perform some further analyses that may be more convincing that this is a learned phenomenon.

The authors may have the data to clarify this - mostly, by analyzing FR1 and NTC-FR4 data, and analyzing calcium data by frequency of response/sequence. If they find there is a frequency effect, that would support my idea. If they find that, in animals not trained on TC-FR4, there is no frequency effect on SST activity, this would support something more about learning. To make this more clear: in the early stages of training - take all trials performed at 0.5 Hz, 1 Hz, 2Hz, 3Hz and 4Hz (separately for D1 and D7) and see if there is any difference in SST activity when sorted that way. Analyze the NTC-FR4 and FR1 data the same way (sorted by frequency of responses).

Fig 4 Interesting analysis and a different way of answering the questions I have above - though this figure further bolsters my hypothesis that the elevated SST activity associated with TC-FR4 is not related to learning per se but instead linked with higher frequency motor activity (this figure seems to demonstrate that even in the well-trained animals, the faster frequency responses are linked with elevated SST activity relative to the lower frequency responses). Thus, again, it is unclear if the SST activity noted is at all a learned phenomenon or simply a cortical execution phenomenon (and a proxy of slow calcium dynamics)

Fig 5. Fig 5 is quite interesting and demonstrates, perhaps some evidence of a learned phenomenon. It would be very

interesting to analyze the NTC-FR4 and the FR1 dataset in this way and see if the presence/distribution of these delayed onset neurons emerges only in learning, or, again is a consequence of repeated actions. These "delayed onset" neurons could reflect some aspect of the motor sequence that is a learned/trained phenomenon

Fig 6-7. Very interesting results - but again, what is the effect of inhibition on frequency of responding?

(Remarks on code availability)

Reviewer #2

(Remarks to the Author)

This study by Lee et al. investigates the dynamics and functions of SST interneurons in motor cortex during a freely moving lever-pressing task. They use miniscopes to record Ca²⁺ signals from genetically defined SST neurons longitudinally throughout task acquisition and adaptations. The studies are well controlled, with rigorous analyses. Rigor is also enhanced by the inclusion of analyses of both neurons and mice as samples (ensuring that the reported effects aren't driven by an outlier mouse). The authors make numerous novel findings related to both the dynamics and functions of SST neurons, across a sequence of behavioral challenges. Furthermore, apparent inconsistencies with the related published literature are appropriately contextualized in the discussion. The comments below are included to help strengthen an already strong manuscript.

Major points

1. Perhaps the most important motivating question (as included in the abstract) is "the mechanisms by which cortical microcircuits actively fine-tune the timing and structure of action sequences, enabling organisms to adaptively maintain motor efficiency across varying task demands, remain unclear." While the manuscript makes multiple important observations towards this goal, the answer is far from complete. To close the gap even further, I recommend the following analyses.

- Fig 1. How might the SST neurons be influencing PYR activity (and vice versa)? From the population activities (Fig 1j,k) is there a phasic relationship between these populations (e.g., SST leading vs lagging)?

- The description of SST relationships to 'rotational dynamics' and neuronal oscillations is compelling in the FR4 schedule. Obviously, this would be best assessed with simultaneous recordings of PYR neurons. However, perhaps more can be done with the existing dataset. As possibly evident in Fig 3p, were there oscillatory dynamics in the SST population activity? This would have to be assessed either on individual trials (and then average the frequency spectra across trials), or across trials with similar motor sequences (minimal jitter of individual lever presses).

- Fig 5j-n – for the delayed-onset SST neurons, were there specific phase relationships to individual lever presses? This can be addressed for individual SST neurons and across the population.

- Fig 6 – Did SST suppression change the frequency distribution of the lever pressing? Again, analyses of individual trials (and then averaging the frequency spectra across trials) would be needed.

2. Early in the study the authors make important observations regarding differences between L2/3 vs L5 SST neurons. Were such laminar analyses performed in the later studies? For the FR4 studies of dissociable population dynamics, are both populations distributed in superficial and deep layers?

3. In the FR1 D7 SST data, there appears to be two peaks in the average data. Are these peaks from separable populations, as described in the FR4 schedule data?

4. The authors describe in the Discussion that their studies cannot rule out changes in more subtle forelimb kinematics during SST suppression. However, the same limitation applies to the observations in Fig 2d vs 2e. Changes in the velocity or trajectory of limb movements may explain the differences in SST activity. If the authors have the body/limb movement data, I encourage them to present the data to address this potential confound. If not, then mentioning this in the discussion would be important.

Minor points

1. Fig 2o – expand y-axis to better visualize data

2. Extended Fig 6c, right- expand y-axis to better visualize data

3. Line 311 - Extended Fig 7b?

4. Please clarify: "Although we also observed slight differences in initial motor task learning between tethered and untethered mice, we did not detect significant circuit inhibition effects between experimental and control groups (Fig.6b and Fig.7b)."

(Remarks on code availability)

Reviewer #3

(Remarks to the Author)

In this manuscript, the authors investigate the role of SST interneurons in learned motor behavior by measuring and manipulating neural activity in mice. Mice were trained to perform lever-pressing tasks of varying difficulty and speed to earn rewards. The authors observed task-related activity modulation in both pyramidal neurons and SST interneurons, with higher SST activity in deeper cortical layers. SST activity was elevated during early task learning and varied with movement type across tasks. SST activity appears elevated during early task learning and varies with movement form across tasks. SST cell activity is varied in its relation toward movement, with a preponderance of cells active at movement initiation. Both chemogenetic and optogenetic perturbations of SST neuron activity lead to performance deficits specifically during the most challenging versions of the task.

The results overall are novel, interesting, and have relevance to our understanding of M1 function during both motor learning and execution. Below I describe a number of concerns that should be addressed during revision.

Support for the specificity of SST cell role

The manuscript focuses on SST neurons, and implies that SST neurons play a specific role in executing more challenging lever press tasks. For example, on line 283 they refer to “a direct and specific contribution of action related SST IN activity,” and on line 434, they refer to “insights into the specific functionality of M1 cortical INs.” While it is certainly appropriate to focus on the role of one cell type, the data included in the manuscript do not make a case for a specific role of SST cells, as distinct from that of other cell types. No comparison is performed to other interneuron types, like PV cells for example, and no differentiation from the overall inhibitory interneuron population is demonstrated. These would involve separate experiments using PV-cre or Vgat-cre mice. I am not suggesting that the authors must do such experiments, but instead that they should avoid suggesting that their data supports a specific role for SST interneurons.

Along these lines, it is questionable whether the specific perturbation effects reported in Figures 6 and 7 result from perturbing SST cells in particular or from the general disruption this causes to M1 activity. The observation from these experiments, of a task-specific effect of SST cell activity perturbations, could reflect either a specific role for SST cells or a role for M1 in general. Moreover, much existing data suggests that the influence of M1 on movement is specific to tasks that challenge dexterity. For example, mammals retain the capacity to execute many motor behaviors after M1 lesions, but struggle with tasks that are particularly dexterous. Based on these previous results, one might expect that perturbations to M1 activity in general would show similar effects as those reported in Figures 6 and 7. Again, I am not necessarily asking for more experiments here, but this alternative interpretation should be highlighted as well.

Key features of cortical involvement in movement execution

The manuscript could do a better job of reviewing current understanding of M1 in movement execution, and how it relates to the previous observations of cortical disengagement across learning and the observations reported in the present manuscript. The authors here do note that previous observations of cortical disengagement have come after experimental animals have completed very large numbers of trials over many weeks and have become highly stereotyped in their execution. Also important is the fact that these observations come from tasks that do not have a particularly strong requirement for dexterity. In fact, in Kawai et al., *Neuron*, 2015, the authors state that their task was specifically designed to avoid dexterity, as their central goal was to distinguish the role of M1 in motor learning from its role in executing dexterous movements, which had not been experimentally decoupled in previous studies. A very large amount of previous literature, ranging from older lesion studies (e.g. Lashley, *Arch Neuro Psychiatr*, 1924 or Travis, *Brain*, 1955), to more recent studies using optogenetics (e.g. Guo et al., *eLife*, 2015 or Miri et al., *Neuron*, 2017) suggest that M1 plays a specific role in the execution of dexterous motor behaviors. Other evidence has implicated M1 specifically in adaptive or corrective movements, when stereotyped movement trajectories need to be deviated from (e.g. Drew et al., *Can J Physiol Pharmacol*, 1996, Warren et al., *eLife*, 2021, or Bollu et al., *J Neuropys*, 2024). These previous results help explain why the role of M1 or M1 SST cells in particular, may be specific to the more challenging lever press task the authors used in their manuscript.

This could also help explain the divergence between results with and without head-fixation: lever press behaviors may become more challenging for mice to perform under head fixation, thereby engaging motor cortex in tasks that might otherwise not involve M1 if performed in a more natural, unconstrained setting. This is important to note, given the prevalence of head-fixed behavioral paradigms in motor system studies.

Concerns about the damage to M1

It appears that the bulk of the paper reports activity measurements from L5/6 SST cells imaged through cylindrical GRIN lens inserted into M1. This lens is 0.5 mm in diameter and was inserted to a depth of 0.5 mm into cortex. This volume of tissue disturbed by the implantation is large relative to the overall volume of forelimb M1 in mice. In addition to destroying upper layer circuits that send substantial input to deeper layers (layer 5A begins less than 0.5 mm from the pial surface), this will also disrupt the apical dendrites of deeper layer cells. Moreover, the authors do mention a slower learning in tethered animals, which might reflect damage to M1. Could the increased activation of SST cells in L5/6 activity measurements reflect a compensation for the loss of inhibitory interneurons superficially? It would be reassuring to see results confirmed using a triangular GRIN lens.

Lack of evidence for discretely separable activity patterns

The analysis of activity patterns in Figures 4 and 5 is described as implying that there are distinct types of SST cells - separate classes or clusters. For example, on line 231, the authors refer to “mutually exclusive neuronal subpopulations.” Instead, the data presented here are consistent with a continuous distribution of response types. This distribution is important to quantify and flesh out, but the data do not support the existence of discretely separable types. Seeing differences across the averages of clusters, which come from algorithms fated to separate populations into groups, does not support the idea that discrete or substantially distinct clusters exist. If you divide a continuous distribution into 2 halves, the means of each half will differ from one another, even though they each derive from a single underlying distribution. The manuscript does not provide evidence that the differences across neurons are beyond those expected by chance for a single distribution. Moreover, the plot in Figure 1f shows the distribution of response types across cells, which looks continuous. Again, the quantification of response diversity here is important, but the interpretation that discretely different response types exist should be avoided.

Missing statistical tests for key comparisons

In several cases throughout the manuscript, the authors refer to differences between experimental groups without reference to appropriate statistical tests in either the text or figure legends. For example, the difference between plots in Figure 1 n and q is highlighted, but these actually look somewhat similar and no statistics are included to support a significant difference. Similarly, the description of results in Figure 5j described in lines 245-7 does not provide statistics, nor does the figure legend.

Time course of perturbation effects across learning

The increased magnitude of the perturbation effects later in training reported in Figures 6 and 7 is a very interesting finding. However, because of the experimental design, one possibility is that the effects are more severe at later time points because they reflect the accumulated effects of repeated activity perturbation. A control to exclude this possibility should be provided. For example, performing perturbations in a set of animals only after they have reached the 8 Hz training stage.

Interpreting the activity changes across FR1 learning

The changes in SST activity between early and late in learning the FR1 task (Figure 2) are striking and are potentially consistent with a specific role of the SST neurons during acquisition but not later execution. However, the revelation later in the paper that SST activity perturbations early in learning have little effect argues against this possibility. It is not clear how to reconcile these two results, and ultimately what to make of the changes seen in Figure 2. The Discussion should address this.

The sequence efficiency metric should be better explained

It seems like the sequence efficiency metric penalizes pressing too many times (more than 4) or too quickly (more than 4 Hz), but were the mice penalized in terms of reward when they overshoot? Also, how much the authors decided to penalize overshooting is unclear. Why not use a function where there is no penalty for pressing more or pressing faster? I suppose pressing more could result in longer trials leading to less rewards/min, and faster presses might be less efficient in terms of energy expenditure, but none of this is clearly explained. In addition, the authors do not describe how they arrived at the amount of skew that they used for the function. This would directly affect analyses like those shown in Figure 6 where inhibiting SST cells resulted in decreased efficiency leading them to claim that “chemogenetic-mediated hyperpolarization of M1 SST interneurons deteriorates sequence organization” (line 274).

Interpretation of activity differences between NTC-FR4 and TC-FR4

When looking at the changes in SST IN activity between the NTC-FR4 and TC-FR4(4Hz) conditions, one explanation for the increase in activity could be that the animals are doing more vigorous movements (more presses and/or faster presses). The division into classes in Figure 4 further highlights this, as the more vigorous SEQ class (class 1) seems to have the largest peak in the calcium transients. It seems that if you did a convolution of the lever presses with a smoothing kernel you could produce something that looks a lot like the average dF/F 's.

Minor concerns:

Line 32 (also line 322): I would caution against using the word “persisted” when talking about results for the TC-FR4 training, as it gives the impression that the activity is unchanging, when it is in fact changing. Maybe a more descriptive phrase like “did not decrease” is better.

Figure 1: When using calcium imaging, one must be careful when interpreting differences in dF/F magnitudes across different cell types. The magnitude of these relative changes depends on many factors unrelated to neuronal firing rates, like the expression level of indicator proteins and the calcium handling within cells (e.g. the intracellular concentration of endogenous calcium buffers). Given this, differences like those reported between SST cells in L2/3 and L5/6, which could be genetically distinct, do not necessarily reflect differences in activity level. The observations reported are meaningful and important, but this caveat in interpreting the differences should be made explicit.

Specifying the site of activity measurements: The end of the Results section on Figure 1 and the beginning of that for Figure 2 (page 4) seem to imply that all neural analysis is done on SST cells specifically in L5/6. However, L5/6 is not mentioned again specifically until late in the Discussion. Because there are repeated references to SST interneurons in the intervening text, it is a bit ambiguous whether all the neural activity analysis was specifically on L5/6 cells. In addition, from the description of the experiments reported in Figure 6 and 7, including the depth of viral injection, it seems that the perturbation experiments reported in these figures involved all SST cells, not just those in L5/6. I do not see big issues with this difference, but it should be made explicit.

Line 61: "M1 PYR neurons can even become independent of movement upon extended training" - this statement is a bit redundant given the preceding text.

Line 98: It would be helpful if this first section of the Results included at least a brief description of the behavioral paradigm.

Line 120: "synchrony" generally refers to simultaneous spike firing, but the measures used here, especially when applied to calcium sensor measurements, will reflect correlations of activity on longer timescales that do not seem to meet the typical definition of synchrony.

Line 171: "at" should be "during"

Line 180: "This indicates the maintenance of SST IN synchrony as mice efficiently reorganized their behavior from simple to more complex motor programs." This statement reflects an interpretation of the absence of a significant difference as reflecting the absence of a difference. Such inferences are not statistically valid. This interpretation should be removed.

Line 207: It should be clarified what is meant by "incomplete". For incomplete sequences, are the animals just choosing not to retrieve the reward, or is the reward not dispensed? If it is the former, is the reward just left there for the animal to get in the next trial, and could this behavioral change be related to satiation and/or motivation?

Line 217: "INs" should be "IN"

Line 234-235: Is the sharper response at higher frequency perhaps because the movement is faster? Also, not entirely clear what "stabilized" means here.

Line 288: "minute-to-hour" should be "minutes-to-hours."

Line 301: The authors justify the analysis shown in Figure 7i-7n as looking for changes "within a single experimental subject." The notion that the statistics here are showing differences within individual mice is not supported - groupwise stats are still used. Moreover, the results in these panels seem expected given the results in 7b-d. A different justification for including this analysis seems necessary, or else further clarification.

Line 312-313: references to Fig.7m and Fig.7l seem to have their locations switched.

Line 390-391: This description is confusing, because earlier (line 259) the authors do refer to what they measured as "kinematics."

Line 442: "pattern" should be "patterns."

Line 447: "might also regulate the termination of action sequences" - this could be easily tested by aligning activity on the last lever push.

Line 453-454: this bit is unclear - it is not clear what aspect of the figure indicates oscillatory activity. Activity in a motor region during an oscillatory movement to some extent has to be oscillatory. The Churchland observation of rotational dynamics was surprising because it was occurring during ballistic reaching at frequencies not apparent in the movement itself.

Methods: There is insufficient explanation of how the hierarchical clustering of the different sequence classes was done (was it single linkage? complete linkage? a density based method like DBSCAN? Also, what was the distance metric used?)

Figures 1b/2a/3a/4a: it is not clear what is being shown in the color bar at the top.

Figure 1q legend: says "activity map in (n)" but they probably mean "activity map in (k)"

Figure 3 legend: on line 90, lever is misspelled.

Figures 6e-h and Figure 7e-n: The figures here and the accompanying text in the Results, do not clarify when during the training schedule these effects were measured (unless I missed it). The previous panels in each figure locate the effects to later in the schedule - were these comparisons done specifically at those later time points? This should be clarified either way.

Extended Data Figure 5: The text here notes the differences between the Instant and Delayed onset populations, in terms of the differences between lower and higher frequency responses. However, in the figure, it does not appear that the same statistical tests were performed - the brackets in h and j indicate different comparisons. In addition, it is important to keep in mind that the absence and presence of significant differences in two statistical tests is not an appropriate way to establish a difference between two conditions. The appropriate approach here, as is standardly done when comparing drug effects versus those of placebo, is to directly measure the difference between the two conditions (here the response types) and show that the difference is significantly greater than 0.

(Remarks on code availability)

Reviewer #4

(Remarks to the Author)

The authors study how Sst cells participate in a lever press and sequential lever press task. They find that Sst activity is synchronous around lever presses, particularly in deep layers, reduces with training, and increases when mice switch to a more difficult task schedule. Chemo- or optogenetic suppression of Sst activity increased lever pressing but disrupted its structure leading to lower reward rates.

The work this paper describes is well done, and the conclusions reasonable. I therefore do not have any major comments or suggestions. Two minor suggestions:

1. In every figure please state which layer you are recording from. My understanding is that except for Fig. 1 it is all in L5/6, but this would be clearer if explicitly stated
2. Extended Figure 5a makes it look like the delayed-onset Sst cells are not synchronous, but rather fire in a sequence. If so, that is a contrast to the highly synchronous activity seen at lever pressing. Is that correct? If so it might be worth pointing this out.

(Remarks on code availability)

Version 1:

Reviewer comments:

Reviewer #1

(Remarks to the Author)

The authors did a truly commendable job in addressing all of the reviewers comments - a strong article is now even stronger and in my opinion is ready for publication.

(Remarks on code availability)

Reviewer #2

(Remarks to the Author)

I commend the authors on their thorough responses to all reviewers. My comments/concerns were adequately addressed. While some of the 'reviewer figures' could be helpful to include in the manuscript, I trust the judgement of the authors on these decisions.

(Remarks on code availability)

Reviewer #3

(Remarks to the Author)

I commend the authors for doing a thorough job of addressing the reviewer concerns. The bulk of my concerns have been suitably addressed. A few issues remain, which I address below.

- Concerns about the damage to M1

Apologies for not making my original concern clear. My concern is specifically with the comparison between L2/3 and L5/6 SST cells. While this comparison is only done in Figure 1, it does seem important, as the authors use it to justify their subsequent focus on L5/6 (line 140). I do not see a control included that assesses whether the difference in activity between L2/3 and L5/6 SST cells is due to the different degree of damage between the L2/3 and L5/6 imaged animals. It is also not clear how such a control could be done, since the cells immediately above the imaged L5/6 SST cells are the ones providing strong input to L5.

I think it is necessary that this confound be explicitly addressed in the Results when describing the differences seen between L2/3 and L5/6 SST cell activity.

- Lack of evidence for discretely separable activity patterns

This result shown in Figure R6 is great - it should be added to the manuscript to support the claim of "mutually exclusive neuronal subpopulations" which I see remains in the manuscript.

- Missing statistical tests for key comparisons (FIG1)

I still do not see a statistical test either in the text or figure legend that supports the claims made about Fig. 5j, that the responses are further delayed. One should be provided, or the claim should be omitted.

- Time course of perturbation effects across learning

The additional data presented here in Figure R7 seem to corroborate my original concern. These data present a very different picture of the effects of SST neurons on task performance at later time points. Though a significant difference is seen in the event distribution, the effect is relatively small compared to what is shown in Figure 6 and 7. Figure R7b does not align with Figure 6b-d or Figure 7b-d. Thus a substantial degree of the effects shown in those panels do seem to be a byproduct of repeated inactivation and not from the immediate effects of silencing SST neurons during the movement. The problem is compounded by the fact that the authors stress the difference in effects between easier and harder versions of the task, especially throughout the Discussion section on task specific inhibition effects. I think Figure 7 should be updated to include the data shown in Figure R7 and the interpretation of the inactivation effects in the manuscript should be updated to account for these new results. This would include eliminating the claims about stronger effects during the higher frequency tasks, which clearly are substantially confounded. If inactivation effects depend on the history of inactivation an animal has experienced during previous sessions, then comparing effects across stages of the TC-FR4 training does not seem appropriate.

- Interpreting the activity changes across FR1 learning

My concern here has not been well addressed. I understand the reasons why we might not see an effect of inactivation on behavior in certain cases. But we are still left without positive evidence that the activity of these neurons is important for the easier versions of the task. This is problematic because much of the paper regards these easier versions of the task, like the changes across learning shown in Figure 2. The manuscript should explicitly describe the sense in which these changes are meaningful, in the absence of evidence that the activity is important for behavior.

Also, to be clear, the discrepancy referred to on line 413 is not the discrepancy between the change in SST activity and the absence of a functional effect, as the authors state in their rebuttal on this issue. Instead, the discrepancy they refer to on line 413 is one between the authors' inactivation results and those from other studies. Thus, the passage beginning on line 413 does not directly address my concern.

- Figure 1: When using calcium imaging, one must be careful when interpreting differences in dF/F magnitudes across different cell types...

The issue here is that because calcium indicators serve as calcium buffers, their expression level influences the magnitude of intracellular free calcium changes after action potentials, which in turn affects the dF/F change caused by each spike. The "background fluorescence" and "non-action related calcium signals" the authors refer to are not the primary problem with comparing across cell types. Cell types could differ systematically in their indicator expression level, or in the expression of endogenous calcium buffers. Both will affect the size of spike-induced dF/F changes, confounding comparisons across cell types.

(Remarks on code availability)

Reviewer #4

(Remarks to the Author)

Again no major comments. However my first minor comment (to explicitly say what layer was recorded) was specifically about the figure legends - I would suggest to add a statement of what layer was recorded to Figures 2, 3, etc, in either the main figure panel or in the legend. It may say so in the main text, but there will be no ambiguity if it is in the figure itself.

(Remarks on code availability)

Version 2:

Reviewer comments:

Reviewer #3

(Remarks to the Author)

The authors have done a thorough and very thoughtful job addressing my concerns. With the updates to clarify and qualify a number of points, I think the manuscript is overall stronger. My concerns have been sufficiently addressed. Just for the sake of clarification (I am not asking for further changes), my previous concern about comparing dF/F values across cell types pertained mostly to the comparisons made between PYR and SST cells in Figure 1 and Extended Data Figure 1. These are distinct types that could exhibit systematic differences in the relation between spiking and dF/F because of differences in calcium handling between the two types. I think the authors are getting the right answer here, but I might have done the analysis using the normalized responses shown in 1e,f.

(Remarks on code availability)

Reviewer #1 (Remarks to the Author):

This paper is a comprehensive investigation of the role of SST neurons on both single and repeated lever pressing (simple vs complex actions). The results as presented suggest that SST neurons play a role in high-frequency lever pressing behavior. This interpretation is supported by both single cell calcium data as well as optogenetic and chemogenetic manipulations. Overall, the paper is a solid contribution to our understanding of how interneurons modulate action.

I mainly have concerns with the framing/presentation of this paper. In the introduction results from a number of different aspects of motor processing are presented/conflated together (skilled motor learning, action generation, sequence generation, etc.). Thus, I can't tell right now if the main point about this paper is related to an argument linked with learning or instead if the main point about this paper is about action execution.

This confusion (in my mind) persists through the results section. Authors seem to suggest that their investigation is on the role of SST neurons on the execution of a learned motor sequence (i.e that its about learning). This could be the case. It could also be the case that SST neurons are simply involved in lever-pressing actions - and thus, when animals lever press at a high frequency, there is more SST neural activity (in other words, that this is unrelated to learning). Despite going over the paper carefully, its unclear to me which argument authors are making...though they seem to be leaning towards the "learning" aspect?

(Response) We appreciate the reviewer's positive overall evaluation. While previous studies have primarily emphasized the role of M1 SST interneurons in motor learning (mostly in layer 2/3), our findings highlight the role of L5 SST interneurons in action control. While the action we are studying is indeed a learned action, our data are most consistent with a role for these neurons in efficient performance of the actions themselves as opposed to learning. In response to this and other comments in the review, we have revised the manuscript to further clarify the focus of our study. Please see our detailed responses to your specific feedback below.

The second issue (which relates directly to the above): "action sequence" to me implies some care about the sequence of the actions being performed. IN other words, if I have to move my arm up, to the right and then down in order to complete some action (or, alternatively, respond in a patterned way "left lever, right lever, right lever, left lever". In this case, there is no "sequence" per se - its just repetitive motor responding and authors are training the animal on how fast they need to respond. Thus in my mind it shouldn't really be framed as action sequence (and, in fact, there is a confound in naming it as an "action sequence", when really the results could just be related to action frequency (its the same action over and over again, and how fast this action is being performed determines the outcome).

(Response) We understand the question of whether our behavioral task qualifies as an "action sequence." While "action sequence" is often intuitively understood as a specific order of *distinct* actions, we adopt a broader view grounded in prior literature that also recognizes temporally structured, repeated actions as meaningful sequences. In this view, both Left-Right (L-R, alternating pattern) and Left-Left (L-L, repeated pattern) lever presses represent structurally distinct forms of action sequences. Even repeated actions can carry distinct functional meanings depending on the number of repetitions (e.g., L-L vs. L-L-L-L) and

temporal features such as rhythm, timing precision, or duration. For instance, a rapid, repeated press pattern (e.g., 'L–L–L' within 1 or 4 seconds) may differ substantially from a slower or more irregular version in how it is learned, initiated, and executed.

To learn and perform the TC-FR4 task (e.g., a 4 Hz schedule requiring 'L–L–L' within 1 second), it is important to note that mice do not simply increase lever press frequency through repetitive pressing. Rather, they explore and reorganize combinations of various behavior elements—including sequential lever presses and non-lever-related actions such as brief pauses or port access. Over the course of training, animals iteratively generate and refine numerous combinations of these behavioral elements to discover more efficient and effective behavioral patterns. Notably, brief pauses (visible as gaps in raster plots in Figure 4c) may play a crucial role in concatenating (or chunking) individual actions or some behavior units into longer, cohesive sequences.

Thus, the resulting behavior patterns (Figure 4b and c) were remarkably variable and diverse, requiring computational methods for classification. While simple measures such as lever press rate are one parameter that assesses overall animal behavior, we found that the concept of an action sequence provides a more meaningful framework for understanding how mice organize distinct individual actions into more efficient, cohesive behavioral structures found in our experimental results.

This broader conceptualization originated from extensive prior studies on the control and organization of action sequences. For example, studies by X. Jin et al. (Nature, 2010; Nature Neuroscience, 2014) defined action sequences primarily as temporally structured series of lever presses—often involving a single lever—rather than alternating lever actions. Similarly, Vandaele et al. (eNeuro, 2021; iScience, 2023) classified repeated lever presses into distinct sequence categories based on timing, structural completeness, and the combination of other motor behavior, such as the presence of within-sequence port entries. Furthermore, Kawai et al. (Neuron, 2015) and Wolff et al. (Science Advances, 2022) treated the temporal spacing between lever presses as a key characteristic of action sequence learning. Notably, all of these studies were conducted in the context of single-lever-press tasks.

In our experiment, we believe that the term “*action sequence*” offers a useful framework for interpreting the diverse behavioral patterns that reflect the organization of temporally cohesive action structures. It also helps compare our findings on cortical contributions to action sequence control in contrast to prior work focused on basal ganglia function—making the interpretation more accessible and coherent to a general neuroscience audience.

Some specific comments below:

1. Intro:

A. As noted above, I found this intro a bit all over the place. In particular, they mention roles of motor cortex in skilled motor learning; skilled motor execution; execution of action sequences; associate learning, etc.) and discuss theories as if all of these were the same. This makes it a bit hard to follow exactly what it know regarding the role of cortical circuits in an action sequence task (i.e. in the task they are studying). I would recommend editing/revising intro to be a bit more focused on what is known about

cortical circuits involved in action sequencing, make that really crystal clear, and then it may be ok to introduce other tasks in a way that makes it clear these are distinct from what is being studied in this paper.

(Response) To address this concern, we have substantially modified the introduction to clarify what is currently known about the role of cortical circuits—particularly M1—in the context of action sequence organization. We also unified key terminology to maintain consistency throughout the introduction. In response to the reviewer’s strong suggestion, we extensively revised the introduction.

(Revised Introduction)

Line 40: Humans and other animals organize action sequences by integrating discrete movements into cohesive units. With extensive, repeated motor training, these learned sequences become automatized and are executed as unified, holistic actions [1-4]. However, in many other behavioral situations, the organization of action sequences requires flexible modulation of their temporal and structural features depending on task demands [5,6]. This neural capacity for dynamic sequence modulation in response to changing behavioral contexts is essential for producing efficient and adaptive motor behaviors [7,8,9]. Despite its importance, the neural circuit mechanisms underlying the regulation of temporal and structural aspects of action sequences remains unclear.

Both cortical and subcortical structures, particularly the striatum and primary motor cortex (M1), are implicated in action sequence organization [10]. The striatum is involved in controlling well-learned action sequences by encoding neural signals related to sequence initiation, execution, and termination [2,3,10]. In contrast, M1 exhibits dynamic activity patterns that vary depending on the task demands and the stage of learning [11,12,13,14]. M1 may become functionally dispensable for executing overtrained behaviors, as evidenced by lesion studies showing that damage to M1 does not impair the performance of well-practiced routines [2, 15]. These findings support the “cortical disengagement” model [16], which proposes that cortical regions are critical for acquiring new skills but become less essential once motor behaviors are extensively trained.

However, other evidence suggests that M1 continues to encode movement or action sequence-related parameters even after extensive training [17]. For example, primate studies have shown that M1 exhibits anticipatory activity representing memorized action sequences during their execution [18]. Additionally, M1 neurons encode discrete elements of rapid, skilled action sequences, maintaining representations of individual movement parameters throughout sequence performance [19]. M1 activity also remains crucial for flexible execution of cue-guided sequences. For instance, cortical lesions impair adaptability to external instructions while sparing the execution of overtrained sequences [20]. Thus, it is possible that the neural circuitry in M1 is actively involved in organizing action sequences on a trial-by-trial basis, particularly when tasks demand temporal modulation or flexible reuse of learned action sequences.

Despite extensive research on M1 in general motor skill learning, how distinct neural circuit components within M1 contribute to the temporal and structural modulation of action sequences remains largely unexplored. Pyramidal (PYR) neurons and inhibitory interneurons (INs) constitute [21] key elements of heterogeneous microcircuitry in M1. They orchestrate action learning and execution through dynamic and plastic activity patterns finely tuned to specific movement parameters [22]. In M1 layer 2/3 (L2/3), PYR neurons progressively refine their activity during associative learning [14], encoding both sensory and motor components as learning advances [23]. Also, neighboring cortical INs, particularly somatostatin-

positive (SST) interneurons, play a critical role in regulating learning-dependent network plasticity by modulating synaptic and structural dynamics of PYR neurons [24]. Notably, SST IN activity in L2/3 exhibits task-specific modulation correlating with retention and maintenance of previously acquired motor skills [25, 26]. Thus, prior work has primarily focused on the role of SST INs in L2/3 during motor skill learning.

In contrast, neural circuits in M1 layer 5/6 (L5/6) are specialized for relaying motor commands to downstream targets such as the brainstem, spinal cord [27], and striatum [11], via pathway-specific subpopulations that independently encode movement direction and amplitude [10]. During action sequence organization, the PYR projection neurons in L5/6 convey sequence-related information; the repetitive components of these sequences are directly regulated via projections to the dorsolateral striatum (DLS), as demonstrated by optogenetic manipulations [28]. Although the activity of L5/6 PYR neurons is strongly influenced by local inhibitory circuits, the functional role of inhibitory interneurons, such as SST INs, in the generation and control of structured action sequences remains unclear. Given the critical role of M1 L5/6 microcircuitry in sequence organization [28], we hypothesize that SST INs play a distinct role in this process. Specifically, we investigate how SST INs in L5 shape the temporal and structural features of action sequences on a trial-by-trial basis.

B. Similarly - I don't really need to know what VIP and PV neurons do in the intro, and the role that SST neurons play is thus a bit opaque and confusing (I had to read the paragraph a few times to figure out what you are trying to say about the SST neurons specifically). I'd recommend focusing on theories specifically related to SST neurons (discussion of other INs can well be relegated to discussion).

(Response) Reflecting this comment, we eliminated discussion VIP and PV neurons (as shown the revised Introduction provided above).

C. Finally, in the last paragraph, authors jump right into "FR1 and FR4 sequences", which MAY BE well know to them, but likely deserves at least a quick introduction for other readers in terms of how the task fits into the larger literature of motor tasks referenced.

(Response) We also updated the meaning of FR1 and FR4 in the manuscript as follows:

(Updated Manuscript)

Line 87: In this study, we investigated single-cell calcium activity of SST INs in M1 L5/6 while freely moving mice progressively reorganized reinforced action programs —from a single lever press (fixed ratio 1, FR1), to four lever presses (fixed ratio 4, FR4), and ultimately to a rapid action sequence requiring four consecutive presses within a specified time window (time-constrained FR4, or TC-FR4).

In summary - please spend more time introducing the points directly relevant to this paper, make the hypotheses a bit more clear (based on other papers, what do you expect SST IN to be doing); and describe the task a bit more.

(Response) We have revised the manuscript to specifically addresses major concerns in framing our work.

RESULTS

I discuss based on figures:

Fig 1

There is no statistical reporting that I could see in the results or figure for any of the conclusions made about data from Fig 1. there are many things that, by eye, look like they would be quite statistically significant...but, again, without any form of testing its just hard to say definitively. Authors need to refine the writing/description of this figure in the results section to reflect that findings are “examples” (and its unclear if they are representative examples at that given the lack of significance testing). In other words: either add some statistics or make it very clear that the point of Fig 1 is purely for illustration purposes.

(Response) The primary aim of Figure 1 is to illustrate the experimental setup and provide representative examples of calcium responses. While our study focuses on SST INs in L5/6, we included calcium profiles of other neurons for highlighting that SST activity in L5/6 is strongly aligned with action. We agree that this idea should be supported by additional statistical analysis. We have included new statistical data in Extended Figure 1 to support the observed differences.

(Updated Manuscript)

Line 104: Our experimental setup and representative calcium traces are shown to illustrate raw activity patterns of cortical neurons in relation to the single lever press action (Fig.1). To examine the activity of identified cortical neuron subtypes during motor program acquisition, we trained mice in a self-paced lever-press task (Fig.1a). We monitored *in vivo* single-cell calcium activity in the M1 with a miniaturized fluorescence microscope and implantable gradient-index (GRIN) lenses.

"Trial consistency" subpanel is just the mean effect, which doesn't really tell much about trial consistency. The only statistical analysis I could see from Fig 1 was in the extended fig. In that analysis, (pearson's correlation with population activity), I wonder if there is a difference in sample size between PYR neurons and SST neurons, and whether this difference in sample size drives differences observed (I imagine the mean Pearson's correlation statistic is sensitive to very small sample sizes which can then spuriously inflate or reduce effects).

(Response) Thank you for pointing this out. We agree that differences in sample size (cell number) can influence the outcomes of Pearson's correlation analysis. In the calcium imaging experiment, a great number of PYR neurons were recorded compared to SST neurons. Given this imbalance, we have decided to separate PYR and SST INs in the Pearson's correlation analysis (updated figure below, **Extended Figure 1 e and f**). Also, we added statistical analysis of fluorescence signal amplitude. We updated the manuscript accordingly.

(Updated Figure)

Extended Figure 1 (c) Average $\Delta F/F_0$ responses of individual PYR neurons and SST INs in L2/3 and L5/6 aligned to lever press (Kruskal–Wallis test, main effect $p < 0.0001$). (d) Population-averaged $\Delta F/F_0$ responses across individual mice (one-way ANOVA, main effect $p < 0.0001$). (e) Cell-to-cell correlation coefficients of PYR neurons in L2/3 and L5/6, each dot denotes an animal measurement on D7 (two-tailed unpaired t-test, $*p=0.0357$) (f) Cell-to-cell correlation coefficient of SST INs in L2/3 and L5/6, each dot denotes an animal measurement on D7 (two-tailed unpaired t-test, $*p=0.0281$)

(Updated Manuscript)

Line 118: As expected, population average of PYR calcium traces in L2/3 and L5/6 exhibit minimal transient amplitude time-locked to the lever press (Fig. 1j). Notably, while SST INs of L2/3 showed small transient amplitude, SST INs in L5/6 exhibited pronounced transient amplitude in the population average of calcium traces around lever presses (Fig. 1k). Notably, while SST INs of L2/3 showed small transient amplitude, SST INs in L5/6 exhibited pronounced transient amplitude in the population average of calcium traces around lever presses (Fig. 1k). Further analysis of the calcium responses of individual neurons revealed that SST IN activity in M1 L2/3 exhibited variability in both calcium traces (Fig. 1l) and the corresponding peak normalized activity map (Fig. 1m,n). The same analysis showed that SST INs in L5/6 displayed consistent action-locked calcium responses (Fig. 1o,p,q).

Line 132: We found that the action-locked $\Delta F/F_0$ amplitude was higher in SST INs in M1 L5/6 compared to that of SST INs in M1 L2/3 and PYR in both L2/3 and L5/6 (Fig. 1j,k, and Extended Fig. 1c,d). Representative raw calcium traces and action-locked $\Delta F/F_0$ profiles that contribute to the population averages of SST INs in L2/3 and L5/6 are shown in Figure 1l–n and 1o–q, respectively. In these example traces, SST activity in layer 2/3 appeared more dynamic, whereas activity in layer 5/6 showed a more consistent pattern. To examine potential differences in network activity across cortical layers, we assessed Pearson’s correlation coefficients across cells. This parameter calculates the linear relationship between a single cell calcium trace and all other neuronal traces during a 10 second time window centered around the lever-press. The analysis revealed that both PYR neurons and SST INs in L5/6 exhibit higher cell-to-cell correlation compared to the calcium responses in L2/3. synchronized calcium response was higher in the SST INs than the PYR neurons in L5/6 (Extended Fig. 1e,f).

Fig 2. Fig 2d-g (and supp fig). Authors show that the “peak” calcium response is unchanged with continued imaging, but the event locked peak goes down. This is a nice finding. It may be also nice to also analyze a “base-line” period pre lever press (i.e. in the -5 to -4 second period prior to the level press, for example) to, show that the background/baseline calcium activity is similar across days, its just the peak effect post-

lever press that is diminished. It is very nice that authors statistically analyzed calcium activity both at a single cell level and at an animal level for the major findings in this figure.

(Response) Using the same dataset presented in Extended Figure 2, we calculated the $\Delta F/F_0$ of non-action-specific background signal, defined as events with peak amplitudes occurring more than 5 seconds away from action onset. The new analysis is added (Extended Figure 2 d,e), and no statistically significant differences were observed.

(Updated Figure)

Extended Figure 2. Action-locked SST activity in M1 L5/6 undergoes changes with motor training.

- (a)** Example $\Delta F/F_0$ population activity map of SST interneurons within a subject on D7 and D21 during an FR1 training session, with vertical bars indicating lever presses.
- (b)** Maximum $\Delta F/F_0$ response of L5/6 SST normalized to Day1 measurement. Fluorescence degradation was not significant (linear regression $R^2=0.0095$, slope non-zero $p=0.7038$) indicating consistent imaging measurement over 21 days. Each dot indicates max $\Delta F/F_0$ of one training session, and
- (c)** Comparison of the maximum $\Delta F/F_0$ ratio between D7 and D21 ($n=8$ mice, two-tailed paired t-test, $p=0.1276$)
- (d)** Spontaneous background $\Delta F/F_0$ over 21 days
- (e)** No statistical difference between D7 and D21 in the amplitude of spontaneous $\Delta F/F_0$ ($n=8$ mice, two-tailed paired t-test, $p=0.1276$)
- (f)** Action-locked $\Delta F/F_0$ response of L5/6 SST normalized to Day1 measurement. Response degradation was significant (linear regression $R^2=0.1595$, slope non-zero $p=0.0004$). Each dot indicates average $\Delta F/F_0$ time-locked to lever-press within a training session, and
- (g)** Action-locked $\Delta F/F_0$ ratio significantly decreased from D7 to D21 ($n=8$ mice, two-tailed paired t-test, $p=0.0020$). In all plots, shaded area and error bars denote SEM.

Fig 3. Authors show greater activity in SST neurons from TC-FR4 (4Hz) compared to NTC-FR4. However, as the authors themselves note, in the time window analyzed, there are (on average) 2-4 times more motor actions in the “motor sequence” of TC-FR4 compared to NTC-FR4 (i.e. there is a higher response frequency). This, coupled with the slow calcium dynamics, complicates interpretation. In my opinion, most

of the results of this paper could be explained by a more simple idea: SST neurons are simply active with every lever press. Thus, with repeated lever presses (at a higher frequency) result in greater calcium influx (smoothed of course by the slow temporal dynamics of calcium sensors). Even the early findings related to FR1 could be explained by this: often, in motor learning, early on animals move a lot...and with training they settled down and have more efficient movements. Thus, it could well be that on D1 FR1 training animals press the levers multiple times (thus resulting in greater SST activity), but that at D7 FR1 training animals only press the lever once, thus resulting in diminished SST activity.

(Response) We appreciate the reviewer's comments and acknowledge the possibility that slow calcium dynamics may lead to signal accumulation during high-frequency lever presses. However, several points that argue against these interpretations are already described in the manuscript, but we would like to highlight them here to clarify our interpretation:

Does higher frequency of lever pressing lead to greater SST calcium activity?

We agree that in some cases, elevated SST calcium activity correlates with tightly clustered lever presses during TC-FR4 training. Specifically, the data in Figure 4c and 4i,j,n are consistent with the interpretation that higher-frequency lever pressing is associated with increased SST activity peak. Similarly, Figure 3i and 3j compare SST activity between NTC-FR4 and TC-FR4 and support the same idea. However, this relationship does not generalize to the FR1 training condition.

We found no statistical evidence of multiple sequential lever presses occurring during early FR1 training. Instead, throughout the FR1 task, most lever presses occurred as single, isolated events. Additionally, despite the frequency of action (one press per reward) remaining constant or slightly increasing across the task duration, we observed that elevated SST activity during early training (Day 7) diminished by Day 21.

As we describe in the Discussion, this observation aligns with the cortical disengagement model, in which SST activity in M1 L5 may diminish as the same motor actions were extensively repeated. Consequently, we believe SST activity dynamics differ between FR1 and TC-FR4 training paradigms. We believe these differences are better explained by the framework of cortical engagement for adaptive tasks and disengagement for repeated tasks, rather than by calcium signal accumulation alone. But we think it is also worth mentioning the calcium signal accumulation concept in relation to Figure 3. We added the following sentences to reflect the reviewer's comments.

(Manuscript)

Line 378: In the more complex, time-constrained action sequences (e.g. TC-FR4), we observed that a strong correlation was maintained between SST INs activation and action sequence execution as task demands increased and proficiency decreased (**Fig.3**). If individual lever presses or discrete action components evoke a calcium transient in SST INs, the elevated calcium signal observed during time-constrained action sequences may reflect the accumulation of calcium across multiple presses or concatenated actions. This SST INs activity was modulated based on the structure of on-going action sequences (**Fig.4**).

Thus - my major point for this (and other figures) - there is no clear evidence (YET) that changes in SST activity are a function of learning per se. Authors either need to make this more clear throughout...or perform some further analyses that may be more convincing that this is a learned phenomenon.

The authors may have the data to clarify this - mostly, by analyzing FR1 and NTC-FR4 data, and analyzing calcium data by frequency of response/sequence. If they find there is a frequency effect, that would support my idea. If they find that, in animals not trained on TC-FR4, there is no frequency effect on SST activity, this would support something more about learning. To make this more clear: in the early stages of training - take all trials performed at 0.5 Hz, 1 Hz, 2Hz, 3Hz and 4Hz (separately for D1 and D7) and see if there is any difference in SST activity when sorted that way. Analyze the NTC-FR4 and FR1 data the same way (sorted by frequency of responses).

(Response) The reviewer makes an interesting point here. We now emphasize in the introduction and discussion our study ended up providing more evidence for the role of layer 5/6 SST IN activity in action sequence control. This is distinct from previous work that primarily focused on the contribution of SST INs to learning-related processes. The reviewer raises the question of whether SST activity correlates with lever press frequency per action sequence (**FPS** = # of lever presses / sequence duration) even prior to sequence training or in early stage of the training. This analysis would help clarify whether emerged SST activity is learning-dependent or the activity merely reflects motor output. The following paragraphs discuss our attempts to apply this analysis.

Is SST activity a function of learning?

A specific request was to analyze SST activity for similar sequence types during FR1, NTC-FR4 or TC-FR4 training. We found that the FR1 and NTC-FR4 sequence tasks did not generate sufficient sequence types for a meaningful analysis (as these do not involve time criteria). In contrast, TC-FR4 training allowed to generate rapid sequence patterns. Therefore, we performed our analysis by comparing SST activity during the TC-FR4 training.

Figure R1. SST activity potentially reflects frequency components in action sequences as learning progresses.

(a) Representative lever press patterns corresponding to FPS of 2Hz, 4Hz, 6Hz. **(b)** Average calcium peak amplitude time-locked to the initiation of action sequences with different action sequence structures, 2 Hz, 4 Hz, and 6 Hz FPS (two-way ANOVA repeated measure, 2Hz vs 4Hz vs 6Hz group effect: * $p=0.0015$, learning effect early vs late: * $p=0.019$, post hoc comparison between early vs late in 2Hz FPS: * $p=0.031$) **(c)** Daily monitoring of calcium transient peaks during execution of action sequences. Two distinct sequence types—2 Hz and 6 Hz FPS—are shown. Sample size ($n=3$ mice) is low.

To address the reviewer's question, we had to select lever press frequency per action sequence (**FPS** = # of lever presses / sequence duration) bins for sequence classification. In fact, sequences with FPS of 1Hz (two lever presses around 2 sec) or 0.5Hz (around 4 sec) were rarely generated. Thus, lower frequency bins such as 0.5 Hz, 1 Hz were not suitable for analyzing our experimental data set. Instead, we selected lever press frequencies of 2 Hz, 4 Hz, and 6 Hz (with a ± 0.2 Hz range for each, **Figure R1.a**), as these were consistently observed throughout the training period. In the existing data set, we first compared two distinct time points representing early (first 3days of TC-FR4 1Hz) and late stages of TC-FR4 training (last 3days of TC-FR4 4Hz). As shown in **Figure R1.b**, SST activity corresponding to the execution of slow sequences (2 Hz FPS) decreased significantly following TC-FR4 training. In contrast, no significant learning-dependent change was observed for faster sequences (4 Hz or 6 Hz FPS). Thus, the SST activity that emerges during the execution of fast-frequency sequence appears to reflect an action-dependent phenomenon rather than a learning-related modulation.

In response to the reviewer's suggestion, we additionally performed a separate experiment and analyzed SST activity across daily training sessions (**Figure R1.c**). When calcium peak progression across training days was plotted for two distinct action sequence types (e.g., 2 Hz and 6 Hz FPS), we observed that SST activity associated with 6 Hz sequences remained elevated, whereas SST activity related to 2 Hz sequences declined. These findings indicate that rapid action sequences consistently engage elevated SST activity from early in training, emphasizing the action execution aspect of SST activity. In contrast, the decline in SST activity for slower sequences may reflect learning-dependent changes or disengagement similar to what was observed with FR1 training. However, as this interpretation remains preliminary and requires further investigation, we prefer not to emphasize this aspect in detail within the current scope of our study. Overall, our findings are most consistent with the idea that SST IN activity is related to the frequency of action performance within a sequence and not the learning of faster action sequences per se.

Fig 4. Interesting analysis and a different way of answering the questions I have above - though this figure further bolsters my hypothesis that the elevated SST activity associated with TC-FR4 is not related to learning per se but instead linked with higher frequency motor activity (this figure seems to demonstrate that even in the well-trained animals, the faster frequency responses are linked with elevated SST activity relative to the lower frequency responses). Thus, again, it is unclear if the SST activity noted is at all a learned phenomenon or simply a cortical execution phenomenon (and a proxy of slow calcium dynamics)

(Response) Our data presented in Figures 4 and 5 underscore the role of SST activity in action execution. While we observed modest learning-dependent changes, the central focus and novel contribution of our study is to highlight the role of SST interneurons in controlling action sequences on a trial-by-trial basis (in the original manuscript, we used the term 'trial-by-trial' twelve times to emphasize this point). We have further elaborated on this perspective in the revised Introduction in response to the reviewer's comments. Also, the current dataset supports this framework, and the additional experimental results provided in Figure R1 further reinforce this concept. Thus, we believe the reviewer's perspective is well aligned with our data and interpretation.

Fig 5. Fig 5 is quite interesting and demonstrates, perhaps some evidence of a learned phenomenon. It would be very interesting to analyze the NTC-FR4 and the FR1 dataset in this way and see if the presence/distribution of these delayed onset neurons emerges only in learning, or, again is a consequence

of repeated actions. These "delayed onset" neurons could reflect some aspect of the motor sequence that is a learned/trained phenomenon

(Response) We appreciate the reviewers' interest in our data and experimental results. The reviewer suggests analyzing the NTC-FR4 and FR1 datasets to determine whether similar "delayed-onset" SST subpopulations can be identified in the early training phase. As shown in Figure 1o, which illustrates the activity of a representative SST neuron, these neurons respond to both behavioral events during FR1 and NTC-FR4. However, during these early training phases, "delayed-onset" neurons are not clearly distinguishable. Moreover, as training progressed under these schedules, lever press-related SST activity gradually diminished, making it difficult to identify distinct subpopulations.

We speculate that a learning-dependent mechanism may underlie the eventual consolidation of SST activity into distinct activity patterns—such as the "instant" and "delayed" onset patterns. While a few other learning-dependent features of SST activity are also of interest, exploring them in detail may broaden the scope of the current study beyond its primary focus. Our experiments and data interpretation are limited once the SST activity is separated into "instant" and "delayed" onset patterns. Therefore, we believe that the learning-dependent reorganization of SST activity during long-term sequence training warrants future investigation. In this manuscript, we have chosen to focus on the role of SST activity in trial-by-trial action sequence control, as supported by our current dataset. However, we added the following comments:

(Manuscript)

Line 545: Future investigations should focus on elucidating how M1 SST INs regulate cortical network output and their impact on the striatal circuit with respect to reorganization of complex motor programs. Also, the established SST activity potentially reflects some learning-related changes, e.g. delayed onset activity. A comprehensive analysis of these dynamics remains an important direction for future research.

Fig 6-7. Very interesting results - but again, what is the effect of inhibition on frequency of responding?

(Response) We define the lever press frequency per action sequence as the number of lever presses divided by the sequence duration (i.e., $FPS = \# \text{ of lever presses} / \text{sequence duration}$). While this measure provides a straightforward summary of behavior, we found that it does not reliably capture meaningful structural differences between sequences, especially in inhibition experiments. For example, the sequences shown in **Figure R2.a**—such as 4 presses in 1 s, 2 presses in 0.5 s, and 8 presses in 2 s—all yield a frequency of 4 Hz, yet they differ substantially in both duration and total number of presses. These differences are behaviorally important but not detected by frequency alone.

Figure R2. Lever Press Frequency per Action Sequence (FPS) Is Not a Sensitive Measure (a) Lever press distribution patterns that are categorized as 4Hz FPS. (b) Effects of chemogenetic inhibition on sequence distribution during TC-FR4 training. Each dot represents a single sequence trial characterized by sequence duration, FPS, and number of lever presses (2879 trials from control mice, $n = 7$; 2065 trials from hM4Di-expressing mice, $n = 7$). Data were collected over 5 days of TC-FR4 training at an 8 Hz schedule. (c) Number of lever presses per action sequence was significantly altered following hM4Di inhibition (Two-tailed Mann-Whitney test, **** $p < 0.0001$), but FPS was not significantly affected by hM4Di-mediated inhibition (5 sessions from hM4Di mice $n=7$, control mice $n=7$, two-tailed Mann-Whitney test, $p=0.3245$). Each dot denotes session average. (d) Number of lever presses per action sequence (Two-tailed unpaired t-test, *** $p=0.0003$) and FPS (Two-tailed unpaired t-test, $p=0.1365$). Each dot denotes a mouse.

Therefore, in the Figure 4 analysis, we focused on classifying sequences into computationally defined categories (e.g., C1, C2, C3) and labeling them as complete or incomplete. This classification incorporated multiple features, including FPS (lever press frequency per sequence), the number of lever presses per sequence, and sequence duration. While structural differences among sequences are often reflected in FPS, the reverse is not always true. For instance, FPS can remain constant even when both the number of lever presses and sequence duration increase proportionally as shown in **Figure R2.a**.

Thus, even when the structure of lever press sequences changes during inhibition experiments, lever press frequency per sequence (FPS) alone may fail to capture these changes—particularly when both the number of lever presses and the sequence duration increase together (i.e., $\text{FPS} = \# \text{ of lever presses} / \text{sequence duration}$). As shown in **Figure R2.b**, changes in the number of lever presses per sequence were more pronounced than changes in FPS, further illustrating the limitation of FPS as a sole descriptor of sequence structure. This is statistically validated as shown in **Figure R2.c and d**. Due to its mathematical definition, it can overlook meaningful structural changes in behavior.

This limitation is particularly relevant in our inhibition experiments, where both measures tend to increase together. We believe that measures such as lever press count or sequence efficiency—which incorporate both press count and temporal dynamics—provide a more sensitive and accurate assessment of structural effects in these experiments. While we appreciate the reviewer’s point, we are hesitant to include these details in the main manuscript. We hope this explanation clarifies our rationale for introducing ‘sequence efficiency’ and for not using FPS as a primary indicator in other analyses.

Reviewer #2 (Remarks to the Author):

This study by Lee et al. investigates the dynamics and functions of SST interneurons in motor cortex during a freely moving lever-pressing task. They use miniscopes to record Ca²⁺ signals from genetically defined SST neurons longitudinally throughout task acquisition and adaptations. The studies are well controlled, with rigorous analyses. Rigor is also enhanced by the inclusion of analyses of both neurons and mice as samples (ensuring that the reported effects aren't driven by an outlier mouse). The authors make numerous novel findings related to both the dynamics and functions of SST neurons, across a sequence of behavioral challenges. Furthermore, apparent inconsistencies with the related published literature are appropriately contextualized in the discussion. The comments below are included to help strengthen an already strong manuscript.

Major points

1. Perhaps the most important motivating question (as included in the abstract) is “the mechanisms by which cortical microcircuits actively fine-tune the timing and structure of action sequences, enabling organisms to adaptively maintain motor efficiency across varying task demands, remain unclear.” While the manuscript makes multiple important observations towards this goal, the answer is far from complete. To close the gap even further, I recommend the following analyses.

- Fig 1. How might the SST neurons be influencing PYR activity (and vice versa)? From the population activities (Fig 1j,k) is there a phasic relationship between these populations (e.g., SST leading vs lagging)?

(Response) We agree that this is an important question for understanding population-level interactions between two neuronal types. In our observations, SST neurons exhibit relatively synchronized activity at action onset, while PYR neurons display sequential activation patterns. Unfortunately, the substantial temporal variability in PYR neuron activity makes it challenging to define a representative peak location that would allow us to explore the phasic relationship analysis between the two neuronal subtypes (PYR and SST) at the population level.

An ideal approach would be to simultaneously monitor both PYR and SST populations using a two-color imaging technique under specific conditions and analyze spatially segregated groups of cells that interact with each other. However, this is beyond our current technological capacity, and it has yet to be demonstrated in freely moving conditions. A more rigorous strategy would involve in vivo electrophysiological recordings combined with two independent optotagging methods to identify and track cell-type-specific activity with precise temporal resolution. However, optotagging is a low throughput approach, particularly for interneurons that represent a low percentage of the total neuronal population. Thus, while we fully agree this is a valuable direction for future work, a comprehensive investigation of SST–PYR interactions opens up an entirely new line of research. We acknowledge its importance and hope future efforts can explore this compelling line of inquiry.

- The description of SST relationships to ‘rotational dynamics’ and neuronal oscillations is compelling in the FR4 schedule. Obviously, this would be best assessed with simultaneous recordings of PYR neurons. However, perhaps more can be done with the existing dataset. As possibly evident in Fig 3p, were there oscillatory dynamics in the SST population activity? This would have to be assessed either on individual trials (and then average the frequency spectra across trials), or across trials with similar motor sequences (minimal jitter of individual lever presses).

(Response) Referring to Figure 3p, the reviewer asks whether oscillatory dynamics are reflected in SST calcium activity during the sequential lever press task (e.g., TC-FR4). This is a valid question and one we also considered during our analysis. However, the data shown in Fig. 3p may not be ideal for investigating oscillatory dynamics, as it aggregates activity from all different SST neurons. For example, if a few different SST neuronal subsets were activated with some latency during the TC-FR4 task, this integration could produce a similar pattern even without oscillatory dynamics at the individual cell level.

Alternatively, to address this question more thoroughly, we believe it is more appropriate to investigate oscillatory dynamics within an SST subset (e.g. delayed-onset SST in Fig.5) after it has already been classified based on their activation timing. This approach allows us to more directly assess the presence of oscillatory components in SST activity with similar activation patterns, avoiding the dilution of effects from different subsets of SST neurons. To this end, we performed a frequency spectral analysis on either delayed-onset or instant-onset SST INs in relation to sequential lever-press structures, as shown in the figure below (Figure R3).

Following the reviewer’s suggestion, we tested the hypothesis that Short SEQs, which presumably exhibit minimal jitter in both lever-pressing behavior and neural activity, would display stronger oscillatory features compared to Long SEQs. We plotted the frequency spectra of calcium activity profiles for individual trials (Fig. R3 a and b) during the execution of TC-FR4 8Hz and their averaged profiles (Figure R3.c). Indeed, our analysis aligned with the reviewer’s expectation, revealing that Short SEQs exhibit stronger oscillatory components in the frequency spectra compared to Long SEQs. These results also appear to be consistent and reflect the oscillatory components in the calcium activation patterns that are intuitively observable in Extended Figure 5. We greatly appreciate the reviewer’s insightful suggestion and have incorporated this analysis into Extended Figure 5.

Figure R3. Frequency spectra analysis of SST INs during the execution of Long and Short SEQs. (a) Frequency spectra of delayed-onset SST IN activity during Long SEQ execution in the TC-FR4 8Hz schedule ($n = 3$ mice, 333 neurons, 80 randomly selected SEQ). (b) Frequency spectra of calcium activity during Short SEQ execution in the TC-FR4 8Hz schedule ($n = 3$ mice, 333 neurons, 80 randomly selected SEQ trials). (c) Average frequency spectra of calcium activity. Stronger frequency components (between 0.5-1Hz) in the calcium activity were found during the execution of short SEQ. Shaded areas represent SEM. (Two-way ANOVA, Freq effect: **** $p < 0.0001$, SEQ effect: $p = 0.59$, Freq x SEQ effect: * $p = 0.0140$) (d) Average frequency spectra of instant-onset SST IN activity are not significantly different between long and short SEQs.

- Fig 5j-n – for the delayed-onset SST neurons, were there specific phase relationships to individual lever presses? This can be addressed for individual SST neurons and across the population.

(Response) In response to the reviewer’s question, we attempted to align the calcium signals of delayed-onset SST neurons to intermediate lever presses within each sequence. However, lever presses occurring at intervals of 100–500 ms were too rapid to be meaningfully resolved given the temporal resolution of the calcium indicator.

Instead, we aligned delayed-onset SST activity to the final lever press of each sequence (see below, **Extended Fig.9a**), which marks sequence termination and the transition to subsequent behavior. Given that variations in sequence duration were reflected in delayed-onset SST activity (**Figure 5** in the main manuscript), we compared the activity profiles of short vs long sequences time-locked to the final lever press (**Extended Fig.9b**). Prior to this final lever press (time < 0), SST activity displayed distinct ramping patterns that differed depending on sequence duration (**Extended Fig.9b**), suggesting that delayed-onset SST neurons encode structural information about the sequence prior to its termination.

Interestingly, a dominant calcium peak in delayed-onset SST neurons consistently emerged following the final lever press, coinciding with reward port access (**Extended Fig.9c**). Thus, delayed-onset SST activity captures both execution phase of the action sequence—reflected in pre-termination ramping (**Extended Fig.9b and d**) and transition components of action sequences —reflected in the post-press peak (**Extended Fig.9c**). Presence of reward is not distinguishable in the peak amplitude (**Extended Fig.9c and e**). In response to the request, we analyzed the calcium data both at the population average and single cell level.

This activity pattern around sequence termination and transition explains our data in Figure 5 that demonstrates the modulation of delayed-onset SST peak as a function of sequence duration.

(Updated Figure)

Extended Figure 9. Delayed onset SST INs profile reflects ongoing action sequence structure and subsequent transition. (a) Examples of lever press event aligned to the final lever press of each sequence during TC-FR4 4Hz training. (b) Prior the final lever press (time 0), calcium profiles of delayed-onset SST INs exhibit distinct ramping patterns from -2 to 0 seconds between short SEQ and long SEQ, but a calcium peak is observed following the final lever press (c) This calcium peak is associated with reward port access and is not influenced by the presence or absence of reward. (d) Average calcium activity of delayed-onset SST INs prior to the final lever press (ramping region from -2 to 0 seconds) varies significantly with sequence structure (two tailed unpaired t-test, delayed onset SST, n= 68 cells, **p=0.0059) (e) Calcium peak amplitude around reward port access does not significantly differ based on reward outcome (two tailed unpaired t-test, delayed onset SST, n= 68 cells, **p=0.1109)

- Fig 6 – Did SST suppression change the frequency distribution of the lever pressing? Again, analyses of individual trials (and then averaging the frequency spectra across trials) would be needed.

(Response) We thank the reviewer for this interesting comment that prompted a deeper analysis of our findings. As the reviewer probably noted, the frequency (or probability) distribution of lever pressing is already presented in Figures 6e and 6f. Figure 6e illustrates raw lever pressing patterns in individual trials, while Figure 6f shows the averaged spectra of lever pressing frequency (or normalized probability). To capture differences in averaged distribution curves across multiple mice and trials, we converted absolute lever press events into a normalized probability density function.

Thus, existing data in Figure 6 (lever pressing distribution analysis) may directly address the reviewer's question. It demonstrates a significant difference in the lever pressing distribution curve between the hM4Di and control groups.

If the reviewer is instead referring to 'frequency distribution' in the context of lever press frequency within individual sequence trials, we have included an additional analysis in **Figure R2.b**. We did not observe clear differences in the lever press frequency per action sequence bout, defined as [**FPS**= # of lever presses / sequence duration]. **Figure R2.b** reveals that the hM4Di effect on individual sequence trials was discernible on # of lever press /sequence, but not in the frequency (e.g. FPS) of action sequences. This observation was consistent in averaged FPS and averaged # of lever press /sequence as shown in in Figure R2. c and d. Please note that the limitation of the FPS measure was also discussed in response to a comment from the first reviewer.

Figure R2. Lever Press Frequency per Action Sequence (FPS) Is Not a Sensitive Measure (a) Lever press distribution patterns that are categorized as 4Hz FPS. (b) Effects of chemogenetic inhibition on sequence distribution during TC-FR4 training. Each dot represents a single sequence trial characterized by sequence duration, FPS, and number of lever presses (2879 trials from control mice, $n = 7$; 2065 trials from hM4Di-expressing mice, $n = 7$). Data were collected over 5 days of TC-FR4 training at an 8 Hz schedule. (c) Number of lever presses per action sequence was significantly altered following hM4Di inhibition (Two-tailed Mann-Whitney test, **** $p < 0.0001$), but FPS was not significantly affected by hM4Di-mediated inhibition (5 sessions from hM4Di mice $n=7$, control mice $n=7$, two-tailed Mann-Whitney test, $p=0.3245$). Each dot denotes session average. (d) Number of lever presses per action sequence (Two-tailed unpaired t-test, *** $p=0.0003$) and FPS (Two-tailed unpaired t-test, $p=0.1365$). Each dot denotes a mouse.

Since the hM4Di group executed more presses over an extended duration, the FPS remained relatively unchanged. Thus, we believe the key characteristics of lever pressing patterns modulated by SST INs are best captured in the probabilistic distribution curves of lever pressing shown in Figures 6e and 6f. This feature seems to be also reflected in other parameters, as shown in Figures 6b, 6c, and 6d, and is critical for interpreting overall behavioral variations. In response to the reviewer's request, we performed FPS analysis in individual trials (Figure R2b), session averages, and across mice (Figure R2c and d). However, due to the fundamental limitations of FPS in capturing the effects of hM4Di-mediated inhibition, we believe the inhibition effects are more appropriately represented in Figure 6b–f.

2. Early in the study the authors make important observations regarding differences between L2/3 vs L5 SST neurons. Were such laminar analyses performed in the later studies? For the FR4 studies of dissociable population dynamics, are both populations distributed in superficial and deep layers?

(Response) This is an important question for understanding neural representation across cortical layers. As L5/6 SST neurons exhibited the most action-related calcium responses in Figure 1, our study primarily focused on monitoring L5/6 SST neurons in the subsequent calcium imaging experiments. Notably, the dissociable population dynamics (instant and delayed SST responses) were observed within the L5/6 SST neurons rather than being distinct properties across different layers. While our study does not provide an extensive functional comparison between L2/3 and L5/6 SST neurons, we agree that such a

comparison is a valuable direction for future research based on our findings. To clarify this information, we have added the following sentences:

(Revised manuscript)

Line 495: ~~In contrast,~~ Another activity pattern of SST INs is characterized by a delayed calcium response following the onset of action sequences. This response was specifically pronounced in the 4Hz or 8Hz schedules compared to 1Hz training schedule (**Extended Fig.5 i,j**). These two distinct population dynamics - ‘instant’ and ‘delayed’ onset - were not layer-dependent; rather, both types of activity were observed within L5/6 SST neurons. We observed that inhibition of SST activity (including inhibition on a trial-by-trial basis) prolonged the temporal distribution of lever press probability (Fig.6f and Fig.7f),

3. In the FR1 D7 SST data, there appears to be two peaks in the average data. Are these peaks from separable populations, as described in the FR4 schedule data?

(Response) Thank you for raising this important point. Our existing data may help address this question. As shown in Figure 1o, two peaks are observed in the same individual cells during FR1 D7 period. As animals learn complex action sequences and adapt their behavior patterns, SST neural representation evolves. After extended training, when animals reliably perform TC-FR4 tasks under the 4Hz and 8Hz schedules (Figure 5), two distinct populations with different peak locations become discernible.

4. The authors describe in the Discussion that their studies cannot rule out changes in more subtle forelimb kinematics during SST suppression. However, the same limitation applies to the observations in Fig 2d vs 2e. Changes in the velocity or trajectory of limb movements may explain the differences in SST activity. If the authors have the body/limb movement data, I encourage them to present the data to address this potential confound. If not, then mentioning this in the discussion would be important.

(Response) As the reviewer noted, limb movement velocity and trajectory are important motor measures and could be related to SST activity. However, the operant training paradigm used in our study—as well as in prior work (Kawai et al., Neuron, 2015)—simplifies these kinematic aspects of motor behavior in order to focus on the role of cortical circuits in holistic action sequence control. We agree that incorporating detailed limb and body movement data would ultimately provide a more comprehensive understanding of the neural mechanisms underlying action sequence regulation. At the moment, we don’t have data to fully address this point. We acknowledge this as a relevant discussion point and have updated our discussion as follows:

(Revised manuscript)

Line 411: Thus, our motor metrics might not fully capture subtle variations in stereotyped forelimb kinematics (some of which may show inter-individual variability), but they do capture the organized action patterns needed to successfully complete the task regardless of these variations. Future studies incorporating detailed analyses of forelimb or whole-body kinematics would provide a more

comprehensive understanding of the underlying motor control mechanisms. In addition, the role of the motor cortex and SST IN activity in isolated forelimb control in head-fixed tasks may differ fundamentally from their role in our freely moving conditions, as the necessity of cortical activity and its correlation with behavior are highly task-specific

Minor points

1. Fig 2o – expand y-axis to better visualize data

(Response) Thank you for pointing this out. We adjusted the y-axis of the figure.

(Updated Figure)

2. Extended Fig 6c, right- expand y-axis to better visualize data

(Response) Thank you for pointing this out. We updated the y-axis accordingly.

(Updated Figure)

3. Line 311 - Extended Fig 7b?

(Response) We have corrected the typo in Line 331 in the revised manuscript, Extended Fig 7b is correct. Thank you for pointing this out.

4. Please clarify: “Although we also observed slight differences in initial motor task learning between tethered and untethered mice, we did not detect significant circuit inhibition effects between experimental and control groups (Fig.6b and Fig.7b).”

(Response) Thank you for seeking clarification. Although it was a relatively minor discussion, we agree that further clarification would be helpful. While addressing other reviewer’s comments, we performed new experiments (please find **Figure R5**) to describe the tethering effect. As this point is related to a technical aspect of experimental control, we moved the information to the Methods section:

(Updated Manuscript)

~~Although we also observed slight differences in initial motor task learning between tethered and untethered mice, we did not detect significant circuit inhibition effects between experimental and control groups (Fig.6b and Fig.7b).~~

(Updated Manuscript: Methods)

Line 719: The lever press rate during TC-FR4 1 Hz and 2 Hz training in tethered mice (**Fig. 7b**) was lower (~10 presses/min) than that observed in the freely moving condition (~20 presses/min, **Fig. 6b**). Nevertheless, this difference did not affect the overall consistency of behavioral results and their interpretation in the context of our experiments.

Reviewer #3 (Remarks to the Author):

In this manuscript, the authors investigate the role of SST interneurons in learned motor behavior by measuring and manipulating neural activity in mice. Mice were trained to perform lever-pressing tasks of varying difficulty and speed to earn rewards. The authors observed task-related activity modulation in both pyramidal neurons and SST interneurons, with higher SST activity in deeper cortical layers. SST activity was elevated during early task learning and varied with movement type across tasks. SST activity appears elevated during early task learning and varies with movement form across tasks. SST cell activity is varied in its relation toward movement, with a preponderance of cells active at movement initiation. Both chemogenetic and optogenetic perturbations of SST neuron activity lead to performance deficits specifically during the most challenging versions of the task.

The results overall are novel, interesting, and have relevance to our understanding of M1 function during both motor learning and execution. Below I describe a number of concerns that should be addressed during revision.

Support for the specificity of SST cell role

The manuscript focuses on SST neurons, and implies that SST neurons play a specific role in executing more challenging lever press tasks. For example, on line 283 they refer to “a direct and specific contribution of action related SST IN activity,” and on line 434, they refer to “insights into the specific functionality of M1 cortical INs.” While it is certainly appropriate to focus on the role of one cell type, the data included in the manuscript do not make a case for a specific role of SST cells, as distinct from that of other cell types. No comparison is performed to other interneuron types, like PV cells for example, and no differentiation from the overall inhibitory interneuron population is demonstrated. These would involve separate experiments using PV-cre or Vgat-cre mice. I am not suggesting that the authors must do such experiments, but instead that they should avoid suggesting that their data supports a specific role for SST interneurons.

(Response) Thank you for the thoughtful comment. Our intention was not to highlight the functional specificity of SST compared PV or VIP interneurons. Rather, we aimed to convey that the calcium activity of SST INs displays a degree of specificity in relation to the structural features of the lever press sequences. To clarify this point and prevent misinterpretation, we have revised the manuscript by replacing the term "specific" or “contribution” or “functionality” with language that more accurately reflects the intended scope of our view.

(Revised Manuscript: Line 300), These data indicate a direct ~~and specific contribution of action-related~~ role of SST IN activity in the organization and execution of efficient operant action sequences.

(Revised Manuscript: Line 483), Therefore, determining whether the neural activity patterns in cortical microcircuit INs in M1 resemble those from sensory inputs or motor outputs ~~could provide valuable insights into the specific functionality of M1 cortical INs~~ is crucial in understanding functional role of M1. Our study specifically focused on monitoring SST IN activity to determine what aspects of action sequence-related information are encoded.

Along these lines, it is questionable whether the specific perturbation effects reported in Figures 6 and 7 result from perturbing SST cells in particular or from the general disruption this causes to M1 activity. The observation from these experiments, of a task-specific effect of SST cell activity perturbations, could reflect either a specific role for SST cells or a role for M1 in general. Moreover, much existing data suggests that the influence of M1 on movement is specific to tasks that challenge dexterity. For example, mammals retain the capacity to execute many motor behaviors after M1 lesions, but struggle with tasks that are particularly dexterous. Based on these previous results, one might expect that perturbations to M1 activity in general would show similar effects as those reported in Figures 6 and 7. Again, I am not necessarily asking for more experiments here, but this alternative interpretation should be highlighted as well.

(Response) The reviewer’s hypothesis suggests that the behavioral effects of SST IN inhibition (shown in Figures 6 and 7) could also result from M1 lesions or general perturbation of M1 activity. We agree this is a reasonable possibility, given M1’s well-established role in motor learning and dexterity.

However, extending our interpretation in this direction would require additional experimental evidence to bridge key conceptual gaps. For instant, our study does not directly address the relationship between SST interneuron and pyramidal (PYR) neuron activity in M1. Thus, it remains unclear how SST INs influence broader M1 network dynamics, and conversely how they are modulated by M1 PYR neuron activity, or how M1 PYR activity correlates with our behavioral tasks. Ideally, addressing these questions would be essential to support the suggested hypothesis.

That said, we remain cautious in speculating about how our observed behavioral effects could be replicated by perturbing generalized M1 activity. We agree with the reviewer's point that broader M1 circuitry likely serves as the principal neural substrate for motor control in our task. Within this framework, SST INs may function as modulatory components embedded in the M1 network. This integrative view helps reconcile our interpretation with the reviewer's broader framework -consistent with prior studies- highlighting M1's central role in motor tasks that require precise kinematics, dexterity, or adaptive control. We have revised our discussion as follows to incorporate this perspective and better contextualize our findings within M1's broader role in motor behavior.

(Revised Manuscript line 537) Overall, our findings implicate SST INs in M1 as fundamental upstream regulators for the efficient execution of complex action sequences. Beyond their crucial role in facilitating plasticity in PYR neurons during motor learning [27, 29], the dynamic activity of M1 SST INs may directly influence the structural regulation of trial-by-trial action sequences. More specifically, it is plausible that the net output of M1 circuitry contributes to the behavioral modulation observed in our experimental results (Figures 6 and 7), while SST INs serve as modulatory elements within M1 circuits. Completion of TC-FR4 task at 4Hz and 8Hz schedules requires substantial motor control. Within this framework, our data This suggest that a key function of M1 SST INs may be to "fine-tune" complex motor programs.

Key features of cortical involvement in movement execution

The manuscript could do a better job of reviewing current understanding of M1 in movement execution, and how it relates to the previous observations of cortical disengagement across learning and the observations reported in the present manuscript. The authors here do note that previous observations of cortical disengagement have come after experimental animals have completed very large numbers of trials over many weeks and have become highly stereotyped in their execution. Also important is the fact that these observations come from tasks that do not have a particularly strong requirement for dexterity. In fact, in Kawai et al., *Neuron*, 2015, the authors state that their task was specifically designed to avoid dexterity, as their central goal was to distinguish the role of M1 in motor learning from its role in executing dexterous movements, which had not been experimentally decoupled in previous studies. A very large amount of previous literature, ranging from older lesion studies (e.g. Lashley, *Arch Neuro Psychiatr*, 1924 or Travis, *Brain*, 1955), to more recent studies using optogenetics (e.g. Guo et al., *eLife*, 2015 or Miri et al., *Neuron*, 2017) suggest that M1 plays a specific role in the execution of dexterous motor behaviors. Other evidence has implicated M1 specifically in adaptive or corrective movements, when stereotyped movement trajectories need to be deviated from (e.g. Drew et al., *Can J Physiol Pharmacol*, 1996, Warren et al., *eLife*, 2021, or Bollu et al., *J Neuropys*, 2024). These previous results help explain why the role of M1 or M1 SST cells in particular, may be specific to the more challenging lever press task the authors used in their manuscript.

This could also help explain the divergence between results with and without head-fixation: lever press behaviors may become more challenging for mice to perform under head fixation, thereby engaging motor cortex in tasks that might otherwise not involve M1 if performed in a more natural, unconstrained setting. This is important to note, given the prevalence of head-fixed behavioral paradigms in motor system studies.

(Response to Literature Summary and Discrepancy between Head-fixed and Freely Moving Conditions)

The reviewer makes an important point in suggested providing a more detailed summary of prior studies involving M1 lesions, optogenetic manipulations, and diverse behavioral paradigms. While several points in the original text (e.g., Lines 393, 400 and 403 in the original manuscript) are already consistent with the reviewer's interpretation,

our discussion of previous studies was admittedly relatively brief. We agree that a more thorough review of the literature would offer readers a broader and more integrated perspective on this topic. Also, the reviewer raised an important point in interpreting the experimental differences between head-fixed conditions and our freely moving conditions.

In summary, our experimental results suggest that M1 SST interneurons may engage differently depending on task complexity and the degree of motor adaptation required. This hypothesis helps explain the discrepancies observed between head-fixed paradigms in earlier studies and our findings in freely moving conditions. It also aligns with our observations of task-dependent neural inhibition during TC-FR4 training. We appreciate that the reviewer shares a similar perspective and interprets our findings as reflecting broader features of general M1 circuitry in motor behavior.

In response to the reviewer's suggestion, we have expanded the discussion to include these additional interpretations and have incorporated the recommended references to support and strengthen our argument.

(Updated Manuscript)

(Line 430) Our training criteria may be significantly 'simpler' than motor tasks in other studies, where motor cortex lesions immediately impaired performance and learning, yet 20,000 to 30,000 training trials were necessary to achieve stereotyped forelimb movement patterns [2,15]. We speculate that the motor tasks involving such stereotyped forelimb tasks may be more "complex" in terms of skill learning, requiring greater cortical involvement. As a result, the motor impairment might become apparent from the beginning of training when using inhibition approaches in the task that requires highly reproducible movements.

For example, learning stereotyped forelimb joystick control [12,24,36] or precise reaching tasks [32,33], under head-fixed conditions likely demands not only dexterous movement control but also significant motor adaptation to compensate for unnatural biomechanical constraints. These motor performances are typically sensitive to cortical perturbations, consistent with a view that M1 is crucial for dexterous motor behaviors and for adaptive or corrective movements when stereotyped trajectories need to be altered [32, 34, 35, 36]. In contrast, more naturalistic movements, such as walking, rely less on M1 [37,38]. The functional role of M1 likely differs depending on the motor task, particularly whether it involves dexterous movements.

In some experimental designs using freely moving animals, it is possible to decouple the precise kinematics or dexterous movements from task completion. Recent studies demonstrate that M1 is not necessary for performing fully automatized, stereotyped lever-press sequences [2] but remains essential for sequences requiring flexibility and external cue guidance [20]. These findings indicate that M1 engagement varies depending on task demands, even when specific dexterity or precise kinematics of movement are not necessarily required. Similarly, our experimental design employs freely moving mice performing various sequential lever-press tasks that are learned within 100-1000 of training trials. Based on previous studies showing that M1 exhibits two distinct engagement dynamics depending on behavioral context [2,20], we hypothesized that local microcircuits within M1, including SST INs, also exhibit task-dependent activity patterns. To test this, we varied the training schedules to progressively increase the demands on motor performance required to complete the action sequences and examined how SST interneurons contributed to shaping their temporal structure.

Using cell-specific calcium imaging, we observed distinct activity patterns of M1 SST INs between (1) the simpler FR1 task—which allows for greater motor variability and more naturalistic movements to complete the task—and (2) the more demanding TC-FR4 SEQ task, which imposes tighter temporal constraints and requires an unnatural, structured lever-press sequence that is more difficult to learn. This distinction is further supported by our behavioral findings, which show that inhibition of M1 SST INs selectively impaired performance on the more demanding TC-FR4 SEQ tasks at 4Hz and 8Hz but had no effect on simpler tasks such as FR1, FR4, or the TC-FR4 SEQ tasks at 1Hz and 2Hz schedules. Overall, it is possible that M1 SST IN activity partially reflects broader M1 circuitry, exhibiting task-dependent engagement characteristics. However, it remains an open question whether broader M1 dynamics shape SST IN activity or, conversely, whether task-dependent M1 SST IN activity directly influences broader M1 circuitry during our behavioral paradigms.

(Response to Cortical Disengagement Argument in Previous Studies and Our Results)

Additionally, the reviewer asked to add more discussion about the cortical disengagement concept while comparing our experimental data and previous literature. In the original manuscript, we summarized prior studies discussing M1 lesion effects and models of M1 disengagement (Line 338 in the original manuscript; see excerpt below), and we attempted to interpret our findings within this framework (Line 354 in the original manuscript, Line 374 in the revised manuscript; also shown below).

Fundamentally, we proposed a similar idea to the reviewer—that changes in M1 SST interneuron activity may reflect broader characteristics of cortical engagement and disengagement within the M1 circuitry. While we intended to convey this view in the original manuscript, particularly in our discussion of cortical disengagement (Lines 338 and 354 in the original manuscript), we recognize that our perspective may not have been fully articulated. We agree that making clearer comparisons between prior M1 studies and our findings would help clarify and enrich the narrative. In response, we have revised the manuscript (**Line 374 in the revised manuscript**) as follows:

(Manuscript)

Line 338 in the original manuscript: These data align with the idea that neural activity in M1 persistently represents action performance and kinematics even after skill learning, although the M1 contribution may undergo some refinement [7,8,10,11]. On the contrary, the cortical disengagement model proposes that M1 activity becomes less critical [14,31,34] or decorrelated from action execution upon consolidation of a given motor program [12]. This model implies that cortical output is less important for movement control as proficiency is gained.

Line 374 in the revised manuscript: In the simple FR1 task schedule, our finding of diminished SST IN activity following extended training (Fig.2) is consistent with the cortical disengagement model [12, 13, 31]. For instance, previous studies have shown a decline in the correlation of neural activity in M1 PYR neurons during repetitive motor training [16]. Similarly, we observed decreased M1 SST IN activity and correlation during the execution of the well-learned FR1 task. These changes in SST IN activity may reflect the broader disengagement feature of M1.

Concerns about the damage to M1

It appears that the bulk of the paper reports activity measurements from L5/6 SST cells imaged through cylindrical GRIN lens inserted into M1. This lens is 0.5 mm in diameter and was inserted to a depth of 0.5 mm into cortex. This volume of tissue disturbed by the implantation is large relative to the overall volume of forelimb M1 in mice. In addition to destroying upper layer circuits that send substantial input to deeper layers (layer 5A begins less than 0.5 mm from the pial surface), this will also disrupt the apical dendrites of deeper layer cells. Moreover, the authors do mention a slower learning in tethered animals, which might reflect damage to M1. Could the increased activation of SST cells in L5/6 activity measurements reflect a compensation for the loss of inhibitory interneurons superficially? It would be reassuring to see results confirmed using a triangular GRIN lens.

(Response to cortical damage and behavior)

Within our surgical setup, we performed two distinct surgical approaches: (1) GRIN lens implantation targeting layer 5/6 for TC-FR4 training and imaging, and (2) layer 2/3 imaging by positioning the GRIN lens on top of the superficial cortex to preserve M1 integrity. We did not observe impairments in learning or performance differences between animals with L5/6 GRIN lens implantation and those imaged in layer 2/3 without cortical damage. However, we appreciate the reviewer's concern and will explicitly acknowledge this technical limitation. The disruption of layer 2/3 caused by GRIN lens implantation may have some compounding effects.

(Response to cortical imaging without damaging L2/3)

The reviewer's concern is that the observed neural activity in L5/6 may reflect surgical effects on L2/3 rather than experimental manipulations. While it is true that GRIN lens implantation can cause superficial cortical disruption, this limitation is not unique to our approach. In fact, many widely used techniques—such as fiber photometry, optogenetics, and even viral injections—also involve some degree of invasive procedures that inherently risk tissue damage. Recognizing this, numerous previous studies practically address this limitation by comparing groups or conditions that have undergone identical surgical procedures.

Although prism lenses are often considered to better preserve columnar architecture, their insertion still requires penetrating M1 or adjacent cortical areas (e.g., M2 or S1) across the full depth—from superficial to deep layers. This can disrupt intercortical communication and potentially induce compensatory changes in neural activity. Therefore, while alternative imaging approaches may offer some advantages, it is very hard to fundamentally eliminate the risk of interpretational confounds associated with tissue damage as long as any surgical intervention is involved.

While GRIN lens implantation has inherent limitations, our study design remains appropriate for generating meaningful insights. We structured our experiments to compare groups and conditions that underwent identical surgical procedures. Importantly, unlike many previous GRIN lens studies that examine only a single behavioral condition, we compare neural activity across distinct training paradigms (e.g., extended FR1 vs. TC-FR4). If the enhanced neural activity observed during TC-FR4 were merely a result of the surgical procedure, we would expect to see similar increases during extended FR1 training—which we did not observe.

In addition, we conducted within-subject analyses (Figure 2 g,m,o, Figure 3l,r,t, Figure 4 and Figure 5), revealing consistent changes in SST IN activity that closely tracked temporal patterns of sequential lever press. These findings support the interpretation that the observed neural dynamics reflect behaviorally driven modulation rather than artifacts of the surgical procedure.

In sum, we agree that surgical disruption is a valid concern. Nevertheless, our experimental design—particularly the inclusion of various behavior schedules and within-subject comparisons—helps mitigate this limitation and reasonably supports our conclusion. Taken together, the data suggest that the observed neural dynamics are more likely driven by task-specific factors than by surgical artifacts.

(Response to slow learning in tethered mice)

We noted in Line 407 (original manuscript) that “we also observed slight differences in initial motor task learning between tethered and untethered mice (Fig.6b and Fig.7b).” To assess whether tethering condition influenced motor performance, we conducted new experiments that replicate the situation in Figure 7b. All mice first underwent virus injection and bilateral fiber implant surgery targeting M1. No laser light was delivered during these tests, ensuring that optogenetic manipulation on circuits did not influence behavior or learning. After mice were trained to perform the TC-FR4 1Hz task under tethered conditions, they were divided into two groups to learn TC-FR4 2Hz task: one group continued training under tethered conditions from Day 1 to Day 5 (control group; filled circles in **Figure R5a**), while the other group (test group; open circles) was tested under untethered conditions only on the final day (D5). Test group mice examined under the untethered condition on D5 exhibited a significant increase in lever press rate compared to their performance on D4, indicating that the tethering condition had been attenuating motor performance in this behavioral paradigm. The change from untethered baseline(D4) to test day (D5) significantly differ between tethered group and control group.

Figure R5. Tethering impairs TC-FR4 performance compared to untethered condition (a) Group comparison: Both the test and control groups were trained under tethered conditions from Day 1 to Day 4. On Day 4, only the test group performed the TC-FR4 2 Hz task under the untethered condition (Two-way ANOVA, day effect: *p=0.0287; day x group effect: *p=0.0274; Sidak’s multiple comparisons test: test group D4 vs D5 **p=0.0083 n = 7mice, control group D4 vs D5: p=0.999, n = 7mice). **(b)** Within-subject comparison: All mice were tested under both untethered

and tethered conditions on the same day, each for 15 minutes, in two separate training sessions (two tailed paired t-test, *** p=0.0006). Each dot denotes one mouse.

To further determine whether the effect of tethering was immediate and observable within individual subjects, we also tested each mouse under both tethered and untethered conditions within a single day (**Figure R5b**). These within-subject comparisons revealed that the same mice consistently showed higher lever press rates when untethered. Together, these findings demonstrate that tethering has a measurable and immediate attenuating effect on motor performance and likely contributed to the differences observed in Figure 7b. In Figure R5, the increased lever press performance (~20 presses /min) of test group mice on D5 is comparable to the performance of freely moving mice without tissue damage from implant surgery (Figure 6b in main manuscript). Thus, it is most likely tethering, and not implantation per se, that underlies the slower learning/performance observed in the early phase of TC-FR4 training.

We originally mentioned tethering effect in the main manuscript, but we added the following information in the methods:

(Updated Manuscript: Methods)

Line 719: The lever press rate during TC-FR4 1 Hz and 2 Hz training in tethered mice (**Fig. 7b**) was lower (~10 presses/min) than that observed in the freely moving condition (~20 presses/min, **Fig. 6b**). Nevertheless, this difference did not affect the overall consistency of behavioral results and their interpretation in the context of our experiments.

Lack of evidence for discretely separable activity patterns

The analysis of activity patterns in Figures 4 and 5 is described as implying that there are distinct types of SST cells - separate classes or clusters. For example, on line 231, the authors refer to “mutually exclusive neuronal subpopulations.” Instead, the data presented here are consistent with a continuous distribution of response types. This distribution is important to quantify and flesh out, but the data do not support the existence of discretely separable types. Seeing differences across the averages of clusters, which come from algorithms fated to separate populations into groups, does not support the idea that discrete or substantially distinct clusters exist. If you divide a continuous distribution into 2 halves, the means of each half will differ from one another, even though they each derive from a single underlying distribution. The manuscript does not provide evidence that the differences across neurons are beyond those expected by chance for a single distribution. Moreover, the plot in Figure 1f shows the distribution of response types across cells, which looks continuous. Again, the quantification of response diversity here is important, but the interpretation that discretely different response types exist should be avoided.

(Response to bimodal distribution of calcium responses)

Thank you for pointing an important aspect of cell activity classification. To examine whether SST INs exhibit a continuous or bimodal distribution of calcium responses, we displayed calcium peak timing of all identified SST INs without pre-classifying cell types (**Figure R6**). For each training session during extended TC-FR4 training (4 Hz, 8 Hz), we computed the average calcium response of each neuron and extracted the timing of its maximum peak. The resulting distribution (**Figure R6**) reveals a clear bimodal pattern, indicating that some cells are preferentially activated at sequence initiation, while others respond several seconds later.

Figure R6. Bimodal distribution of SST IN calcium events during TC-FR4SEQ training at 4 Hz and 8 Hz schedules. A total of 712 calcium profiles were identified as session averages. Within each profile, representative peak was detected and quantified as calcium events to generate the histogram.

Missing statistical tests for key comparisons (FIG1)

In several cases throughout the manuscript, the authors refer to differences between experimental groups without reference to appropriate statistical tests in either the text or figure legends. For example, the difference between plots in Figure 1 n and q is highlighted, but these actually look somewhat similar and no statistics are included to support a significant difference. Similarly, the description of results in Figure 5j described in lines 245-7 does not provide statistics, nor does the figure legend.

(Response) We have included new statistical data in Extended Figure 1 to support the observed differences.

Extended Figure 1 (c) Average $\Delta F/F_0$ responses of individual PYR neurons and SST INs in L2/3 and L5/6 aligned to lever press (Kruskal–Wallis test, main effect $p < 0.0001$). (d) Population-averaged $\Delta F/F_0$ responses across individual mice (one-way ANOVA, main effect $p < 0.0001$). (e) Cell-to-cell correlation coefficients of PYR neurons in L2/3 and L5/6, each dot denotes an animal measurement on D7 (two-tailed unpaired t-test, $*p=0.0357$) (f) Cell-to-cell correlation coefficient of SST INs in L2/3 and L5/6, each dot denotes an animal measurement on D7 (two-tailed unpaired t-test, $*p=0.0281$)

Time course of perturbation effects across learning

The increased magnitude of the perturbation effects later in training reported in Figures 6 and 7 is a very interesting finding. However, because of the experimental design, one possibility is that the effects are more severe at later

time points because they reflect the accumulated effects of repeated activity perturbation. A control to exclude this possibility should be provided. For example, performing perturbations in a set of animals only after they have reached the 8 Hz training stage.

(Response) We thank the reviewer for raising important point related to the role of past versus ongoing neuronal activity. We conducted new experiments to address this comment. We prepared mice (eNpHR n=8 and eYFP n=6) under the same surgical procedures as Figure 7. These mice were trained on the TC-FR4 task without any optical manipulation until they reached proficiency at the 1Hz, 2Hz, and 4Hz schedules. The optogenetic inhibition of M1 SST interneurons was applied only during the 8Hz schedule. Thus, this experiment tests whether the previous chronic opto-inhibition treatment was critical and necessary to induce the significant behavioral difference during the 8Hz schedule in Figure 7.

The result was interesting. In the absence of prior opto-inhibition (**Figure R7**), we found no significant difference in lever presses per minute between the eNpHR and eYFP groups during the 8Hz schedule. This suggests that the behavioral divergence observed in Figure 7 reflects the cumulative effects of opto-inhibition throughout the training period. In contrast, once mice had already acquired proficiency in performing the TC-FR4 sequence task (**Figure R7.a**), acute opto-inhibition alone did not produce a discernible behavioral difference on lever press rate within our training paradigm (**Figure R7.b**). However, these new experiments revealed that the event distribution—where each data point indicates a lever press—coupled with respective probability histograms of lever press was significantly different between the opto-inhibited and control eYFP groups (**Figure R7.c**). This data further clarifies the contribution of SST activity in action sequence execution, and support roles of both cumulative and online activity of SST INs in efficient high frequency lever pressing.

Figure R7. Overall lever press rate during TC-FR4 training was not significantly different by late-phase selective optoinhibition, but differences emerged in the lever-press event distribution. (a) Diagram of overall and within-session late-phase opto-inhibition paradigm (b) Average lever presses per minute between groups in late-phase opto-inhibition (two-way ANOVA. Group $p > 0.050$); (c) Event distribution plots of lever pressing structure between groups where each data point indicates a lever press coupled with respective probability histograms of lever press occurrence after sequence initiation (Kolmogorov-Smirnov unpaired t-test, $p = 0.0032$)

Interpreting the activity changes across FR1 learning

The changes in SST activity between early and late in learning the FR1 task (Figure 2) are striking and are potentially consistent with a specific role of the SST neurons during acquisition but not later execution. However, the revelation later in the paper that SST activity perturbations early in learning have little effect argues against this possibility. It is not clear how to reconcile these two results, and ultimately what to make of the changes seen in Figure 2. The Discussion should address this.

(Response) Thank you for your comments on important aspects in our experimental results and interpretation. We agree that the change in SST activity during extended FR1 learning is intriguing, especially given the absence of

behavioral difference following SST inhibition. We discussed this discrepancy in the manuscript (line 413 “To understand this discrepancy, it is ...”) because previous studies investigating head-fixed mice performing stereotyped forelimb tasks demonstrated motor skill impairment from the beginning with optogenetic manipulations.

SST activity may have a function, but inhibition effect might be not captured by our behavior metrics:

One possibility is that SST activity may be specifically important for establishing stereotyped movements, as suggested by previous head-fixed studies (Chen. S.X. Nature Neuroscience 2015), whereas completion of the FR1 task in freely moving conditions does not critically rely on such movement patterns. Mice are often able to perform the FR1 task even using highly variable, naturalistic movements, especially during the early training phase. If SST activity contributes to the establishment of stereotyped movement patterns or the fine-tuning of general motor output, the motor assessment metrics used in the FR1 task may not be sensitive enough to detect subtle effects of SST-specific perturbations. To address the reviewer’s point, we have incorporated the following perspective into the Discussion section.

(Updated manuscript)

Line 415: Previous studies using head-fixed mice implemented a constant training schedule to establish stereotyped forelimb movements, focusing on forelimb trajectory or **motor parameters (e.g. speed or acceleration)** as indicators of motor learning [12,14,16,22,24]. In contrast, by analyzing lever press patterns on a trial-by-trial basis in our study, we assessed action performance and efficiency as opposed to **motor parameters reflecting isolated forelimb movements**. Thus, our motor metrics might not fully capture subtle variations in stereotyped forelimb **movements** (some of which may show inter-individual variability), but they do capture the organized action patterns needed to successfully complete the task regardless of these variations. **Future studies incorporating detailed analyses of forelimb or whole-body kinematics would provide a more comprehensive understanding of the underlying motor control mechanisms.**

SST INs might have fundamentally different neural mechanisms in two tasks:

Essentially, the neural dynamics of SST INs during extended FR1 training and TC-FR4 might be fundamentally different. In the FR1 task, particularly under freely moving conditions, SST inhibition did not lead to marked behavioral deficits—possibly because the task could be solved using variable, naturalistic movements that did not require highly precise cortical modulation. In contrast, both chemogenetic and optogenetic inhibition of SST neurons significantly disrupted the temporal structure and efficiency of action sequences during the more demanding 4 Hz and 8 Hz TC-FR4 training schedules (Fig. 6d and Fig. 7d). Even under freely moving conditions, these schedules impose substantial biomechanical and timing constraints that likely recruit cortical inhibitory control to a greater extent. Similarly, head-fixed motor tasks requiring repeated, stereotyped forelimb movements in a constrained kinematic space are also sensitive to SST IN manipulation. In contrast, tasks such as treadmill walking or FR1 operant schedules, which permit higher motor variability and are easier to learn, were largely unaffected by SST inhibition. Together, these findings and our interpretation suggest that the behavioral impact of SST inhibition is more apparent under tasks that demand challenging temporal criteria or motor coordination. In the revised manuscript, we have added the following sentences:

(Updated manuscript)

Line 453: Using cell-specific calcium imaging, we observed distinct activity patterns of M1 SST INs between (1) the simpler FR1 task—which allows for greater motor variability and more naturalistic movements to complete the task—and (2) the more demanding TC-FR4 SEQ task, which imposes tighter temporal constraints and requires an unnatural, structured lever-press sequence that is more difficult to learn. This distinction is further supported by our behavioral findings, which show that inhibition of M1 SST INs selectively impaired performance on the more demanding TC-FR4 SEQ tasks at 4Hz and 8Hz but had no effect on simpler tasks such as FR1, FR4, or the TC-FR4 SEQ tasks at 1Hz and 2Hz schedules. Overall, it is possible that M1 SST IN activity partially reflects broader M1 circuitry, exhibiting task-dependent engagement characteristics.

The sequence efficiency metric should be better explained

It seems like the sequence efficiency metric penalizes pressing too many times (more than 4) or too quickly (more than 4 Hz), but were the mice penalized in terms of reward when they overshoot? Also, how much the authors decided to penalize overshooting is unclear. Why not use a function where there is no penalty for pressing more or pressing faster? I suppose pressing more could result in longer trials leading to less rewards/min, and faster presses might be less efficient in terms of energy expenditure, but none of this is clearly explained. In addition, the authors do not describe how they arrived at the amount of skew that they used for the function. This would directly affect analyses like those shown in Figure 6 where inhibiting SST cells resulted in decreased efficiency leading them to claim that “chemogenetic-mediated hyperpolarization of M1 SST interneurons deteriorates sequence organization” (line 274).

(Response) Thank you for this critical comment. We find this to be a constructive and valuable argument, and we recognize that some important points were not sufficiently conveyed in our manuscript.

The reviewer’s first concern can be summarized as follows: if overshooting in lever pressing could result in earning more rewards, then penalizing overshooting in the efficiency metric may be inappropriate. We understand the reasoning behind this argument. In our motor training protocol, it is true that mice were not penalized in terms of reward delivery for performing additional lever presses or pressing at excessively high rates. Therefore, the reviewer raises the possibility that overshooting could be advantageous if it indeed leads to increased total reward intake, and in such a case, it should not be discounted in the efficiency function.

To address this concern, we examined whether a higher number of lever presses per session (**Figure 6b**) or per sequence (**Figure 6c, e, f**) was associated with greater reward acquisition. The answer is **no**. As shown in our new analysis (**Figure R8, see below**), the hM4Di group and the control group did not differ significantly in the reward earning rate, despite the hM4Di group exhibiting markedly altered lever-pressing patterns (**Figure 6c, e, f**). In other words, hM4Di inhibition led to an increase in lever presses (unnecessary lever presses) without a corresponding increase in reward acquisition. This could reflect a less efficient motor strategy, when considering the reviewer’s perspective.

Figure R8. Reward earning rate is not significantly different with hM4Di inhibition (a) Reward per minute in hM4Di and control groups (two-way repeated measures ANOVA, time effect : **** $p < 0.0001$, treatment effect : $p = 0.9327$, time x treatment : $p = 0.5632$) **(b)** Total rewards per session, maximum number is 20 (two-way repeated measures ANOVA, time effect : **** $p < 0.0001$, treatment effect : $p = 0.7805$, time x treatment : $p = 0.8546$)

The reviewer’s point highlights that reward acquisition or reward rate could be an important additional factor in defining behavioral efficiency. However, our original definition of motor efficiency did not incorporate reward rate into the coefficient calculation. Instead, the metric was designed to capture efficiency from the perspective of energy consumption during a single motor sequence trial. Specifically, any lever pressing beyond the sequence criterion, or pressing at rates exceeding 4 Hz, was considered unnecessary muscle activity and energy expenditure rather than an optimal strategy. This is supported by the observation that the control group consistently generated approximately 4–5 lever presses per sequence trial (Figure 6c). Our sequence efficiency function was designed to incentivize this pattern of optimal sequence generation observed in wild-type mice.

In the discussion, we clarified that, although one might expect overshooting to improve performance by increasing rewards, our data show that the hM4Di group's behavior reflects excessive, unproductive lever pressing without a corresponding reward gain. We also explicitly state that our efficiency definition emphasizes movement optimization (Δ = excessive lever presses – optimal lever presses) and energy use, and does not directly incorporate reward rate into the calculation.

(Updated Manuscript)

Line 519: In the cell-specific inhibition experiments (**Figures 6 and 7**), we assessed the effects of SST interneuron inhibition on sequence organization by defining sequence (SEQ) efficiency. Although increased lever pressing in the hM4Di group (**Figure 6b**) might be expected to enhance reward earning (e.g., higher reward rate), we found no significant difference in reward rate between hM4Di and control groups across all training schedules (two-way repeated measures ANOVA: time effect **** $p < 0.0001$; treatment effect $p = 0.9327$; time \times treatment interaction $p = 0.5632$). Thus, despite the increased lever pressing observed in the hM4Di group (**Figures 6c, e, f**), these additional presses did not translate into more advantageous action patterns for reward earning. Consequently, SST IN inhibition causes mice to generate unnecessary lever presses unrelated to reward acquisition.

Although our definition of SEQ efficiency does not incorporate reward rate directly, it specifically penalizes unnecessary actions by counting lever presses that exceed the optimal number (Δ = actual presses – optimal presses, for values > 0) within individual action sequences. In this view, lever presses beyond the sequence criterion, or at rates exceeding 4 Hz, are considered unnecessary muscle activity and energy expenditure rather than an optimal strategy for the task. This interpretation aligns with the observation that control mice typically produce ~ 4 – 5 lever presses per sequence trial (**Figure 6c or 7c**). The sequence efficiency metric is therefore designed to favor the optimal sequence generation pattern naturally exhibited by wild-type mice. Although our efficiency function relies on empirical parameters due to the absence of a theoretical behavioral model, this analysis may provide insight into the functional role of SST INs in organizing optimal, energy-efficient action sequences.

Interpretation of activity differences between NTC-FR4 and TC-FR4

When looking at the changes in SST IN activity between the NTC-FR4 and TC-FR4(4Hz) conditions, one explanation for the increase in activity could be that the animals are doing more vigorous movements (more presses and/or faster presses). The division into classes in Figure 4 further highlights this, as the more vigorous SEQ class (class 1) seems to have the largest peak in the calcium transients. It seems that if you did a convolution of the lever presses with a smoothing kernel you could produce something that looks a lot like the average dF/F 's.

(Response) Thank you for suggesting this interesting idea. We agree that, in some cases, elevated SST calcium activity correlates with tightly clustered lever presses during TC-FR4 training. Thus, the probability distribution of lever presses resembles the $\Delta F/F$ profile. Specifically, the data shown in Figure 4c and 4i, j, n are consistent with the interpretation that higher-frequency lever pressing is associated with increased peak SST activity. Similarly, the comparison of SST activity between NTC-FR4 and TC-FR4 in Figure 3i and 3j supports this view.

However, this relationship does not generalize to the FR1 training condition. While we agree that the reviewer's suggestion is valuable and worth considering, we interpret the difference in SST activity between FR1 (Day 7) and FR1 (Day 21) as a form of **disengagement**, reflecting reduced cortical involvement over time as learning progresses. A similar pattern of disengagement is observed between NTC-FR4 (Day 1) and NTC-FR4 (Day 7), likely due to the relative simplicity of the task. This interpretation is supported by the lack of changes in structural features (e.g., sequence efficiency, lever press frequency/sequence, number of lever presses/sequence) in Extended Figure 3c–f.

By contrast, the difference in SST activity between NTC-FR4 and TC-FR4 (Figure 3) reflects **re-engagement** of M1 activity to support more complex action sequence execution.

The convolution idea would be correct assuming calcium activity is related to individual lever presses. However, what we found is instant onset and delayed onset SST INs encode specific behavioral events. We don't have the ability to discern individual lever press with calcium. But reflecting the reviewer's suggestion we would like to add an interpretation as follows.

(Manuscript)

(Line 378) In the more complex, time-constrained action sequences (e.g. TC-FR4), we observed that a strong correlation was maintained between SST INs activation and action sequence execution as task demands increased and proficiency decreased (Fig.3). If individual lever presses or discrete action components evoke a calcium transient in SST INs, the elevated calcium signal observed during time-constrained action sequences may reflect the accumulation of calcium across multiple presses or concatenated actions. This SST INs activity was modulated based on the structure of on-going action sequences (Fig.4).

Minor concerns:

Line 32 (also line 322): I would caution against using the word "persisted" when talking about results for the TC-FR4 training, as it gives the impression that the activity is unchanging, when it is in fact changing. Maybe a more descriptive phrase like "did not decrease" is better.

(Response) Thank you for pointing this. We changed the word in the revised manuscript as suggested.

(Updated manuscript)

Line32: the action-related SST IN activity redistributed and ~~did not decrease~~ **persisted**.

Line 342: but ~~did not decrease~~ and temporally redistributed as the timing

Figure 1: When using calcium imaging, one must be careful when interpreting differences in dF/F magnitudes across different cell types. The magnitude of these relative changes depends on many factors unrelated to neuronal firing rates, like differences in the expression level of indicator proteins and the calcium handling within cells (e.g. the intracellular concentration of endogenous calcium buffers). Given this, differences like those reported between SST cells in L2/3 and L5/6, which could be genetically distinct, do not necessarily reflect differences in activity level. The observations reported are meaningful and important, but this caveat in interpreting the differences should be made explicit.

(Response) We appreciate the reviewer's important point regarding the interpretation of calcium imaging data across different cell types. We are aware that $\Delta F/F_0$ (%) magnitude can be influenced by several confounding factors, our analysis focused on detecting transient calcium signals in a way that minimizes the influence of non-transient background fluorescence (e.g. endogenous calcium buffer effect) and cell-to-cell variability in expression level of calcium indicator. Baseline fluorescence (F_0) was defined individually for each cell based on spontaneous background activity, and only calcium traces exceeding a signal-to-noise threshold and exhibiting clear transients were identified as valid signals. Population-level activity was then computed by averaging across a large number of identified cells and multiple behavioral trials. In theory, this approach aims at reducing variability unrelated to action-locked calcium responses. Nonetheless, we agree that interpreting calcium imaging data is fundamentally challenging compared to electrophysiological measurements. We have explicitly mentioned this limitation in the revised Discussion section.

(Updated manuscript)

Line 511: Our study mainly focuses on neural activity of SST INs in M1 L5/6, which exhibit consistent action-locked calcium responses (Figure 1k, o, p, q, and Extended Figure.1) compared to those in L2/3 or PYR neurons. However, limitations inherent to fluorescence intensity measurements ($\Delta F/F_0$ %)—such as variability in background signal or contributions from endogenous calcium fluctuations—should be considered. Although our $\Delta F/F_0$ (%) analysis was

intended to minimize cell-to-cell variability in background fluorescence and to exclude non-action related calcium signals (see Methods), future studies utilizing in vivo electrophysiological recordings will be valuable for further characterizing layer-specific neural dynamics.

Specifying the site of activity measurements: The end of the Results section on Figure 1 and the beginning of that for Figure 2 (page 4) seem to imply that all neural analysis is done on SST cells specifically in L5/6. However, L5/6 is not mentioned again specifically until late in the Discussion. Because there are repeated references to SST interneurons in the intervening text, it is a bit ambiguous whether all the neural activity analysis was specifically on L5/6 cells. In addition, from the description of the experiments reported in Figure 6 and 7, including the depth of viral injection, it seems that the perturbation experiments reported in these figures involved all SST cells, not just those in L5/6. I do not see big issues with this difference, but it should be made explicit.

(Response) It is correct that our calcium imaging data and analyses were focused on SST INs in L5/6. The manuscript already indicated in “Line 124 : ... how the heightened SST IN calcium activity in M1 L5/6 changes during motor program consolidation and action sequence reorganization.” , “Line 130: In M1 L5/6 SST INs, we measured single-cell calcium activity...” . However, we added further information as follows.

(Updated manuscript)

Line 167: Next, we examined how stabilized SST network activity in M1 L5/6 changes....

Line 203: profile modulation (e.g. $\Delta F/F_0$ of GCaMP6f) in the action-locked SST network activity in M1 L5/6 may reflect structural changes of ...

Line 240: SST IN activity (Fig.3 and Fig.4) in M1 L5/6 into single-cell calcium responses.

(Response) Additionally, while our optogenetic perturbation was designed to selectively target L5/6 using an optical fiber, the chemogenetic inhibition approach is not layer-specific. In the revised manuscript, we have noted the layer information, and mentioned that SST activity in L2/3 may also contribute to the effects observed in the chemogenetic experiments

(Updated manuscript)

Line 404: However, both chemogenetic and optogenetic inhibition eventually disrupted the structure and efficiency of action sequences specifically during 4Hz and 8Hz TC-FR4 lever-press training (Fig.6d and Fig.7d). While optogenetic inhibition was somewhat layer-specific—given that the placement of the optical fiber likely targeted SST INs in L5/6—the chemogenetic approach was not layer-specific. Nonetheless, both manipulations resulted in consistent behavioral disruptions. This disruption coincided with the emergence of temporally confined lever-pressing patterns (Fig.6 e,f and Fig.7 e,f), leading to an increase in incomplete lever press sequences and a decrease in complete sequences (Fig. 6g,h and Fig.7g,h).

Line 61: “M1 PYR neurons can even become independent of movement upon extended training” - this statement is a bit redundant given the preceding text.

(Response) As we revised the introduction in response to another reviewer’s request, this redundant statement was removed.

Line 98: It would be helpful if this first section of the Results included at least a brief description of the behavioral paradigm.

(Response) Thank you for this constructive suggestion to improve the readability of our manuscript. In response, we have now included a brief description of the behavioral paradigm at the beginning of the Results section, as shown below.

(Updated manuscript)

Line 104: Our experimental setup and representative calcium traces are shown to illustrate raw activity patterns of cortical neurons in relation to the single lever press action (Fig.1). To examine the activity of identified cortical neuron subtypes during motor program acquisition, we trained mice on a self-paced lever-press task (Fig.1a). In the first paradigm, mice were trained to perform a single lever press to obtain a reward, known as a fixed-ratio 1 (FR1) schedule. This task was implemented over 7 days to characterize M1 neural responses associated with individual lever presses. The same training was then extended 21 days to examine how calcium activity changes during the execution of a simple, repeated motor program. In a second set of experiments, mice were trained on a more complex paradigm requiring four lever presses per reward (fixed-ratio 4 or FR4). Initially, the task was performed under a non-time-constrained FR4 schedule (NTC-FR4). The timing requirement was then progressively tightened so that mice were required to complete four or more lever presses within a specific time window, forming a time-constrained FR4 schedule (TC-FR4). This design allowed us to examine how neural activity correlates with the organization of lever presses into structured action sequences. During task performance, we monitored *in vivo* single-cell calcium activity in the M1 with a miniaturized fluorescence microscope and implantable gradient-index (GRIN) lenses.

Line 120: “synchrony” generally refers to simultaneous spike firing, but the measures used here, especially when applied to calcium sensor measurements, will reflect correlations of activity on longer timescales that do not seem to meet the typical definition of synchrony.

(Response) We appreciate this point and agree that the traditional definition of spike synchrony may not fully apply in the context of calcium imaging. To better convey the nature of our observations, we have revised our terminology to use “synchronized calcium responses” rather than “synchrony,” as we believe this more accurately conveys the concept of co-occurring calcium activation across cells.

(Updated manuscript)

Line 93 : ~~network synchrony~~ amplitude of synchronized calcium transients in L5/6 SST INs actually decreased with task consolidation.

Line 136: We then assessed M1 SST network synchrony as Pearson’s cell-to-cell correlation coefficient.

→ To examine potential differences in network activity across cortical layers, we assessed Pearson’s correlation coefficients across cells.

Line 139: ~~The analysis revealed that network synchrony was higher in the SST INs than the PYR neurons in L5/6, but not in L2/3~~

→ The analysis revealed that both PYR neurons and SST INs in L5/6 exhibit higher cell-to-cell correlation compared to the calcium responses in L2/3.

Line 162: Thus, SST IN activity exhibited a progressive decrease in action- locked ~~synchrony~~ calcium responses (Extended Fig.2d)

Line 196: ~~This indicates the maintenance of SST IN synchrony as mice efficiently reorganized their behavior from simple to more complex motor programs.~~ (we deleted this sentence)

Line 171: “at” should be “during”

(Response) Thank you for pointing this out. We have corrected the term.

(Updated manuscript)

Line 188: increased ~~at~~ during 4Hz-TC-FR4 compared to NTC-FR4

Line 180: “This indicates the maintenance of SST IN synchrony as mice efficiently reorganized their behavior from simple to more complex motor programs.” This statement reflects an interpretation of the absence of a significant difference as reflecting the absence of a difference. Such inferences are not statistically valid. This interpretation should be removed.

(Response) Thank you for your valuable comments. We deleted the statement in the revised manuscript in response to the comment, and also modified the sentence as follows:

(Updated Manuscript)

Line 197: Consequently, these data demonstrate that, while the temporal distribution of M1 SST network activity shifts during the reorganization of complex motor programs, the cell-to-cell correlation of SST IN activity was not significantly altered.

Line 207: It should be clarified what is meant by “incomplete”. For incomplete sequences, are the animals just choosing not to retrieve the reward, or is the reward not dispensed? If it is the former, is the reward just left there for the animal to get in the next trial, and could this behavioral change be related to satiation and/or motivation?

(Response) we already have definition in the original manuscript, but we updated our definition reflecting the reviewer’s confusion.

(Updated Manuscript)

Line 222: In contrast, a subset of C3 SEQs were defined as “incomplete SEQ” when mice executed slow and elongated lever press sequences that failed to meet the temporal requirement for consecutive four or more lever presses. In these cases, no reward was dispensed—without reward retrieval (Fig.4j).

Line 217: “INs” should be “IN”

(Response) We have corrected the typo in the revised manuscript

(Updated Manuscript)

Line 234: SEQs were distinctively encoded in SST IN activity

Line 234-235: Is the sharper response at higher frequency perhaps because the movement is faster? Also, not entirely clear what “stabilized” means here.

(Response) We recognize that the term “stabilized” may be open to interpretation. Since this is a supporting data in the Result section, we have revised the sentence to describe the observation more neutrally, minimizing interpretation.

(Updated Manuscript)

Line 251: Notably, the instant calcium response, time-locked to the onset of action sequences during the 8Hz training schedule, was reduced and showed less variability compared to the 2Hz and 4Hz schedules as training progressed. ~~became sharper and stabilized during performance on the 8Hz schedule as training progressed~~ (Extended Fig.5g,h).

Line 288: “minute-to-hour” should be “minutes-to-hours.”

(Response) We have corrected the typo in the revised manuscript

(Updated Manuscript)

Line 305: perhaps leading to long-lasting effects on basal SST activity and broader cortical networks on a timescale of minutes to hours ~~a minute-to-hour timescale.~~

Line 301: The authors justify the analysis shown in Figure 7i-7n as looking for changes “within a single experimental subject.” The notion that the statistics here are showing differences within individual mice is not supported - groupwise stats are still used. Moreover, the results in these panels seem expected given the results in 7b-d. A different justification for including this analysis seems necessary, or else further clarification.

(Response) We appreciate the reviewer’s careful reading. We realize that our initial description may have been unclear. The optogenetic protocol used in Figure 7b–d triggered laser activation with each action initiation (i.e., the first lever press of a sequence). While this protocol allowed us to assess group-level effects of SST IN inhibition by comparing eNpHR and eYFP groups, it did not address whether action sequence patterns within a session would be altered under laser ON versus

OFF conditions within eNpHR expressing mice. To address this, we implemented a different protocol—the Periodic Laser Modulation (PLM) paradigm—illustrated in Extended Figure 7a. In this design, laser illumination was alternated in five-minute ON and OFF blocks within the same session. Essentially, it highlights the same concept, but the PLM paradigm additionally clarifies that optogenetically induced behavioral modulation can also occur within a session as opposed to the continuous action-activated optogenetic protocol in Figure 7b-d.

(Updated Manuscript)

Line 319 : ~~Next, we investigated whether optogenetic silencing of SST IN activity would directly modulate kinematic structure of action sequences within a single experimental subject.~~ Next, we asked whether mice expressing eNpHR would exhibit different patterns of lever pressing depending on the presence (Laser-ON) or absence (Laser-OFF) of laser illumination within a session. Note that the laser protocol used in Figure 7b-d does not include laser-OFF intervals.

Line 322: “ In our Periodic Laser Modulation (PLM) protocol, the action-specific laser-illumination was alternatively enabled and disabled every five-minutes (**Extended Fig. 7a**). Thus, this test allowed us to assess whether SST IN activity actively modulates the structure of action sequences within a session.

Line 312-313: references to Fig.7m and Fig.7l seem to have their locations switched.

(Response) We have corrected the typo in the revised manuscript Line 332 and Line 334. Thank you for pointing this out.

Line 390-391: This description is confusing, because earlier (line 259) the authors do refer to what they measured as “kinematics.”

(Response) We appreciate the reviewer’s observation and acknowledge the confusion caused by terminology. To clarify our key point, we revised the phrasing as follows:

(Updated Manuscript)

Line 416: ...focusing on forelimb trajectory or ~~kinematics~~ motor parameters (e.g. speed or acceleration) as indicators of motor learning [10] [29, 31]. In contrast, by analyzing lever press patterns on a trial-by-trial basis in our study, we assessed action performance and efficiency as opposed to ~~individual movement kinematics~~—motor parameters reflecting isolated forelimb movements. Thus, our motor metrics might not fully capture subtle variations in stereotyped forelimb ~~kinematics~~ movements (some of which may show inter-individual variability),

Line 442: “pattern” should be “patterns.”

(Response) We appreciate the suggestion; however, we believe that "pattern" is appropriate in the context of the following sentence (Line 495 in the updated manuscript): “Another activity **pattern** of SST INs is characterized by a delayed calcium response following the onset of action sequences.”

Line 447: “might also regulate the termination of action sequences” - this could be easily tested by aligning activity on the last lever push.

(Response) Thank you for the helpful suggestion. In response, we re-aligned the delayed onset SST activity and found interesting features. Delayed onset SST activity exhibits a ramping pattern before the action sequence termination and generates a calcium transient peak associated with the subsequent transition to reward port access (see below, **extended Figure 9**). This data supports the suggested concept. We updated the manuscript as follows:

(Manuscript)

Line 500: “ ...suggesting that the ongoing activity of SST INs (especially those that code for SEQ duration) might also regulate the termination of action sequences, which is crucial for organizing and reorganizing the temporal structure of motor programs [46]. While delayed-onset SST INs show a dispersed activity pattern when aligned to the initiation of the action sequence (**Extended Fig.5a**), likely due to the probabilistic distribution in sequence duration (**Fig. 5g**), their responses become more temporally aligned when referenced to the final lever press or the subsequent transition to reward port access (**Extended Fig.9**).

(Updated Figure)

Extended Figure 9. Delayed onset SST INs profile reflects ongoing action sequence structure and subsequent transition. (a) Examples of lever press event aligned to the final lever press of each sequence during TC-FR4 4Hz training. (b) Prior the final lever press (time 0), calcium profiles of delayed-onset SST INs exhibit distinct ramping patterns from -2 to 0 seconds between short SEQ and long SEQ, but a calcium peak is observed following the final lever press (c) This calcium peak is associated with reward port access and is not influenced by the presence or absence of reward. (d) Average calcium activity of delayed-onset SST INs prior to the final lever press (ramping region from -2 to 0 seconds) varies significantly with sequence structure (two tailed unpaired t-test, delayed onset SST, n= 68 cells, **p=0.0059) (e) Calcium peak amplitude around reward port access does not significantly differ based on reward outcome (two tailed unpaired t-test, delayed onset SST, n= 68 cells, **p=0.1109)

Line 453-454: this bit is unclear - it is not clear what aspect of the figure indicates oscillatory activity. Activity in a motor region during an oscillatory movement to some extent has to be oscillatory. The Churchland observation of rotational dynamics was surprising because it was occurring during ballistic reaching at frequencies not apparent in the movement itself.

(Response) We appreciate the reviewer's request for clarification. In the original manuscript, we were referring to Figure 5j and Extended Figures 5d and 5f. We noted that the observed activity patterns resemble those shown in prior work (Churchland et al., Nature, 2012). A key insight from that study is that oscillatory modulation of neural firing rates is a fundamental feature of M1 activity. It is observed not only during rhythmic movements but also during ballistic reaching. Based on this, we speculated that the calcium profiles observed in our **Extended Figure 5** (c,d,e,f) may reflect underlying oscillatory dynamics in M1. We understand electrical and optical measurements could not be equivalent. Nevertheless, we performed frequency spectra analysis on the delayed-onset SST activity and found significant frequency component modulation depending on the sequence structure (e.g. short SEQ vs long SEQ) as shown in **Figure R3**. We believe the reference paper would be useful for general readers seeking related information about M1 neural dynamics.

(Updated Manuscript)

Line 508: It is also interesting that the activity pattern in M1 SST INs (**Fig.5j** and **Extended Fig.5 d,f**) resembles eventually ~~mimors~~ the neuronal population response in the motor cortex which exhibits brief but strong oscillatory patterns across species [17].

Figure R3. Frequency spectra analysis of SST INs during the execution of Long and Short SEQs. (a) Frequency spectra of delayed-onset SST IN activity during Long SEQ execution in the TC-FR4 8Hz schedule ($n = 3$ mice, 333 neurons, 80 randomly selected SEQ). (b) Frequency spectra of calcium activity during Short SEQ execution in the TC-FR4 8Hz schedule ($n = 3$ mice, 333 neurons, 80 randomly selected SEQ trials). (c) Average frequency spectra of calcium activity. Stronger frequency components (between 0.5-1Hz) in the calcium activity were found during the execution of short SEQ. Shaded areas represent SEM. (Two-way ANOVA, Freq effect: $***p < 0.0001$, SEQ effect: $p = 0.59$, Freq x SEQ effect: $* p = 0.0140$) (d) Average frequency spectra of instant-onset SST IN activity are not significantly different between long and short SEQs.

Methods: There is insufficient explanation of how the hierarchical clustering of the different sequence classes was done (was it single linkage? complete linkage? a density based method like DBSCAN? Also, what was the distance metric used?)

(Response) We appreciate the reviewer’s request for clarification. We used python script for hierarchical clustering with Ward linkage as implemented in `scipy.cluster.hierarchy.linkage` and `sklearn.cluster.AgglomerativeClustering`. This method is neither single nor complete linkage. The distance metric used was Euclidean, as required by the Ward method. We did not use density-based methods like DBSCAN in this analysis.

(Updated Method)

Line 742: The hierarchical clustering and dendrogram analyses were conducted using a custom Python script (Python 3.10.6) with Ward linkage as implemented in `scipy.cluster.hierarchy.linkage` and `sklearn.cluster.AgglomerativeClustering`. The distance metric used was Euclidean.

Figures 1b/2a/3a/4a: it is not clear what is being shown in the color bar at the top.

(Response) We appreciate the reviewer’s comment. In Figure 2a, the color bar at the top indicates training phases: purple represents early training (corresponding to the purple trace in Figure 2d), and gray represents late training (corresponding to the gray trace in Figure 2e). A similar scheme is applied in Figure 3a, where gray indicates NTC-FR4 and magenta indicates TC-FR4, consistent with the color-coded traces in subsequent panels. Figures 1b and 4a include replicated plots. We tried to maintain the same color scheme in the training bars to preserve consistency in the calcium profiles.

Figure 1q legend: says “activity map in (n)” but they probably mean “activity map in (k)”

(Response) Thank you for pointing this out, we found there were two typos. The averaged profiles in panels (n) and (q) correspond to the activity maps shown in panels (m) and (p), respectively. We have corrected these errors in the updated figure legend.

(Updated Caption for Figure1)

- (n) Averaged of peak normalized $\Delta F/F_0$ activity map in (m)
- (q) Averaged of peak normalized $\Delta F/F_0$ activity map in (p).

Figure 3 legend: on line 90, lever is misspelled.

(Response) Thank you for pointing this out, we have updated the typo

Figures 6e-h and Figure 7e-n: The figures here and the accompanying text in the Results, do not clarify when during the training schedule these effects were measured (unless I missed it). The previous panels in each figure locate the effects to later in the schedule - were these comparisons done specifically at those later time points? This should be clarified either way.

(Response) We thank the reviewer for pointing this out. The comparisons shown in Figures 6e–h and 7e–n were indeed performed during the late phase of training. We have now clarified this in both the figure legends and the corresponding sections of the Results text to avoid any ambiguity.

(Updated Figure Caption)

Figure 6 (e) Representative lever-press event distribution in the control group (left, gray) and hM4Di group (right, magenta) during TC-FR4 8Hz training. Total 500 sequence (SEQ) trials were randomly selected from each group (control n=7 mice, hM4Di n=6 mice)

Figure 6 (e) Representative lever-press event distribution in the eYFP group (left, gray) and eNpHR group (right, green) during TC-FR4 8Hz training. Each dot denotes individual lever-press. Laser illumination is on for 5 seconds. Total 500 sequence (SEQ) trials were randomly selected from each group (eYFP n=6 mice, eNpHR n=5 mice).

Extended Data Figure 5: The text here notes the differences between the Instant and Delayed onset populations, in terms of the differences between lower and higher frequency responses. However, in the figure, it does not appear that the same statistical tests were performed - the brackets in h and j indicate different comparisons. In addition, it is important to keep in mind that the absence and presence of significant differences in two statistical tests is not an appropriate way to establish a difference between two conditions. The appropriate approach here, as is standardly done when comparing drug effects versus those of placebo, is to directly measure the difference between the two conditions (here the response types) and show that the difference is significantly greater than 0.

(Response) We appreciate the reviewer’s attention to this important statistical issue. In Extended Data Figure 5, our intent was to describe two distinct patterns of SST IN activity—those with instant versus delayed onset responses—each characterized independently. We did not perform a direct statistical comparison between data in g,h vs i,j. Instead, our analyses focused on the response properties within each population separately. To avoid potential confusion, we have revised the relevant section of the Discussion to clarify that these patterns are independently characterized and not directly compared.

(Updated Manuscript)

Line 495: ~~In contrast,~~ Another activity pattern of SST INs is independently characterized by a delayed calcium response following the onset of action sequences. This response was specifically pronounced in the 4Hz or 8Hz training schedules compared to 1Hz training schedule (Extended Fig.5 i,j).

Reviewer #4 (Remarks to the Author):

The authors study how Sst cells participate in a lever press and sequential lever press task. They find that Sst activity is synchronous around lever presses, particularly in deep layers, reduces with training, and increases when mice switch to a more difficult task schedule. Chemo- or optogenetic suppression of Sst activity increased lever pressing but disrupted its structure leading to lower reward rates.

The work this paper describes is well done, and the conclusions reasonable. I therefore do not have any major comments or suggestions. Two minor suggestions:

(Response) We appreciate the reviewer's positive feedback and are grateful for recognizing the key conceptual contributions of our study. We are pleased that the reviewer found the conclusions well supported and the overall study well executed. We have carefully considered the two minor suggestions and addressed them in the revised manuscript. Thank you for your thoughtful and encouraging evaluation.

1. In every figure please state which layer you are recording from. My understanding is that except for Fig. 1 it is all in L5/6, but this would be clearer if explicitly stated

(Response) Thank you for pointing this out. It is correct that our calcium imaging data and analyses were focused on SST INs in L5/6. Only in figure 1, we characterized other layers and PYR neurons to highlight the pattern of SST IN activity in L5/6. The manuscript already indicated in " Line 124 : ... how the heightened SST IN calcium activity in **M1 L5/6** changes during motor program consolidation and action sequence reorganization." , "Line 130: In **M1 L5/6** SST INs, we measured single-cell calcium activity..." . However, for further clarification, we added information as follows.

(Updated manuscript)

Line 167: Next, we examined how stabilized SST network activity in **M1 L5/6** changes....

Line 203: profile modulation (e.g. $\Delta F/F_0$ of GCaMP6f) in the action-locked SST network activity in **M1 L5/6** may reflect structural changes of ...

Line 240: SST IN activity (**Fig.3** and **Fig.4**) in **M1 L5/6** into single-cell calcium responses.

2. Extended Figure 5a makes it look like the delayed-onset Sst cells are not synchronous, but rather fire in a sequence. If so, that is a contrast to the highly synchronous activity seen at lever pressing. Is that correct? If so it might be worth pointing this out.

(Response) Thank you for this insightful comment. The instant-onset SST interneurons (INs) exhibit tightly time-locked activity aligned with the initiation of action sequences, consistent with their synchronous recruitment at movement onset. In contrast, delayed-onset SST INs display greater temporal variability when aligned to the initiation point. This variability likely reflects the distribution of action sequence durations, which are not fixed but follow a probabilistic pattern across trials (**Figure 5g**). As a result, when activity is aligned to sequence initiation, delayed-onset responses appear more dispersed.

However, in our updated analysis (see below, **Extended Figure 9**), we found that delayed-onset SST IN activity becomes more temporally aligned when time-locked to the final lever press. Also, the synchronized calcium activation aligned to the subsequent transition (reward port access) can be seen in Extended Figure 9c and Figure R9. This suggests that these neurons may encode both the structural features of the ongoing action sequence and the transition associated with its termination. Thus, in different behavioral events, delayed-onset SST INs still maintain the synchronized calcium activation. We have incorporated the new data into the revised manuscript.

(Manuscript)

Line 502: While delayed-onset SST INs show a dispersed activity pattern when aligned to the initiation of the action sequence (**Extended Figure 5a**), likely due to the probabilistic distribution in sequence duration (**Figure 5g**),

their responses become more temporally aligned when referenced to the final lever press or the subsequent transition to reward port access (**Extended Figure 9**). Notably, ~~the delayed calcium response emerged after action sequence initiation, and our optogenetic protocol was specifically targeted to silence this activity.~~

(Updated Figure)

Extended Figure 9. Delayed onset SST INs profile reflects ongoing action sequence structure and subsequent transition. (a) Examples of lever press event aligned to the final lever press of each sequence during TC-FR4 4Hz training. (b) Prior the final lever press (time 0), calcium profiles of delayed-onset SST INs exhibit distinct ramping patterns from -2 to 0 seconds between short SEQ and long SEQ, but a calcium peak is observed following the final lever press (c) This calcium peak is associated with reward port access and is not influenced by the presence or absence of reward. (d) Average calcium activity of delayed-onset SST INs prior to the final lever press (ramping region from -2 to 0 seconds) varies significantly with sequence structure (two tailed unpaired t-test, delayed onset SST, n = 68 cells, **p=0.0059) (e) Calcium peak amplitude around reward port access does not significantly differ based on reward outcome (two tailed unpaired t-test, delayed onset SST, n = 68 cells, **p=0.1109)

Figure R9. Delayed-onset SST INs detected during TC-FR4 4 Hz training (n = 68 cells), aligned to two different behavioral events. **Left:** activity aligned to sequence (SEQ) initiation (time 0). **Right:** the same cells aligned to reward port access, showing more temporally aligned and synchronized calcium activation.

REVIEWER COMMENTS

Reviewer #1 (Remarks to the Author):

The authors did a truly commendable job in addressing all of the reviewers comments - a strong article is now even stronger and in my opinion is ready for publication.

Response:

We sincerely thank the Reviewer for the very positive feedback and for recognizing the improvements made in response to the previous round of comments. We are delighted that the revisions have strengthened the manuscript and appreciate your recommendation for publication.

Reviewer #2 (Remarks to the Author):

I commend the authors on their thorough responses to all reviewers. My comments/concerns were adequately addressed. While some of the 'reviewer figures' could be helpful to include in the manuscript, I trust the judgement of the authors on these decisions.

Response:

We greatly appreciate the thoughtful evaluation and encouraging comments. We are pleased that our revisions have addressed all concerns satisfactorily. In the process of addressing comments from other reviewers, we have also updated several figures and datasets in the main manuscript to further strengthen the work.

Reviewer #3 (Remarks to the Author):

I commend the authors for doing a thorough job of addressing the reviewer concerns. The bulk of my concerns have been suitably addressed. A few issues remain, which I address below.

Response:

We appreciate the reviewer's thoughtful and constructive feedback. We recognize that some of these concerns are critical and some are fundamental questions that extend beyond the current scope of our study. Nevertheless, we fully agree that they raise important discussion points that have further strengthened the manuscript. In response, we have performed additional experiments and analyses, and revised relevant sections accordingly. We hope that the new data and updated descriptions adequately address the remaining issues and provide greater clarity.

- Concerns about the damage to M1

Apologies for not making my original concern clear. My concern is specifically with the comparison between L2/3 and L5/6 SST cells. While this comparison is only done in Figure 1, it does seem important, as the authors use it to justify their subsequent focus on L5/6 (line 140). I do not see a control included that assesses whether the difference in activity between L2/3 and L5/6 SST cells is due to the different degree of damage between the L2/3 and L5/6 imaged animals. It is also not clear how such a control could be done, since the cells immediately above the imaged L5/6 SST cells are the ones providing strong input to L5.

I think it is necessary that this confound be explicitly addressed in the Results when describing the differences seen between L2/3 and L5/6 SST cell activity.

Response:

We fully acknowledge the reviewer's concern that precise layer-specific characterization of activity between L2/3 and L5/6 could be confounded by damage to the superficial layer during deep layer recordings. While this is a risk inherent to any invasive technique (including the now-widespread use of GRIN lens-based microendoscopic imaging), we agree that it is important to specifically point out the possible confound in our study. At present, there is not a suitable method to image deep layer M1 SST-INs without some damage to more superficial layers in some cortical area. To address the point in the Results we have added the following statement:

Updated Manuscript (from Line 131, orange text is updated)

We found that the action-locked $\Delta F/F_0$ amplitude was higher in SST INs in M1 L5/6 compared to that of SST INs in M1 L2/3 and PYR in both L2/3 and L5/6 (**Fig. 1j,k**, and **Extended Fig. 1c,d**). Representative raw calcium traces that contribute to the population-averaged $\Delta F/F_0$ profiles of SST INs in L2/3 and L5/6 are shown in **Fig. 1l,m,n** and **Fig. 1o,p,q**, respectively. In these example traces, SST activity in layer 2/3 was increased around individual lever presses, but the occurrence and amplitude of the increases showed considerable variability around each press (**Fig. 1l**). In contrast, activity in layer 5/6 showed more consistent increases, including relatively consistent transient amplitudes, around each press (**Fig. 1o**). Nevertheless, there are limitations to comparing cell-to-cell activity solely based on $\Delta F/F_0$ values. ~~To examine potential differences in network activity across cortical layers~~ To examine this characteristic at the network level independently of $\Delta F/F_0$ amplitude, we assessed Pearson's correlation coefficients across cells. This parameter calculates the linear relationship between a single cell calcium trace and all other neuronal traces during a 10 second time window centered around the lever-press. The analysis revealed that both PYR neurons and SST INs in L5/6 exhibit higher cell-to-cell correlation compared to the calcium responses in L2/3. (**Extended Fig. 1e,f**). It must be noted that placement of the GRIN lens for imaging in L5/6 entails some damage to superficial layers in M1, and this could affect L5/6 SST-IN activity by disrupting cross layer communication. Thus, the differences in lever-press related SST-IN activity in the different cortical layers must be interpreted with caution. The remainder of the study focused on L5/6, a

layer that plays a central role in motor output control but remains less well characterized than L2/3 in previous imaging studies [24,25,26].

We have also added this text related to the concern about this confound in the discussion:

Updated Manuscript (from Line 424, orange text is updated):

As mentioned in the Results, implantation of the GRIN lens can cause damage to superficial cortical layers, which may introduce a potential confound when measuring L5/6 SST-IN activity. This is a risk for all invasive procedures that require hardware implants in brain. At present, it remains challenging to image deep-layer M1 SST-INs in freely moving mice without any cortical damage, and the consequent potential circuit effects should be considered when interpreting the results. Nevertheless, it is notable that neurons can often survive initial damage from local implantation, as indicated by the many single-unit recordings performed with indwelling electrodes and electrode arrays [3,4,10,17]. In addition, a previous study [13] reported that action-locked M1 activity of superficial and deep layer cortical pyramidal neurons assessed with non-invasive 2-photon imaging in a head-fixed situation showed layer-dependent neural activity differences similar to what we observed in the present study (**Extended Fig 1.e,f**). Thus, the layer-specific activity of cortical neurons may be a general feature of action coding. Our observation that L5/6 SST-INs show distinct activity patterns with different operant training schedules (**Fig. 2** and **Fig.3**) and tasks (**Fig.4** and **Fig.5**) indicate that these neurons retain the ability to change their activity patterns even when measured with this invasive imaging procedure. In future work, head-mounted two-photon imaging or prism-based cross-sectional imaging may help address concerns regarding superficial layer damage in our characterization.

- Lack of evidence for discretely separable activity patterns

This result shown in Figure R6 is great - it should be added to the manuscript to support the claim of "mutually exclusive neuronal subpopulations" which I see remains in the manuscript.

Response:

We appreciate the reviewer's positive feedback. We have now incorporated this plot into the revised manuscript to strengthen the claim regarding mutually exclusive neuronal subpopulations.

Updated Manuscript (from Line 246, orange text is updated):

To better understand how trial-by-trial action components in TC-FR4 SEQs relate to SST IN activity at a single-cell level, we broke down the population average of SST IN activity (**Fig.3** and **Fig.4**) in M1 L5/6 into single-cell calcium responses (**Fig.5 a,b**). The population average of calcium events exhibited a bimodal distribution (**Extended Data Fig. 5a**), suggesting the presence of two distinct modes of calcium activity.

We examined whether these activity modes were represented within individual cells or by mutually exclusive groups of cells.

Updated Material (Extended Figure 5a):

(a) Bimodal distribution of SST IN calcium events during TC-FR4SEQ training (histogram generated from 712 averaged calcium profiles). Representative color coded-map of single cell calcium responses (session normalized $\Delta F/F_0$) for ‘instant-onset SST’ (n=62 cells, 3mice; bottom left) and ‘delayed-onset SST’ (n=163 cells, 3mice; bottom right), color values represent the session normalized $\Delta F/F_0$ of each SST IN, averaged over a training session

- Missing statistical tests for key comparisons (FIG1)

I still do not see a statistical test either in the text or figure legend that supports the claims made about Fig. 5j, that the responses are further delayed. One should be provided, or the claim should be omitted.

Response:

We apologize for not clearly stating that Figures 5k and 5l (calcium peak of individual profiles) present the statistical summaries of Figure 5j. We have now added this clarification to the Figure5 legend (j,l) and further included additional details regarding the statistical analyses as follows.

Updated Figure Legend Fig.5:

(j) Average calcium response of ‘delayed-onset SST’ for short SEQs (solid lines) and long SEQs (dotted lines) during the last 3 days of 4Hz (left; two-way repeated-measure ANOVA: time x SEQ, ** $p=0.0064$; time, **** $p<0.0001$; SEQ $p=0.278$) and 8Hz (right; two-way repeated-measure ANOVA: time x SEQ, **** $p<0.0001$, time, **** $p<0.0001$, SEQ $p=0.9779$) training schedules. Shaded areas denote SEM. The vertical bar represents a value of 0.1 in the session-normalized $\Delta F/F_0$.

(k,l) Statistical summary of plot (j), showing calcium peak locations of individual SST INs shifted significantly within a training session depending on short and long SEQs during the 4Hz schedule (two-tailed Wilcoxon paired test, $n=46$ cells, $p<0.0001$) and 8Hz schedule (two-tailed Wilcoxon paired test, $n=66$ cells, $p<0.0001$)

- Time course of perturbation effects across learning

The additional data presented here in Figure R7 seem to corroborate my original concern. These data present a very different picture of the effects of SST neurons on task performance at later time points. Though a significant difference is seen in the event distribution, the effect is relatively small compared to what is shown in Figure 6 and 7. Figure R7b does not align with Figure 6b-d or Figure 7b-d. Thus a substantial degree of the effects shown in those panels do seem to be a byproduct of repeated inactivation and not from the immediate effects of silencing SST neurons during the movement. The problem is compounded by the fact that the authors stress the difference in effects between easier and harder versions of the task, especially throughout the Discussion section on task specific inhibition effects. I think Figure 7 should be updated to include the data shown in Figure R7 and the interpretation of the inactivation effects in the manuscript should be updated to account for these new results. This would include eliminating the claims about stronger effects during the higher frequency tasks, which clearly are substantially confounded. If inactivation effects depend on the history of inactivation an animal has experienced during previous sessions, then comparing effects across stages of the TC-FR4 training does not seem appropriate.

Response: Immediate optogenetic inactivation of SST-INs is task-specific.

We noted that the reviewer suggested removing the emphasis on task-specificity, as the immediate effect (**Updated Figure 7p**) differs from the long-term optogenetic inactivation effect (**Updated Figure 7b**). The reviewer’s concern seems to be that the long-term behavioral differences are mainly driven by cumulative effects of repeated opto-inactivation rather than genuine task-specific modulation.

While it is reasonable to stress that altered lever press rate requires SST-IN activity suppression during less demanding tasks, the behavioral effect of suppression is expressed in a task-specific manner with increased press rates only observed at the highest frequency schedules in TC-FR4 training (**Figs. 6b, 7b**) and then only with ongoing suppression (**Fig.7j**).

We favor this interpretation because it consistently explains both our previous and newly updated data, whereas the cumulative-effect explanation alone encounters several difficulties. For instance, if the behavioral differences during TC-FR4 8Hz schedule were purely cumulative and independent from the task factor, one would expect a gradual or progressive change in behavior over time. Instead, we observed an abrupt emergence of behavioral differences that coincided precisely with the transition to a more demanding task stage (**Figs. 6b, 7b**). Notably, given the overall training period (4-6 weeks), the single-day difference between the last day of TC-FR4 training at 4 Hz and the first session at 8 Hz should not produce a dramatic change if the effect were cumulative rather than event-driven. Therefore, it is unlikely that long-term cumulative opto-inactivation effects would onset exactly at the task-switching point. Rather, the transition in behavioral schedule—from 4 Hz to 8 Hz—appears to be the critical factor triggering the animals' response to optogenetic inactivation, manifested as increased lever-press rates when subjects had prior optogenetic manipulation. Thus, the behavioral change is best explained as task-specific rather than a purely cumulative effect.

Nonetheless, we acknowledge that our experimental design cannot completely exclude the possibility that the behavioral effects observed under high-frequency TC-FR4 schedules reflect a combination of cumulative treatment exposure and task-specific modulation.

To clarify the task-specificity of immediate optogenetic inactivation effects, we added new data from TC-FR4 training schedules (1Hz and 8Hz) without previous optogenetic suppression (**Extended Figure 7i**). We prepared one cohort of eYFP- and eNpHR-expressing mice and examined their optogenetic inactivation responses on the first day of TC-FR4 training at the 8Hz schedule. Another cohort of eYFP- and eNpHR-expressing mice was tested on the first day of TC-FR4 training at the 1Hz schedule. We found a statistically significant difference in sequence parameters during 8Hz training, but not during 1Hz training (**Extended Figure 7j-l**). We believe these results address some portion of the reviewer's concerns. The text in the previous manuscript version mainly referred to the behavioral difference within the same cohort as the training schedule changed from 1Hz, 4Hz, to 8Hz. We have now updated these results when we refer to the task-specific aspect of immediate optogenetic inactivation. We revised the text as follows:

Updated Extended Figure 7:

Extended Figure 7. Lever-press distributions are actively modulated by laser ON/OFF controls, and immediate optogenetic inactivation effects of SST-INs differ depending on task

(a) TC-FR4 sequence training combined with the PLM schedule performed after long-term optogenetic suppression (top) and TC-FR4 sequence training conducted without prior optogenetic inactivation (bottom).

(b) Average peak-normalized lever-press probability distribution after long-term optogenetic suppression. Time = 0 corresponds to sequence initiation (SEQ initiation press excluded). The lever-press probability distribution was unchanged in the eYFP group with laser ON/OFF modulation (two-way ANOVA: Time, **** $p < 0.0001$; Laser ON/OFF, $p = 0.1505$; Time \times Laser ON/OFF, $p = 0.6518$).

(c) Average peak-normalized lever-press probability distribution aligned to sequence initiation (SEQ initiation press excluded) after long-term optogenetic suppression. The eNpHR group exhibited significant difference in lever-press probability distribution depending on laser ON/OFF modulation (two-way ANOVA: Time, ** $p = 0.0042$; Laser ON/OFF, $p = 0.012$; Time \times Laser ON/OFF, $p = 0.5317$).

(d) Average peak-normalized lever-press probability distribution across five sessions (30 min each) conducted without prior optogenetic inactivation. Time = 0 corresponds to sequence initiation (SEQ initiation press excluded). The eYFP group showed no significant difference between laser ON and OFF trials (two-way ANOVA: Time, $p < 0.0001$; Laser ON/OFF, $p = 0.0950$; Time \times Laser ON/OFF, $p = 0.2752$).

(e) Average peak-normalized lever-press probability distribution across five sessions (30 min each) conducted without prior optogenetic inactivation. Time = 0 corresponds to sequence initiation (SEQ initiation press excluded). The eNpHR group showed an elongated lever-press probability distribution on laser ON trials compared to laser OFF trials (two-way ANOVA: Time, **** $p < 0.0001$; Laser ON/OFF, *** $p = 0.0009$; Time \times Laser ON/OFF, $p = 0.0433$).

(f) Lever-presses per sequence (SEQ) (two-way repeated measure ANOVA: Group, $p = 0.9242$; Laser, ** $p = 0.0018$; Group \times Laser, * $p = 0.0192$; Sidak's multiple comparisons test: eYFP, $p = 0.6576$; eNpHR, *** $p = 0.0005$)

(g) Average SEQ duration (two-way repeated measure ANOVA: Group, $p = 0.3705$; Laser, * $p = 0.0341$; Group \times Laser, $p = 0.0715$; Sidak's multiple comparisons test: eYFP, $p = 0.9555$; eNpHR, * $p = 0.0118$)

(h) SEQ efficiency (two-way repeated measure ANOVA: Group, $p = 0.4145$; Laser, ** $p = 0.0080$; Group \times Laser, * $p = 0.0429$; Sidak's multiple comparisons test: eYFP, $p = 0.8049$; eNpHR, ** $p = 0.0027$)

(i) TC-FR4 sequence training without prior optogenetic inactivation, the laser activation was enabled at 1Hz schedule (top) and 8Hz schedule (bottom)

(j) Lever-presses per sequence (SEQ) showed non-significant Group \times Schedule interaction and a main effect of Schedule (two-way ANOVA: Group, *** $p < 0.0001$; Schedule, * $p = 0.0140$; Group \times Schedule, $p = 0.1160$; Sidak's multiple comparisons test: 1Hz, $p = 0.7360$; 8Hz, ** $p = 0.0092$)

(k) Average SEQ duration (s) showed a significant Group \times Schedule interaction and main effects of both factors (two-way ANOVA: Group, *** $p = 0.0001$; Schedule, * $p = 0.0124$; Group \times Schedule, $p = 0.0848$; Sidak's multiple comparisons test: 1Hz, $p = 0.7841$; 8Hz, ** $p = 0.0063$)

(l) SEQ efficiency also showed significant main effects and interaction (two-way ANOVA: Group, **p = 0.0010; Schedule, p = 0.1479; Group × Schedule, **p = 0.0066; Sidak's multiple comparisons test: 1Hz, p = 0.5310; 8Hz, **p = 0.0066). 1Hz eYFP n=6 mice, 1Hz eNpHR n= 6mice, 8Hz eYFP n=6 mice, 8Hz eNpHR n=8 mice.

Updated Manuscript (orange text is added):

Line 457: Given previous findings, it was unexpected that we did not observe significant motor impairments in freely moving mice during FR1 training, or during NTC-FR4 and TC-FR4 training at 1Hz and 2Hz, even with the application of chemogenetic inhibition and closed-loop optogenetic silencing of SST INs (**Fig. 6** and **Fig.7**). However, both chemogenetic and optogenetic inhibition eventually disrupted the structure and efficiency of action sequences specifically during 4Hz and 8Hz TC-FR4 lever-press training (**Fig.6d** and **Fig.7d**). This difference could be attributed to both the cumulative effects of long-term suppression and the immediate optogenetic inactivation during ongoing sequence production. Even when we examined how immediate optogenetic inactivation disrupts sequence organization without any prior opto-inactivation history, significant alterations in sequence parameters were observed only during the 8 Hz schedule (**Fig.7o-v** and **Extended Fig.7j-k**), whereas these differences were not detectable during the 1 Hz schedule (**Extended Fig.7j-k**), suggesting that involvement of SST-IN activity in behavior differs depending on the task.

While optogenetic inhibition was somewhat layer-specific—given that the placement of the optical fiber likely targeted SST INs in L5/6—the chemogenetic approach was not layer-specific. Nonetheless, both manipulations resulted in consistent behavioral disruptions. This disruption coincided with the emergence of temporally confined lever-pressing patterns (**Fig.6 e,f** and **Fig.7 e,f**), leading to an increase in incomplete lever press sequences and a decrease in complete sequences (**Fig. 6g,h** and **Fig.7g,h**). Thus, there appears to be an inconsistency: while SST IN inhibition specifically affected complex motor sequences in our study (e.g., TC-FR4 4Hz and 8Hz), previous studies with head-fixed mice performing stereotyped forelimb tasks observed motor skill impairment from the beginning.

Line 516 : Using cell-specific calcium imaging, we observed distinct activity patterns of M1 SST-INs between (1) the simpler FR1 task—which allows for greater motor variability and more naturalistic movements to complete the task—and (2) the more demanding TC-FR4 SEQ task, which imposes tighter temporal constraints and requires an unnatural, structured lever-press sequence that is more difficult to learn. This distinction is further supported by our behavioral findings, which show that inhibition of M1 SST-INs selectively impaired performance on the more demanding TC-FR4 SEQ tasks at 4Hz and 8Hz but had no effect on simpler tasks such as FR1, FR4, or the TC-FR4 SEQ tasks at 1Hz and 2Hz schedules: under the 8Hz schedule, compared to the 1Hz schedule (**Extended Fig.7i-l**). Overall, it is possible that M1 SST-IN activity partially reflects broader M1 circuitry, exhibiting task-dependent engagement characteristics. However, it remains an open question whether broader M1 dynamics shape SST IN

activity or, conversely, whether task-dependent M1 SST IN activity directly influences broader M1 circuitry during our behavioral paradigms.

Nevertheless, our experimental result underscores the task-specific properties of M1 SST INs activity within our assessment metrics. Notably, when the calcium activation in ‘delayed-onset SST’ became more pronounced (e.g., in TC-FR4 4Hz and 8Hz, **Extended Fig.5i,j**), the effects of chemo- and opto-inhibition also became more noticeable (**Fig. 6 and Fig.7**). Particularly, the abrupt change that coincided precisely with the transition to a more demanding task stage appears to be associated with the task-specific characteristics of SST-INs (**Extended Fig. 7i–l**). However, the effects of long-term suppression are likely superimposed on these results, as immediate optogenetic inactivation alone was not sufficient to induce changes in lever-press rate (**Fig. 7b, p**).

Previously, task-specific properties of SST INs activity have been shown by examining sequential activity patterns of PYR neurons in forward and backward treadmill running tasks [25]. This suggests that the contribution of SST INs may vary between motor programs that are previously learned or newly acquired. Our study extends this view by demonstrating that the involvement of M1 SST INs in action control varies with the complexity of motor tasks. Despite difference in motor tasks and testing configurations, both previous work and our results support the same idea of task-specific functional contribution of M1 SST INs, consistent with recent predictions regarding cortical IN functions [25, 26, 40] and task-dependent necessity of cortical regions [29] [31].

Response: immediate and cumulative effects of optogenetic suppression

We appreciate the opportunity to clarify the distinction between immediate and long-term optogenetic inactivation effects. The reviewer is correct in pointing out that effects on lever-press rate require cumulative effects of inactivation, raising the possibility that long-term suppression of activity throughout the training contributes to this behavioral change.

However, the demonstration in our periodic laser ON/OFF optogenetic experiment that enhanced press rate only occurs during optogenetic suppression of activity under the TC-FR4 8Hz schedule (**Updated Fig.7j**)—even when suppression was applied throughout TC-FR4 training (**Updated Fig.7i**)—supports the idea that ongoing SST-IN activity actively modulates lever press rate under this most challenging task condition. Therefore, if one assumes that cumulative effects arise earlier in the circuit and influence behavioral output independently of SST-IN activity, our evidence argues against this interpretation, as active laser ON/OFF control of SST-INs immediately produced behavioral differences during the TC-FR4 8 Hz task (**Updated Fig.7i-n**).

We instead propose that the cumulative effect represents the formation of a “permissive state” for inefficient high-frequency responding. While cumulative suppression appears to set the condition for overly exuberant lever pressing, it can only be expressed when activity is also suppressed during ongoing high-frequency action production (**Updated Fig.7b**). This most likely relates to the development of specific patterns of SST-IN activity (especially delayed activation) that begin to occur during TC-FR4 training at 1Hz

schedule and grow more pronounced as the pressing frequency requirement increases (**Extended Fig. 5**). However, control of lever pressing can be re-established if suppression is discontinued in the TC-FR4 8Hz condition (**Updated Fig.7j-n**), supporting the idea that SST-IN activity has an ongoing role in action control under this condition.

Updated Figure 7:

Figure 7. Action-specific inhibition of SST interneurons disrupts execution of efficient action sequence.

(a) Time-constrained FR4 sequence training schedule and optogenetic modulation protocol. Each lever press triggers 5 sec of laser illumination (532 nm, ~4mW)

- (b) Lever-press rate (levers/min) during a session increased in the eNpHR expressing mice (n=5), compared to the eYFP expressing control group (n=6) with the same opto-inhibition protocol. (two-way ANOVA repeated measure. Schedule ****p<0.0001, Treatment *p=0.0237, Schedule x Treatment ***p=0.0007)
- (c) Number of lever presses per sequence increased in the eNpHR mice, compared to the eYFP mice (two-way ANOVA repeated measure. Schedule ****p<0.0001, Treatment *p=0.0293, Schedule x Treatment ***p=0.0002)
- (d) Sequence efficiency coefficient (SEC) decreased in the eNpHR group compared to the eYFP group (two-way ANOVA repeated measure. Schedule **p=0.0024, Treatment *p=0.0114, Schedule x Treatment ****p<0.0001)
- (e) Representative lever-press event distribution in the eYFP group (left, gray) and eNpHR group (right, green) during TC-FR4 8Hz training. Each dot denotes individual lever-press. Laser illumination is on for 5 seconds. Total 500 sequence (SEQ) trials were randomly selected from each group (eYFP n=6 mice, eNpHR n=5 mice).
- (f) Peak-normalized lever-press distribution probability in eNpHR and eYFP mice. Time 0 is sequence initiation. The very first lever-press (SEQ initiation lever-press) was not included in the probability distribution curve. (two-way ANOVA. Time ****p<0.0001, Group **p=0.0078, Time x Group ****p<0.0001)
- (g) Proportion (%) of incomplete sequence within in the total unrewarded trials increased in the eNpHR group compared to the control group (two-tailed unpaired t-test **p=0.0071). Each dot denotes average proportion of incomplete sequence per animal.
- (h) Proportion (%) of complete sequence in the total rewarded SEQ trials decreased in the eNpHR group (two-tailed unpaired t-test **p=0.0066).
- (i) Periodic Laser ON/OFF modulation (PLM) training schedule
- (j) eNpHR group exhibited alteration in lever-press rate (levers/min) depending on the periodic laser modulation (5 min ON and 5 min OFF). Each dot represents an animal average (two-way ANOVA repeated measure, eYFP vs eNpHR effect: *p=0.0177, OFF vs ON effect: **p=0.001, Sidak multiple comparison within a group: OFF vs ON within eYFP p=0.1924, OFF vs ON within eNpHR ****p<0.0001)
- (k) In the same training, only eNpHR group increased number of lever presses per sequence in the laser ON period than laser OFF period (Two-way ANOVA repeated measure, OFF vs ON effect: *p=0.0285, eYFP vs eNpHR effect p=0.0624, Sidak's multiple comparison test, laser on/off effect within eYFP p=0.5593, within eNpHR **p=0.0032)
- (l) In the same PLM training schedule, proportion (%) of complete sequence was not significantly altered.
- (m) Within the same eNpHR expressing subject, proportion (%) of incomplete sequence increased during laser ON periods compared to laser OFF periods (two-tailed unpaired t-test *p=0.0105). Each dot denotes average proportion (%) of incomplete sequence per animal.
- (n) Sequence efficiency coefficient (SEC) decreased during laser ON periods compared to laser OFF periods (two-tailed Wilcoxon test *p=0.0312). In all plots, shaded area and error bars denote SEM.
- (o) TC-FR4 sequence training schedule and optogenetic modulation protocol during the 8 Hz task without prior treatment.
- (p) Average lever-press rate (lever presses / min) during late-phase opto-inhibition showed no significant difference between eYFP (n = 6 mice) and eNpHR (n = 8 mice) groups (two-way repeated-measures ANOVA: Group, p = 0.68; Training Schedule, ***p < 0.0001; Group x Schedule, p = 0.37).
- (q) Lever-press rate (lever presses / min) on the first day of the TC-FR4 8 Hz schedule was not significantly different between eYFP and eNpHR groups relative to baseline performance at the TC-FR4 4 Hz schedule (two-way repeated-measures ANOVA: Group, p = 0.3928; Opto-inactivation, p = 0.1079; Group x Opto-inactivation, p = 0.2758).
- (r) Representative lever-press event distribution in the eYFP group (left, gray) and eNpHR group (right, green) on the first day of TC-FR4 8Hz training in (i)
- (s) Lever-press distribution (two-way ANOVA: Time, ****p<0.0001; Group, *p=0.0060, Time x Group p=0.2889) and corresponding area under the curve (AUC, two-tailed unpaired-t test: *p=0.0378) are significantly different between eYFP and eNpHR.
- (t) Lever-presses per sequence (SEQ) showed a significant Group x Schedule interaction and a main effect of Schedule (two-way repeated-measures ANOVA: Group, p = 0.1194; Opto-inactivation, p = 0.3188; Group x Opto-inactivation, **p = 0.0078; Sidak's multiple comparisons test: eYFP, p = 0.3285; eNpHR, *p = 0.0144)
- (u) Average SEQ duration (s) showed a significant Group x Schedule interaction and main effects of both factors (two-way repeated-measures ANOVA: Group, p = 0.095; Opto-inactivation, *p = 0.0314; Group x Opto-inactivation, *p = 0.0174; Sidak's multiple comparisons test: eYFP, p = 0.9731; eNpHR, **p = 0.0038).
- (v) SEQ efficiency also showed significant main effects and interaction (two-way repeated-measures ANOVA: Group, p = 0.1005; Opto-inactivation, **p = 0.001; Group x Opto-inactivation, *p = 0.0206; Sidak's multiple comparisons test: eYFP, p = 0.5127; eNpHR, ***p = 0.0004).

This framework also accounts for the immediate optogenetic inactivation result (**Updated Fig. 7o-v**), which we performed following the reviewer's suggestion. Without long-term suppression, SST-INs were functioning normally and the circuit had not entered a permissive state; thus, acute inactivation during the TC-FR4 8 Hz schedule did not produce marked changes in lever-press rate (**Updated Fig.7 p-q**). In contrast, long-term suppression appears to disrupt this stability, creating a circuit state that is more sensitive to SST-IN inactivation and results in exaggerated lever-pressing behavior during the highest-frequency sequence schedule (**Updated Fig.7b**). We offer this interpretation in response to the reviewer's concern that the observed optogenetic and chemogenetic effects might simply represent byproducts of repeated inactivation with an assumption that the immediate and long-term effects should be identical to be meaningful. Our data demonstrate that this is not the case: the differences between immediate and long-term reflect distinct mechanisms. Specifically, cumulative suppression induces a permissive circuit state, whereas immediate inactivation reveals the ongoing contribution of SST-IN activity to the temporal organization of behavior.

We also understand that the reviewer's comments and questions particularly focused on the identification of immediate effects of SST-INs inactivation. In this context, our findings indicate that the sequence-structure measure (lever-press distribution probability) captures a different aspect of action control that

requires ongoing SST-IN activity without any cumulative, permissive effect of past optogenetic suppression (**Updated Fig.7r-v**).

To reinforce this latter point we updated analyses (**Fig.7o,s**, and **Extended Fig.7**) and summarized the key findings. Short-term inactivation only during the highest frequency time-constrained schedule primarily affected the lever-press distribution, indicating disruptions in the temporal structure of action sequences. In contrast, long-term/sustained inactivation combined with short-term inactivation disrupted both the temporal structure and altered the average lever-press rate (while long-term/sustained inactivation without short-term inactivation was ineffective) .

As the lever-press distribution was altered under both protocols, we infer that changes in the action-sequence structure likely represent the immediate contribution of SST-INs to our behavioral task in the absence of any past neuronal manipulation. If effects of activity suppression accumulate over time, they may eventually manifest as changes in lever-press rate, which we interpreted as a ‘permissive state’. We think this interpretation addresses the reviewer’s concern regarding the increased lever-press rate observed after long-term/sustained opto-inactivation but not during immediate inactivation in the absence of suppression history. Overall, we appreciate the reviewer’s suggestion, which helped us clarify this distinction more clearly in the revised manuscript.

Updated Manuscript (Line 327, orange text is updated):

...Additionally, SST optogenetic silencing elongated the lever-press probability distribution at action initiation (**Fig. 7e and f**), increased the proportions of incomplete sequences and decreased the percentage of complete sequences (**Fig. 7g and h**) compared to controls.

To determine whether the pronounced behavioral differences observed during the TC-FR4 8 Hz schedule (**Fig.7b-d, f-g**) required ongoing optogenetic inactivation during task performance, we introduced a periodic laser ON/OFF Modulation (PLM) protocol after animals completed TC-FR4 training at 1, 2, 4, and 8 Hz schedules (**Fig. 7i**) and examined how alternating laser ON/OFF conditions influenced sequence organization within a session. When we analyzed the action sequences generated during laser-OFF periods, the eNpHR group demonstrated significantly lower lever-press rate (**Fig. 7j**) and fewer lever presses per sequence (**Fig.7k**) compared with laser-ON periods. This modulation was not observed in the eYFP group (**Fig.7j,k**). Moreover, the elongated lever-press probability distribution was significantly altered by the Laser-ON/OFF control only in eNpHR group (**Extended Fig.7a-c**). While the proportion of complete SEQs remained unaffected within the PLM training schedule (**Fig.7l**), the proportion of incomplete SEQs within the eNpHR group during the laser-OFF periods was significantly lower than laser-ON periods (**Fig.7m**). Sequence efficiency was also reduced when the condition switched from laser-OFF to laser-ON (**Fig.7n**).

These findings demonstrate that dynamic laser ON/OFF control continued to modulate behavior, indicating that SST-IN activity remains actively engaged in organizing high-frequency action sequences. However, this finding does not necessarily indicate that the change in TC-FR4 8Hz task was only due to immediate optogenetic inactivation as it is possible that cumulative effects induced by long-term optogenetic suppression may also contribute to these behavioral changes.

To clarify the contribution of immediate optogenetic inactivation, we next investigated whether a single-session, immediate SST-IN inactivation could reproduce the behavioral differences observed after long-term optogenetic suppression. We applied a short-term optogenetic inactivation protocol exclusively during the 8 Hz schedule, without any prior treatment history (**Fig. 7o**). On the first day of TC-FR4 8 Hz training, SST-IN activity in eNpHR-expressing mice was silenced using lever-locked closed loop laser activation, and behavioral metrics were examined. Lever presses per minute did not differ significantly between eYFP and eNpHR mice (**Fig. 7p,q**). However, both the lever-press distribution probabilities and their corresponding area under the curve (AUC) showed significant differences between eYFP- and eNpHR-expressing mice (**Fig. 7r,s**), indicating that even a single session of SST-IN inactivation was sufficient to alter sequence structure.

To determine how these changes in lever-press distribution probabilities were reflected in sequence parameters, we compared each mouse's performance during the 8 Hz schedule against their own baseline performance at 4 Hz schedule (**Fig. 7o**). Both eYFP controls and eNpHR mice exhibited a significant Group \times Laser interaction for sequence parameters, including lever presses per sequence (**Fig. 7t**; two-way ANOVA, Group \times Laser interaction: $p = 0.0078$), sequence duration (**Fig. 7u**; $p = 0.0174$), and sequence efficiency (**Fig. 7v**; $p = 0.0206$). Post hoc tests confirmed that the laser-training effect was specific to the eNpHR group. However, the main effects of group (eYFP vs. eNpHR) were not significant for lever presses per sequence ($p = 0.1194$), sequence duration ($p = 0.0950$), or sequence efficiency ($p = 0.1005$). We further introduced the PLM protocol and found similar alteration in lever-press probability distributions (**Extended Fig. 7 d,e**) and significant laser-dependent modulation in sequence parameters (**Extended Fig. 7f-h**). Thus, a single-session, immediate optogenetic inactivation produced changes in sequence structure (**Fig. 7s-v**), but was not sufficient to alter overall lever-press rate (**Fig. 7p,q**).

In summary, both immediate and long-term SST-IN inactivation disrupted the temporal organization of lever-press sequences (**Fig. 7f,s**). However, significant alteration of lever-press rate was observed only after long-term optogenetic suppression (**Fig. 7b, j**). This suggests that cumulative suppression of SST-INs may create a condition permissive for overly exuberant lever pressing. In our TC-FR4 training, this effect was only observed when SST-IN activity was inactivated during high-frequency sequence execution, even following long-term SST-IN inactivation (**Fig. 7b**).

- Interpreting the activity changes across FR1 learning

My concern here has not been well addressed. I understand the reasons why we might not see an effect of inactivation on behavior in certain cases. But we are still left without positive evidence that the activity of these neurons is important for the easier versions of the task. This is problematic because much of the paper regards these easier versions of the task, like the changes across learning shown in Figure 2. The manuscript should explicitly describe the sense in which these changes are meaningful, in the absence of evidence that the activity is important for behavior.

Also, to be clear, the discrepancy referred to on line 413 is not the discrepancy between the change in SST activity and the absence of a functional effect, as the authors state in their rebuttal on this issue. Instead, the discrepancy they refer to on line 413 is one between the authors' inactivation results and those from other studies. Thus, the passage beginning on line 413 does not directly address my concern.

Response:

We thank the reviewer for raising this important point and for prompting us to clarify the interpretation of SST-IN activity during early FR1 training. In our data, SST-INs exhibited robust, action-locked calcium responses during early sessions; however, optogenetic inactivation produced no measurable behavioral effect, as the reviewer rightly points out. In our first rebuttal, we outlined possible explanations for why the effects of optogenetic inactivation might not be detected in this schedule: (1) SST-IN activity may have a functional role, but this role might not be captured by our behavioral metrics; (2) SST-INs may operate via fundamentally different neural mechanisms across tasks. Previously, we mainly focused on explaining the absence of the optogenetic inactivation effect during FR1 or easier tasks by comparing our tasks and behavior conditions with those of previous studies. However, we now understand that the reviewer is asking us to specify (or discuss) the behavioral conditions under which these early FR1 activities might be meaningful.

There are several possibilities, as the reviewer clearly understands, neural activation does not necessarily imply functional causality, as we discuss below. Particularly, cortical dynamics often display widespread activation, however, only specific subsets functionally contribute to the behavior. Thus, strong neural activity during a behavior does not guarantee that a population is causally necessary for that behavior.

For example, in Zatzka-Haas et al. (2021), wide-field calcium imaging revealed extensive, cortex-wide activity during a visual decision-making task. Primary motor (MOp) and somatosensory (SSp) cortices were highly active and strongly correlated with action execution. However, while optogenetic silencing in some of these regions had significant effects, silencing in other regions had only minor effects on performance, indicating that although these areas are engaged during movement, they are not essential for executing the action itself.

Similarly, in Musall et al. (Nature Neuroscience, 2022), the authors used cell-type-specific wide-field imaging to differentiate pyramidal neuron subtypes during an auditory decision task. In parietal cortex, all pyramidal neurons (PyNs) were recruited during stimulus perception, but inactivation experiments revealed that pyramidal tract (PT) neurons had the largest causal influence on task performance, whereas inactivating intratelencephalic (IT) neurons had only modest effects. Thus, widespread activation does not indicate the specific populations that exert true behavioral control.

Taken together, these studies highlight the importance of distinguishing between neural activity and their functional necessity. From this perspective, we cannot assume that all observed activation has functional implications. Nonetheless, we could only compare SST-IN activity under different task demand conditions if we recorded under all conditions. Thus, while we cannot draw any conclusions concerning the role of SST-IN activity in task performance under some of the operant schedules, the activity data are still useful for comparison to activity during schedules when inactivation altered behavior. For example, the emergence of delayed-onset increases in activity is one feature that develops with TC-FR4 sequence training and can be compared with activity in both early and late FR1 training to determine what patterns of SST-IN activity occur under conditions where activity suppression has clear behavioral effects.

Interpretation within Our Framework

We initially hesitated to include speculative statements without direct evidence in the manuscript. However, as the reviewer suggested providing a discussion, we can now summarize possible ideas again to address the reviewer's concern. Critical points were already mentioned in our original manuscript, but we additionally emphasize this context:

1. **Reflection of broad cortical engagement during early learning:**

Early in motor skill acquisition, primary motor cortex is broadly recruited for task initiation, performance, and learning (Peters et al., Nature 2017; Huber et al., Nature 2012; Kawai et al., Neuron 2015). Such widespread activation likely reflects generalized cortical coordination, performance monitoring, or sensorimotor integration, rather than region-specific motor output. In this context, early SST-IN activity in M1 may represent participation in a broadly engaged cortical state that becomes progressively refined and pruned as the task becomes automatized.

2. **Sensory and associative signaling:**

SST-INs in M1 may initially respond to sensory or associative information related to the lever-press context—such as lever recognition, movement initiation, or reward anticipation. These signals may accompany early learning but are not essential for simple task execution. As training progresses and task representations consolidate, these associative responses likely diminish, consistent with the observed reduction in SST activity.

3. **Subtle motor modulation exists, but below detection threshold in our behavioral metrics:**

It is also possible that SST-INs subtly influence fine-scale aspects of forelimb kinematics or movement precision during early training—effects that our behavioral metrics could not detect under freely moving conditions. This is a completely different idea from prior suggestions 1 and 2 above, suggesting that these neurons were indeed functionally important during the early FR1 stage, even though their impact was just not captured by our metrics. Based on previous studies [Simon X Chen, Nature Neuroscience 2015], we believe this is likely, and we have already described this point in detail in our previous rebuttal and original manuscript as the reviewer noted. In our data, SST-INs exhibited action-related calcium activity during TC-FR4 training (**Extended Fig. 5g–j**); however, their inhibition did not impair task performance under this schedule (**Fig. 7b** and **Extended Fig. 7i–l**). Similarly, SST-INs displayed action-related activity during the early FR1 stage, but the functional relevance may not be measurable in our behavioral assays if such activity is not critical for the given task—consistent with the idea that SST-IN function has schedule-dependent implications during TC-FR4 training. Nevertheless, the observed activity patterns during the early FR1 still serve as a useful reference for understanding how these neural activities change—and when such changes become meaningful and functionally relevant—within our experimental framework. As this point is already described in detail in the main manuscript (lines 465–503), we have added the following context to further clarify it:

Updated Manuscript (Line 414, orange text is updated):

Similarly, we observed decreased M1 SST IN activity and correlation during the execution of the well-learned FR1 task. As cell-specific inhibition during the FR1 task produced no measurable behavioral effect, these changes in SST IN activity may reflect the broader disengagement feature of M1, rather than region-specific motor output. Although cortical dynamics often display widespread activation, only specific subsets functionally contribute to the behavior. In the more complex, time-constrained action sequences (e.g. TC-FR4), we observed that a strong correlation was maintained between SST INs activation and action sequence execution as task demands increased and proficiency decreased (**Fig.3**).

- Figure 1: When using calcium imaging, one must be careful when interpreting differences in dF/F magnitudes across different cell types...

The issue here is that because calcium indicators serve as calcium buffers, their expression level influences the magnitude of intracellular free calcium changes after action potentials, which in turn affects the dF/F change caused by each spike. The “background fluorescence” and “non-action related calcium signals” the authors refer to are not the primary problem with comparing across cell types. Cell types could differ systematically in their indicator expression level, or in the expression of endogenous calcium buffers. Both will affect the size of spike-induced dF/F changes, confounding comparisons across cell types.

Response:

We thank the reviewer for raising this fundamental question regarding calcium imaging. We agree that comparison of dF/F changes across neurons is difficult. Thus, we did not make comparisons of this parameter between SST-INs and pyramidal neurons. While we did compare action time-locked calcium increases across superficial and deep layers (taking into account the caveats about possible superficial layer damage in the deep layer measurements) there may be some misunderstanding of the interpretation of these data (presented in Figs. 1k-q). The data shown in Fig1k are averages of many trials time-locked to lever-pressing for the two neuronal subtypes. The lower dF/F values in the SST L2/3 neurons do not reflect the fact that each individual neuron or bout of activity shows smaller amplitude lever-press-associated responses. Indeed, as shown in Figs. 1i and 1o, L2/3 neurons show dF/F changes in relation to individual lever presses and during periods with no presses that are well within the range of the largest amplitude responses observed in the SST L5/6 neurons. The difference in average dF/F shown in Fig. 1k is due to the relative inconsistency of generation of large amplitude lever-press-related responses in L2/3 compared to L5/6. This inconsistency is unlikely to be due to differences in indicator expression levels or buffering, as the SST-INs in both layers are capable of producing comparable amplitude transients. The difference we highlight in Fig. 1k is instead related specifically to differences in the amplitude and consistency of lever-press related neuronal activation. It is possible that one class of neuron may be better able to show repeated bursts in response to temporally contiguous presses, but nothing in our data indicate such a difference.

In practice, we applied a consistent analysis approach, so any subtype-dependent differences are reflected in the population average of dF/F . Given that a few dozen cells were captured per field of view (Extended Figure 1a), no single subtype would dominate the population average. Furthermore, in our experiments, action-related SST-INs consistently exhibited similar dF/F amplitude (Figure 1d). Additionally, non-action-locked activity tends to cancel out in our analysis, therefore any variability in GCaMP expression across cells would be naturally compensated to some extent. We mainly used SST-Cre mice. SST-expressing neurons might include multiple genetic sub-types. However, there is limited information on whether GCaMP expression differs systematically across these subtypes.

We have provided alternative analyses with multiple measures that are independent of cell-specific dF/F magnitude, including calcium event rates and probabilities (Figures 2h–k, 3i–p), correlations (Extended Figure 1e,f; Figures 2p, 3u), peak timing (Figure 5j), and normalized spontaneous activity within individual cells (Figures 2f, 3k; Extended Figure 5g,i). PYR neurons may exhibit greater variability, which makes correlation analysis particularly useful, as it does not depend on signal amplitude. Nevertheless, we still present dF/F values, as we believe that the raw event-locked dF/F provides informative insights into the relationship between neuronal activity and action production. We are not aware of other calcium imaging studies that present such a comprehensive set of metrics to validate the same concept within each experiment.

Nevertheless, as noted in the discussion, we suggested that alternative approaches such as “in vivo electrophysiological recordings” will be useful to complement and validate the calcium imaging data. We do note that our layer 5/6 SST-INs exhibit consistent and similar activity profiles, indicating that cell type-dependent dF/F variability was minimal compared to other calcium imaging studies.

As we are aware of the limitations of using dF/F as an absolute indicator, we believe following information could compensate somewhat the reviewer’s concern.

Updated Manuscript (from Line 136, orange text is updated)

In contrast, activity in layer 5/6 showed more consistent increases, including relatively consistent transient amplitudes, around each press (Fig. 1o). **Nevertheless, there are limitations to comparing cell-to-cell activity solely based on $\Delta F/F_0$ values.** To examine this characteristic at the network level independently of $\Delta F/F_0$ amplitude, we assessed Pearson’s correlation coefficients across cells. This parameter calculates the linear relationship between a single cell calcium trace and all other neuronal traces during a 10 second time window centered around the lever-press.

Reviewer #4 (Remarks to the Author):

Again no major comments. However my first minor comment (to explicitly say what layer was recorded) was specifically about the figure legends - I would suggest to add a statement of what layer was recorded to Figures 2, 3, etc, in either the main figure panel or in the legend. It may say so in the main text, but there will be no ambiguity if it is in the figure itself.

Response:

We thank Reviewer #4 for the careful follow-up and helpful clarification. We appreciate the suggestion to specify the recorded cortical layers directly in the figure panels or legends to avoid ambiguity. Following this advice, we have updated the relevant figure legends (Figures 2, 3, and others as appropriate) to clearly indicate the recording layers, ensuring consistency and clarity throughout the manuscript.

Updated Manuscript Line 146 :

The remainder of the study focused on L5/6, a layer that plays a central role in motor output control but remains less well characterized than L2/3 in previous imaging studies [24,25,26].

Updated Figure Captions:

Figure 2. Action-locked SST-IN activity in M1 L5/6 correlate with action acquisition and consolidation

Figure 3. SST IN activity in M1 L5/6 is redistributed during execution of complex sequence

Extended Figure 3. SST-IN activity in M1 L5/6 was stabilized while the similar action sequence structure was maintained.

Figure5. Two distinct activity patterns of SST-INs in M1 L5/6 encode the initiation and trial-by-trial structural modulation of complex action sequences.